# Faster Algorithms for Structured Linear and Kernel Support Vector Machines

**Yuzhou Gu**
New York University
yuzhougu@nyu.edu

**Zhao Song**
Simons Institute for the Theory of Computing, UC Berkeley
magic.linuxkde@gmail.com

**Lichen Zhang**
MIT CSAIL
lichenz@csail.mit.edu

## Abstract

Quadratic programming is a ubiquitous prototype in convex programming. Many machine learning problems can be formulated as quadratic programming, including the famous Support Vector Machines (SVMs). Linear and kernel SVMs have been among the most popular models in machine learning over the past three decades, prior to the deep learning era.

Generally, a quadratic program has an input size of $\Theta(n^2)$, where $n$ is the number of variables. Assuming the Strong Exponential Time Hypothesis (SETH), it is known that no $O(n^{2-o(1)})$ time algorithm exists when the quadratic objective matrix is positive semidefinite (Backurs, Indyk, and Schmidt, NeurIPS'17). However, problems such as SVMs usually admit much smaller input sizes: one is given $n$ data points, each of dimension $d$, and $d$ is oftentimes much smaller than $n$. Furthermore, the SVM program has only $O(1)$ equality linear constraints. This suggests that faster algorithms are feasible, provided the program exhibits certain structures.

In this work, we design the first nearly-linear time algorithm for solving quadratic programs whenever the quadratic objective admits a low-rank factorization, and the number of linear constraints is small. Consequently, we obtain results for SVMs:

- For linear SVM when the input data is $d$-dimensional, our algorithm runs in time $\widetilde{O}(nd^{(\omega+1)/2}\log(1/\epsilon))$ where $\omega \approx 2.37$ is the fast matrix multiplication exponent;
- For Gaussian kernel SVM, when the data dimension $d = O(\log n)$ and the squared dataset radius is sub-logarithmic in $n$, our algorithm runs in time $O(n^{1+o(1)}\log(1/\epsilon))$. We also prove that when the squared dataset radius is at least $\Omega(\log^2 n)$, then $\Omega(n^{2-o(1)})$ time is required. This improves upon the prior best lower bound in both the dimension $d$ and the squared dataset radius.

## 1 Introduction

Quadratic programming (QP) represents a class of convex optimization problems that optimize a quadratic objective over the intersection of an affine subspace and the non-negative orthant[1]. QPs naturally extend linear programming by incorporating a quadratic objective, and they find extensive applications in operational research, theoretical computer science, and machine learning (Kozlov et al., 1979; Wright, 1999; Gould & Toint, 2000; Gould et al., 2001; Propato & Uber, 2004; Cornuejols & Tütüncü, 2006). The quadratic objective introduces challenges: QPs with a general (not necessarily positive semidefinite) symmetric quadratic objective matrix are NP-hard to solve (Sahni, 1974; Pardalos & Vavasis, 1991). When the quadratic objective matrix is positive semidefinite, the problem becomes weakly polynomial-time solvable, as it can be reduced to convex empirical risk minimization (Lee et al., 2019) (refer to Section C for further discussion).

---

[1]There are classes of QPs with quadratic constraints as well. However, in this paper, we focus on cases where the constraints are linear.

Formally, the QP problem is defined as follows:

**Definition 1.1** (Quadratic Programming). Given an $n \times n$ symmetric, positive semidefinite objective matrix $Q$, a vector $c \in \mathbb{R}^n$, and a polytope described by a pair $(A \in \mathbb{R}^{m \times n}, b \in \mathbb{R}^m)$, the linearly constrained quadratic programming (LCQP) or simply quadratic programming (QP) problem seeks to solve the following optimization problem:

$$\min_{x \in \mathbb{R}^n} \quad \frac{1}{2} x^\top Q x + c^\top x \tag{1}$$
$$\text{s.t.} \quad Ax = b$$
$$x \geq 0.$$

A classic application of QP is the Support Vector Machine (SVM) problem (Boser et al., 1992; Cortes & Vapnik, 1995). In SVMs, a dataset $x_1, \ldots, x_n \in \mathbb{R}^d$ is provided, along with corresponding labels $y_1, \ldots, y_n \in \{\pm 1\}$. The objective is to identify a hyperplane that separates the two groups of points with opposite labels, while maintaining a large margin from both. Remarkably, this popular machine learning problem can be formulated as a QP and subsequently solved using specialized QP solvers (Muller et al., 2001). Thus, advancements in QP algorithms could potentially lead to runtime improvements for SVMs.

Despite its practical and theoretical significance, algorithmic quadratic programming has garnered relatively less attention compared to its close relatives in convex programming, such as linear programming (Cohen et al., 2019; Jiang et al., 2021; Brand, 2020; Song & Yu, 2021), convex empirical risk minimization (Lee et al., 2019; Qin et al., 2023), and semidefinite programming (Jiang et al., 2020; Huang et al., 2022; Gu & Song, 2022). In this work, we take a pioneering step in developing fast and robust interior point-type algorithms for solving QPs. We particularly focus on improving the runtime for high-precision hard- and soft-margin SVMs. For the purposes of this discussion, we will concentrate on hard-margin SVMs, with the understanding that our results naturally extend to soft-margin variants. We begin by introducing the hard-margin linear SVMs:

**Definition 1.2** (Linear SVM). Given a dataset $X \in \mathbb{R}^{n \times d}$ and a collection of labels $y_1, \ldots, y_n$ each in $\pm 1$, the *linear SVM problem* requires solving the following quadratic program:

$$\max_{\alpha \in \mathbb{R}^n} \quad \mathbf{1}_n^\top \alpha - \frac{1}{2} \alpha^\top (yy^\top \circ XX^\top) \alpha, \tag{2}$$
$$\text{s.t.} \quad \alpha^\top y = 0,$$
$$\alpha \geq 0.$$

where $\circ$ denotes the Hadamard product.

It should be noted that this formulation is actually the *dual* of the SVM optimization problem. The primal program seeks a vector $w \in \mathbb{R}^d$ such that

$$\min_{w \in \mathbb{R}^d} \quad \frac{1}{2} \|w\|_2^2,$$
$$\text{s.t.} \quad y_i(w^\top x_i - b) \geq 1, \qquad \forall i \in [n],$$

where $b \in \mathbb{R}$ is the bias term. Given the solution $\alpha \in \mathbb{R}^n$, one can conveniently convert it to a primal solution: $w^* = \sum_{i=1}^n \alpha_i^* y_i x_i$. At first glance, one might be inclined to solve the primal problem directly, especially in cases where $d \ll n$, as it presents a lower-dimensional optimization problem compared to the dual. The dual formulation becomes particularly advantageous when solving the kernel SVM, which maps features to a high or potentially infinite-dimensional space.

**Definition 1.3** (Kernel SVM). Given a dataset $X \in \mathbb{R}^{n \times d}$ and a positive definite kernel function $\mathsf{K} : \mathbb{R}^d \times \mathbb{R}^d \to \mathbb{R}$, let $K \in \mathbb{R}^{n \times n}$ denote the kernel matrix, where $K_{i,j} = \mathsf{K}(x_i, x_j)$. With a collection of labels $y_1, \ldots, y_n$ each in $\{\pm 1\}$, the *kernel SVM problem* requires solving the following quadratic program:

$$\max_{\alpha \in \mathbb{R}^n} \quad \mathbf{1}_n^\top \alpha - \frac{1}{2} \alpha^\top (yy^\top \circ K) \alpha, \tag{3}$$
$$\text{s.t.} \quad \alpha^\top y = 0,$$
$$\alpha \geq 0.$$

The positive definite kernel function $\mathsf{K}$ corresponds to a feature mapping, implying that $\mathsf{K}(x_i, x_j) = \phi(x_i)^\top \phi(x_j)$ for some $\phi : \mathbb{R}^d \to \mathbb{R}^s$. Thus, solving the primal SVM can be viewed as solving the optimization problem on the transformed dataset. However, the primal program's dimension depends on the (transformed) data's dimension $s$, which can be infinite. Conversely, the dual program, with dimension $n$, is typically easier to solve. Throughout this paper, when discussing the SVM program, we implicitly refer to the dual quadratic program, not the primal.

One key aspect of the SVM program is its minimal equality constraints. Specifically, for both linear and kernel SVMs, there is only a single equality constraint of the form $\alpha^\top y = 0$. This constraint arises naturally from the bias term in the primal SVM formulation and its Lagrangian. The limited number of constraints enables us to design QP solvers with favorable dependence on the number of data points $n$, albeit with a higher dependence on the number of constraints $m$, thus offering effective end-to-end guarantees for SVMs.

Previous efforts to solve the SVM program typically involve breaking down the large QP into smaller, constant-sized QPs. These algorithms, while easy to implement and well-suited to modern hardware architectures, oftentimes lack tight theoretical analysis and the estimation of iteration count is usually pessimistic (Chang & Lin, 2011). Theoretically, Joachims (2006) systematically analyzed this class of algorithms, demonstrating that to achieve an $\epsilon$-approximation solution, $\widetilde{O}(\epsilon^{-2} B \cdot \mathrm{nnz}(A))$ time is sufficient, where $B$ is the squared-radius of the dataset and $\mathrm{nnz}(\cdot)$ denotes the number of nonzero entries. This is subsequently improved in Shalev-Shwartz et al. (2011) with a subgradient-based method that runs in $\widetilde{O}(\epsilon^{-1} d)$ time. Unfortunately, the polynomial dependence on the precision $\epsilon^{-1}$ makes them hard to be adapted for even moderately small $\epsilon$. For example, when $\epsilon$ is set to be $10^{-3}$ to account for the usual machine precision errors, these algorithms would require at least $10^3$ iterations to converge.

To develop a high-precision algorithm with $\mathrm{poly}\log(\epsilon^{-1})$ dependence instead of $\mathrm{poly}(\epsilon^{-1})$, we focus on second-order methods for QPs. A variety of approaches have been explored in previous works, including the interior point method (Karmarkar, 1984), active set methods (Murty, 1988), augmented Lagrangian techniques (Delbos & Gilbert, 2003), conjugate gradient, gradient projection, and extensions of the simplex algorithm (Dantzig, 1955; Wolfe, 1959; Murty, 1988). Our interest is particularly piqued by the interior point method (IPM). Recent advances in the robust IPM framework have led to significant successes for convex programming problems (Cohen et al., 2019; Lee et al., 2019; Brand, 2020; Jiang et al., 2020; Brand et al., 2020; Jiang et al., 2021; Song & Yu, 2021; Jiang et al., 2022; Huang et al., 2022; Gu & Song, 2022; Qin et al., 2023; Song et al., 2023). These successes are a result of combining robust analysis of IPM with dedicated data structure design.

Applying IPM to solve QPs with a constant number of constraints is not entirely novel; existing work (Ferris & Munson, 2002) has already adapted IPM to solve the linear SVM problem. However, the runtime of their algorithm is sub-optimal. Each iteration of their algorithm requires multiplying a $d \times n$ matrix with an $n \times d$ matrix in $O(nd^{\omega-1})$ time, where $\omega \approx 2.37$ is the fast matrix multiplication exponent (Duan et al., 2023; Williams et al., 2024; Gall, 2024; Alman et al., 2025). Moreover, the IPM requires $O(\sqrt{n}\log(1/\epsilon))$ iterations to converge. This ends up with an overall runtime $O(n^{1.5} d^{\omega-1} \log(1/\epsilon))$, which is super-linear in the dataset size even when the dimension $d$ is small. In practical scenarios where $n$ is usually large, the $n^{1.5}$ dependence becomes prohibitive. Therefore, it is crucial to develop an algorithm with almost- or nearly-linear dependence on $n$ and logarithmic dependence on $\epsilon^{-1}$.

For linear SVM, we propose a nearly-linear time algorithm with high-precision guarantees, applicable when the dimension of the dataset is smaller than the number of points:

**Theorem 1.4** (Low-rank QP and Linear SVM, informal version of Theorem E.1)**.** *Given a quadratic program as defined in Definition 1.1, and assuming a low-rank factorization of the quadratic objective matrix $Q = UV^\top$, where $U, V \in \mathbb{R}^{n \times k}$, there exists an algorithm that can solve the program* (1) *up to $\epsilon$-error[2] in $\widetilde{O}(n(k+m)^{(\omega+1)/2} \log(n/\epsilon))$ time.*

*Specifically, for linear SVM (as per Definition 1.2) with $d \le n$, one can solve program* (2) *up to $\epsilon$-error in $\widetilde{O}(nd^{(\omega+1)/2} \log(n/\epsilon))$ time.*

---

[2] We say an algorithm that solves the program up to $\epsilon$-error if it returns an approximate solution vector $\widetilde{\alpha}$ whose objective value is at most $\epsilon$ more than the optimal objective value.

While a nearly-linear time algorithm for linear SVMs is appealing, most applications look at kernel SVMs as they provide more expressive power to the linear classifier. This poses significant challenge in algorithm design, as forming the kernel matrix exactly would require $\Omega(n^2)$ time. Moreover, the kernel matrix could be full-rank without any structural assumptions, rendering our low-rank QP solver inapplicable. In fact, it has been shown that for data dimension $d = \omega(\log n)$, no algorithm can approximately solve kernel SVM within an error $\exp(-\omega(\log^2 n))$ in time $O(n^{2-o(1)})$, assuming the famous Strong Exponential Time Hypothesis (SETH)[3] (Backurs et al., 2017).

Conversely, a long line of works aim to speed up computation with the kernel matrix faster than quadratic, especially when the kernel has certain smooth and Lipschitz properties (Alman et al., 2020; Aggarwal & Alman, 2022; Bakshi et al., 2023; Charikar et al., 2024). For instance, when kernel functions are sufficiently smooth, efficient approximation using low-degree polynomials is feasible, leading to an approximate low-rank factorization. A prime example is the Gaussian RBF kernel, where Aggarwal & Alman (2022) showed that for dimension $d = \Theta(\log n)$ and squared dataset radius (defined as $\max_{i,j\in[n]} \|x_i - x_j\|_2^2$) $B = o(\log n)$, there exists low-rank matrices $U, V \in \mathbb{R}^{n \times n^{o(1)}}$ such that for any vector $x \in \mathbb{R}^n$, $\|(K - UV^\top)x\|_\infty \leq \epsilon\|x\|_1$. They subsequently develop an algorithm to solve the Batch Gaussian KDE problem in $O(n^{1+o(1)})$ time.

Based on this dichotomy in fast kernel matrix algebra, we establish two results: 1) Solving Gaussian kernel SVM in $O(n^{1+o(1)} \log(1/\epsilon))$ time is feasible when $B = o(\frac{\log n}{\log\log n})$, and 2) Assuming SETH, no sub-quadratic time algorithm exists for $B = \Omega(\log^2 n)$ in SVMs without bias and $B = \Omega(\log^6 n)$ in SVMs with bias. This improves the lower bound established by Backurs et al. (2017) in terms of dimension $d$.

**Theorem 1.5** (Gaussian Kernel SVM, informal version of Theorem G.7 and G.12). *Given a dataset $X \in \mathbb{R}^{n \times d}$ with dimension $d$ and squared radius denoted by $B$, let $\mathsf{K}(x_i, x_j) = \exp(-\|x_i - x_j\|_2^2)$ be the Gaussian kernel function. Then, for the kernel SVM problem defined in Definition 1.3,*

- *If $d = O(\log n), B = o(\frac{\log n}{\log\log n})$, there exists an algorithm that solves the Gaussian kernel SVM up to $\epsilon$-error in time $O(n^{1+o(1)} \log(1/\epsilon))$;*

- *If $d = \Omega(\log n), B = \Omega(\log^2 n)$, then assuming SETH, any algorithm that solves the Gaussian kernel SVM without a bias term up to $\exp(-\omega(\log^2 n))$ error would require $\Omega(n^{2-o(1)})$ time;*

- *If $d = \Omega(\log n), B = \Omega(\log^6 n)$, then assuming SETH, any algorithm that solves the Gaussian kernel SVM with a bias term up to $\exp(-\omega(\log^2 n))$ error would require $\Omega(n^{2-o(1)})$ time.*

To our knowledge, this is the first almost-linear time algorithm for Gaussian kernel SVM even when $d = \log n$ and the radius is small. Our algorithm effectively utilizes the rank-$n^{o(1)}$ factorization of the Gaussian kernel matrix alongside our low-rank QP solver.

## 1.1 RELATED WORK

**Support Vector Machines.** SVM, one of the most prominent machine learning models before the rise of deep learning, has a rich literature dedicated to its algorithmic speedup. For linear SVM, Joachims (2006) offers a first-order algorithm that solves its QP in nearly-linear time, but with a runtime dependence of $\epsilon^{-2}$, limiting its use in high precision settings. This runtime is later significantly improved by Shalev-Shwartz et al. (2011) to $\widetilde{O}(\epsilon^{-1}d)$ via a stochastic subgradient descent algorithm. For SVM classification, existing algorithms such as SVM-Light (Joachims, 1999), SMO (Platt, 1998), LIBSVM (Chang & Lin, 2011), and SVM-Torch (Collobert & Bengio, 2001) perform well in high-dimensional data settings. However, their runtime scales super-linearly with $n$, making them less viable for large datasets. Previous investigations into solving linear SVM via interior point methods (Ferris & Munson, 2002) have been somewhat basic, leading to an overall

---

[3]SETH is a standard complexity theoretical assumption (Impagliazzo et al., 1998; Impagliazzo & Paturi, 2001). Informally, it states that for a Conjunctive Normal Form (CNF) formula with $m$ clauses and $n$ variables, there is no algorithm for checking its feasibility in time less than $O(c^n \cdot \text{poly}(m))$ for $c < 2$.

runtime of $O(n^{1.5}d^{\omega-1}\log(1/\epsilon))$. For a more comprehensive survey on efficient algorithms for SVM, refer to Cervantes et al. (2020). On the hardness side, Backurs et al. (2017) provides an efficient reduction from the Bichromatic Closest Pair problem to Gaussian kernel SVM, establishing an almost-quadratic lower bound assuming SETH.

**Interior Point Method.** The interior point method, a well-established approach for solving convex programs under constraints, was first proposed by Karmarkar (1984) as a (weakly) polynomial-time algorithm for linear programs, later improved by Vaidya (1989) in terms of runtime. Recent work by Cohen et al. (2019) has shown how to solve linear programs with interior point methods in the current matrix multiplication time, utilizing a robust IPM framework. Subsequent studies (Lee et al., 2019; Brand, 2020; Jiang et al., 2021; Song & Yu, 2021; Huang et al., 2022; Jiang et al., 2022; Qin et al., 2023) have further refined their algorithm or applied it to different optimization problems.

**Kernel Matrix Algebra.** Kernel methods, fundamental in machine learning, enable feature mappings to potentially infinite dimensions for $n$ data points in $d$ dimensions. The kernel matrix, a crucial component of kernel methods, often has a prohibitive quadratic size for explicit formation. Recent active research focuses on computing and approximating kernel matrices and related tasks in sub-quadratic time, such as kernel matrix-vector multiplication, spectral sparsification, and Laplacian system solving. The study by Alman et al. (2020) introduces a comprehensive toolkit for solving these problems in almost-linear time for small dimensions, leveraging techniques like polynomial methods and ultra Johnson-Lindenstrauss transforms. Alternatively, Backurs et al. (2021); Bakshi et al. (2023) reduce various kernel matrix algebra tasks to kernel density estimation (KDE), which recent advancements in KDE data structures (Charikar & Siminelakis, 2017; Backurs et al., 2018; Charikar et al., 2020) have made more efficient. A recent contribution by Aggarwal & Alman (2022) provides a tighter characterization of the low-degree polynomial approximation for the $e^{-x}$ function, leading to more efficient algorithms for the Batch Gaussian KDE problem. Another line of works aim to apply *oblivious sketching* to the kernel matrix, so that a spectral approximation can be computed in time nearly-linear in the dataset size and a low-rank factorization with rank being the *statistical dimension* of the problem can be computed (Avron et al., 2014; Ahle et al., 2020; Zandieh et al., 2021; Woodruff & Zandieh, 2022). Unfortunately, for the statistical dimension to be smaller than $n$, one has to add a regularization parameter $\lambda$, which usually appears when one wants to solve a kernel ridge regression. The work (Song et al., 2021) does not require the parameter $\lambda$, but it obtains runtime improvements for $d \geq n$, and the sketching dimension is super-linear in $n$.

## 2 Technique Overview

In this section, we provide an overview of the techniques employed in our development of two nearly-linear time algorithms for structured QPs. In Section 2.1, we detail the robust IPM framework, which forms the foundation of our algorithms. Subsequent section, namely Section 2.2 and 2.3, delves into dedicated data structures designed for efficiently solving low-treewidth and low-rank QPs, respectively. Finally, in Section 2.4, we discuss the adaptation of these advanced QP solvers for both linear and kernel SVMs.

Due to the heavily-technical nature, we recommend that in the first read, the audience can skip Section 2.2 and 2.3.

### 2.1 General Strategy

Our algorithm is built upon the robust IPM framework, an efficient variant of the primal-dual central path method (Renegar, 1988). This framework maintains a primal-dual solution pair $(x, s) \in \mathbb{R}^n \times \mathbb{R}^n$. To understand the central path for QPs, we first consider the central path equations for linear programming (see Cohen et al. (2019); Lee et al. (2019) for reference):

$$s/t + \nabla\phi(x) = \mu,$$
$$Ax = b,$$
$$A^\top y + s = c,$$

where $x$ is the primal variable, $s$ is the slack variable, $y$ is the dual variable, $\phi(x)$ is a self-concordant barrier function, and $\mu$ denotes the error. The central path is defined by the trajectory of $(x, s)$ as $t$ approaches 0.

In quadratic programming, we modify these equations:

$$s/t + \nabla\phi(x) = \mu,$$
$$Ax = b,$$
$$-Qx + A^\top y + s = c,$$

where $Q$ is the positive semidefinite objective matrix. The key difference in the central path equations for LP and QP is the inclusion of the $-Qx$ term in the third equation, significantly affecting algorithm design.

Fundamentally, IPM is a Newton's method in which we update the variables $x, y$ and $s$ through the second-order information from the self-concordant barrier function. We derive the update rules for QP (detailed derivation in Section F.2):

$$\delta_x = tM^{-1/2}(I - P)M^{-1/2}\delta_\mu,$$
$$\delta_y = -t(AM^{-1}A^\top)^{-1}AM^{-1}\delta_\mu,$$
$$\delta_s = t\delta_\mu - t^2 HM^{-1/2}(I - P)M^{-1/2}\delta_\mu,$$
$$\text{where} \quad H = \nabla^2\phi(x), \qquad M = Q + tH,$$
$$P = M^{-1/2}A^\top(AM^{-1}A^\top)^{-1}AM^{-1/2},$$

where $\delta_x$, $\delta_y$, $\delta_s$, and $\delta_\mu$ are the incremental steps for $x$, $y$, $s$, and $\mu$, respectively.

The robust IPM approximates these updates rather than computing them exactly. It maintains an approximate primal-dual solution pair $(\overline{x}, \overline{s}) \in \mathbb{R}^n \times \mathbb{R}^n$ and computes the steps using this approximation. Provided the approximation is sufficiently accurate, it can be shown (see Section F for more details) that the algorithm converges efficiently to the optimal solution along the robust central path.

Therefore, the critical challenge lies in efficiently maintaining $(\overline{x}, \overline{s})$, an approximation to $(x, s)$, when $(x, s)$ evolves following the robust central path steps. The primary difficulty is that explicitly managing the primal-dual solution pair $(x, s)$ is inefficient due to potential dense changes. Such changes can lead to dense updates in $H$, slowing down the computation of steps. The innovative aspect of robust IPM is recognizing that $(x, s)$ are only required at the algorithm's conclusion, not during its execution. Instead, we can identify entries with significant changes and update the approximation $(\overline{x}, \overline{s})$ correspondingly. With IPM's lazy updates, only a nearly-linear number of entries are adjusted throughout the algorithm:

$$\sum_{t=1}^{T} \|\overline{x}^{(t)} - \overline{x}^{(t-1)}\|_0 + \|\overline{s}^{(t)} - \overline{s}^{(t-1)}\|_0 = \widetilde{O}(n\log(1/\epsilon))$$

where $T = \widetilde{O}(\sqrt{n}\log(1/\epsilon))$ is the number of iterations for IPM convergence. This indicates that, on average, each entry of $\overline{x}$ and $\overline{s}$ is updated $\log(1/\epsilon)$ times, facilitating rapid updates to these quantities and, consequently, to $H$.

In the special case where $Q = 0$, the path reverts to the LP case, with $M = tH$ being a diagonal matrix, allowing for efficient computation and updates of $M^{-1}$. This simplifies maintaining $AM^{-1}A^\top$, as updates to $M^{-1}$ correspond to row and column scaling of $A$. However, in the QP scenario, where $M$ is symmetric positive semidefinite, maintaining the term $AM^{-1}A^\top$ becomes more complex. Nevertheless, when the number of constraints is small, as in SVMs, this issue is less problematic. Yet, even with this simplification, the challenge is far from trivial, given the presence of terms like $M^{-1/2}$ in the robust central path steps. While the matrix Woodbury identity could be considered, it falls short when maintaining a square root term. Despite these hurdles, we construct efficient data structures for $M^{-1/2}$ maintenance when $Q$ possesses succinct representations, such as low-rank.

Before diving into the particular techniques for low-rank QPs, we start by exploring the *low-treewidth QPs*, which could be viewed as a structured sparsity condition. It provides valuable insights for the low-rank scenario.

## 2.2 Low-Treewidth Setting: How to Leverage Sparsity

Treewidth is parameter for graphs that captures the sparsity pattern. Given a graph $G = (V, E)$ with $n$ vertices and $m$ edges, a *tree decomposition* of $G$ arranges its vertices into bags, which collectively form a tree structure. For any two bags $X_i, X_j$, if a vertex $v$ is present in both, it must also be included in all bags along the path between $X_i$ and $X_j$. Additionally, each pair of adjacent vertices in the graph must be present together in at least one bag. The treewidth $\tau$ is defined as the maximum size of a bag minus one. Intuitively, a graph $G$ with a small treewidth $\tau$ implies a structure akin to a tree. For a formal definition, see Definition A.1. When relating this combinatorial structure to linear algebra, we could treat the quadratic objective matrix $Q$ as a generalized adjacency matrix, where we put a vertex $v_i$ on $i$-th row of $Q$, and put an edge $\{v_i, v_j\}$ whenever the entry $Q_{i,j}$ is nonzero. The low-treewidth structure of the graph corresponds to a sparsity pattern that allows one to compute a column-sparse Cholesky factorization of $Q$. Since $M = Q + tH$ and $H$ is diagonal, we can decompose $M = LL^\top$ into sparse Cholesky factors[4].

Under any coordinate update to $\overline{x}$, $M$ is updated on only one diagonal entry, enabling efficient updates to $L$. The remaining task is to use this Cholesky decomposition to maintain the central path step.

By expanding the central path equations and substituting $M = LL^\top$, we derive

$$
\begin{aligned}
\delta_x &= tM^{-1/2}(I - P)M^{-1/2}\delta_\mu \\
&= tM^{-1}\delta_\mu - tM^{-1}A^\top(AM^{-1}A^\top)^{-1}AM^{-1}\delta_\mu \\
&= tL^{-\top}L^{-1}\delta_\mu - tL^{-\top}L^{-1}A^\top(AL^{-\top}L^{-1}A^\top)^{-1}AL^{-\top}L^{-1}\delta_\mu, \\
\delta_s &= t\delta_\mu - t^2 HM^{-1/2}(I - P)M^{-1/2}\delta_\mu \\
&= t\delta_\mu - t^2 L^{-\top}L^{-1}\delta_\mu + t^2 L^{-\top}L^{-1}A^\top(AL^{-\top}L^{-1}A^\top)^{-1}AL^{-\top}L^{-1}\delta_\mu.
\end{aligned}
$$

Updates to the diagonal of $M$ do not change $L$'s nonzero pattern, allowing for efficient utilization of the sparse factor and maintenance of $L^{-1}A^\top \in \mathbb{R}^{n \times m}$ and $L^{-1}\delta_\mu \in \mathbb{R}^n$. Terms like $(AL^{-\top}L^{-1}A^\top)^{-1}AL^{-\top}L^{-1}\delta_\mu \in \mathbb{R}^m$ can also be explicitly maintained.

With this approach, we propose the following implicit representation for maintaining $(x, s)$:

$$
x = \widehat{x} + H^{-1/2}\mathcal{W}^\top(h\beta_x - \widetilde{h}\widetilde{\beta}_x + \epsilon_x), \tag{4}
$$

$$
s = \widehat{s} + H^{1/2}c_s\beta_{c_s} - H^{1/2}\mathcal{W}^\top(h\beta_s - \widetilde{h}\widetilde{\beta}_s + \epsilon_s), \tag{5}
$$

where $\widehat{x}, \widehat{s} \in \mathbb{R}^n$, $\mathcal{W} = L^{-1}H^{1/2} \in \mathbb{R}^{n \times n}$, $h = L^{-1}\overline{\delta}_\mu \in \mathbb{R}^n$, $c_s = H^{-1/2}\overline{\delta}_\mu \in \mathbb{R}^n$, $\beta_x, \beta_s, \beta_{c_s} \in \mathbb{R}$, $\widetilde{h} = L^{-1}A^\top \in \mathbb{R}^{n \times m}$, $\widetilde{\beta}_x, \widetilde{\beta}_s \in \mathbb{R}^m$, $\epsilon_x, \epsilon_s \in \mathbb{R}^n$. All quantities except for $\mathcal{W}$ can be explicitly maintained. For linear programming, the implicit representation is as follows:

$$
x = \widehat{x} + H^{-1/2}\beta_x c_x - H^{-1/2}\mathcal{W}^\top(\beta_x h + \epsilon_x)
$$

$$
s = \widehat{s} + H^{1/2}\mathcal{W}^\top(\beta_s h + \epsilon_s),
$$

with $\mathcal{W} = L^{-1}AH^{-1/2}$ maintained implicitly and the other terms explicitly.

The representation in (4) and (5) enables us to maintain the central path step using a combination of "coefficients" $h + \widetilde{h}\widetilde{\beta}_x$ and "basis" $\mathcal{W}^\top$. We need to detect entries of $\overline{x}$ that deviate significantly from $x$ and capture these changes with $\|H^{1/2}(\overline{x} - x)\|_2$. We maintain this vector using $x_0 + \mathcal{W}^\top(h + \widetilde{h}\widetilde{\beta}_x)$. Here, $\mathcal{W}^\top$ acts as a wavelet basis and the vector $h + \widetilde{h}\widetilde{\beta}_x$ as its multiscale coefficients. While computing and maintaining $\mathcal{W}^\top h$ seems challenging, leveraging column-sparsity of $L^{-1}$ is possible through contraction with a vector $v$:

$$
\begin{aligned}
v^\top \mathcal{W}^\top &= (\mathcal{W}v)^\top \\
&= (L^{-1}H^{1/2}v)^\top.
\end{aligned}
$$

By applying the Johnson-Lindenstrauss transform (JL) in place of $v$, we can quickly approximate $\|\mathcal{W}^\top h\|_2$ by maintaining $\Phi\mathcal{W}^\top$ for a JL matrix $\Phi$. Similarly, we handle $\mathcal{W}^\top\widetilde{h}\widetilde{\beta}_x$ by explicitly computing $A^\top\widetilde{\beta}_x$ and using the sparsity of $L^{-1}$ for $\widetilde{h}\widetilde{\beta}_x$.

---

[4]Note that adding a non-negative diagonal matrix to $Q$ does not change its sparsity pattern, hence $M$ also retains the treewidth $\tau$.

We focus on entries significantly deviating from $x_0$, the heavy entries of $\mathcal{W}^\top(h + \widetilde{h}\widetilde{\beta}_x)$. Here, the treewidth-$\tau$ decomposition enables quick computation of an elimination tree based on $L^{-1}$'s sparsity, facilitating efficient estimation of $\|(\mathcal{W}^\top(h + \widetilde{h}\widetilde{\beta}_x))_{\chi(v)}\|_2$ for any subtree $\chi(v)$[5]. With an elimination tree of height $\widetilde{O}(\tau)$, we can employ heavy-light decomposition (Sleator & Tarjan, 1981) for an $O(\log n)$-height tree.

Using these data structures, convergence is established using the robust IPM framework (Ye, 2020; Lee & Vempala, 2021). While the framework is generally applicable to QPs, computing an initial point remains a challenge. We propose a simpler objective $x_0 = \arg\min_{x \in \mathbb{R}^n} \sum_{i=1}^n \phi_i(x_i)$ with $\phi_i$ as the log-barrier function, resembling the initial point reduction in Lee et al. (2019). This initial point enables us to solve an augmented quadratic program that increases dimension by 1.

## 2.3   LOW-RANK SETTING: HOW TO UTILIZE SMALL FACTORIZATION

The low-treewidth structure can be considered a form of sparsity, allowing for a sparse factorization $M = LL^\top$. Another significant structure arises when the matrix $Q$ admits a low-rank factorization. Let $Q = UV^\top$ where $U, V \in \mathbb{R}^{n \times k}$ and $k \ll n$, then $M = Q + tH = UV^\top + tH$. Although $Q$ has a low-rank structure, $M$ may not be low-rank due to the diagonal matrix being dense. However, in the central path equations, we need only handle $M^{-1}$, which can be efficiently maintained using the matrix Woodbury identity:

$$M^{-1} = t^{-1}H^{-1} - t^{-2}H^{-1}U(I + t^{-1}V^\top H^{-1}U)^{-1}V^\top H^{-1},$$

Given that $H$ is diagonal, the complex term $(I + t^{-1}V^\top H^{-1}U)^{-1}$ can be quickly updated under sparse changes to $H^{-1}$ by simply scaling rows of $U$ and $V$. With only a nearly-linear number of updates to $H^{-1}$, the total update time across $\widetilde{O}(\sqrt{n}\log(1/\epsilon))$ iterations is bounded by $\widetilde{O}(nk^{\omega-1} + k^\omega)$. We modify the $(x, s)$ implicit representation as follows:

$$x = \widehat{x} + H^{-1/2}h\beta_x + H^{-1/2}\widehat{h}\widehat{\beta}_x + H^{-1/2}\widetilde{h}\widetilde{\beta}_x, \tag{6}$$

$$s = \widehat{s} + H^{1/2}h\beta_s + H^{1/2}\widehat{h}\widehat{\beta}_s + H^{1/2}\widetilde{h}\widetilde{\beta}_s, \tag{7}$$

where $\overline{x}, \overline{s} \in \mathbb{R}^n$, $h = H^{-1/2}\overline{\delta}_\mu \in \mathbb{R}^n$, $\widehat{h} = H^{-1/2}U \in \mathbb{R}^{n \times k}$, and $\widetilde{h} = H^{-1/2}A^\top \in \mathbb{R}^{n \times m}$, with $\widetilde{\beta}_x, \widetilde{\beta}_s \in \mathbb{R}^m$ and $\beta_x, \beta_s \in \mathbb{R}$. The nontrivial terms to maintain are $\widehat{h}$ and $\widetilde{h}$, but both can be managed straightforwardly: updates to $H^{-1/2}$ correspond to scaling rows of $U$ and $A^\top$, and can be performed in total $\widetilde{O}(nk)$ and $\widetilde{O}(nm)$ time, respectively. The key observation is that we *never explicitly form* $M^{-1/2}$, hence matrix Woodbury identity suffices for fast updates.

The remaining task is to design a data structure for detecting heavy entries. Instead of starting with an elimination tree and re-balancing it through heavy-light decomposition, we construct a balanced tree on $n$ nodes, hierarchically dividing length-$n$ vectors by their indices. Sampling is then performed by traversing down to the tree's leaves. While a heavy-hitter data structure could lead to improvements in poly-logarithmic and sub-logarithmic factors, we primarily focus on polynomial dependencies on various parameters and leave this enhancement for future exploration.

## 2.4   GAUSSIAN KERNEL SVM: ALGORITHM AND HARDNESS

Our specialized QP solvers provide fast implementations for linear SVMs when the data dimension $d$ is much smaller than $n$. However, for kernel SVM, forming the kernel matrix exactly would take $\Theta(n^2)$ time. Fortunately, advancements in kernel matrix algebra (Alman et al., 2020; Backurs et al., 2021; Aggarwal & Alman, 2022; Bakshi et al., 2023) have enabled sub-quadratic algorithms when the data dimension $d$ is small or the kernel matrix has a relatively large minimum entry. Both Alman et al. (2020) and Bakshi et al. (2023) introduce algorithms for spectral sparsification, generating an approximate matrix $\widetilde{K} \in \mathbb{R}^{n \times n}$ such that $(1 - \epsilon) \cdot K \preceq \widetilde{K} \preceq (1 + \epsilon) \cdot K$, with $\widetilde{K}$ having only $O(\epsilon^{-2}n\log n)$ nonzero entries. Alman et al. (2020) achieves this in $O(n^{1+o(1)})$ time for multiplicatively Lipschitz kernels when $d = O(\log n)$, while Bakshi et al. (2023) overcomes limitations for Gaussian kernels by basing their algorithm on KDE and the magnitude of the minimum entry of the kernel matrix, parameterized by $\tau$. Their algorithm for Gaussian kernels runs in time

---

[5]Given any tree node $v$, we use $\chi(v)$ to denote the subtree rooted at $v$.

$\widetilde{O}(nd/\tau^{3.173+o(1)})$. Unfortunately, spectral sparsifiers do not aid our primitives since a sparsifier only reduces the number of nonzero entries, but not the rank of the kernel matrix.

Besides spectral sparsification, Alman et al. (2020); Aggarwal & Alman (2022) also demonstrate that with $d = d = O(\log n)$ and suitable kernels, there exists an $O(n^{1+o(1)})$ time algorithm to multiply the kernel matrix with an arbitrary vector $v \in \mathbb{R}^n$. This operation is crucial in Batch KDE as shown in Aggarwal & Alman (2022). Moreover, Aggarwal & Alman (2022) establishes an almost-quadratic lower bound for this operation when the squared dataset radius $B = \omega(\log n)$, assuming SETH. These results rely on computing a rank-$n^{o(1)}$ factorization for the Gaussian kernel matrix. The function $e^{-x}$ can be approximated by a low-degree polynomial of degree

$$q := \Theta(\max\{\sqrt{B \log(1/\epsilon)}, \frac{\log(1/\epsilon)}{\log(\log(1/\epsilon)/B)}\})$$

for $x \in [0, B]$. Using this polynomial, one can create matrices $U, V$ with rank $\binom{2d+2q}{2q} = n^{o(1)}$ in time $O(n^{1+o(1)})$. Given this factorization, multiplying it with a vector $v$ as $U(V^\top v)$ takes $O(n^{1+o(1)})$ time. Let $\widetilde{K} = UV^\top$ where $\widetilde{K}_{i,j} = f(\|x_i - x_j\|_2^2)$, we have for any $(i,j) \in [n] \times [n]$,

$$|f(\|x_i - x_j\|_2^2) - \exp(-\|x_i - x_j\|_2^2)| \leq \epsilon,$$

and for any row $i \in [n]$,

$$\begin{aligned}
|(\widetilde{K}v)_i - (Kv)_i| &= |\sum_{j=1}^n v_j(f(\|x_i - x_j\|_2^2) - \exp(-\|x_i - x_j\|_2^2))| \\
&\leq (\max_{j \in [n]} |f(\|x_i - x_j\|_2^2) - \exp(-\|x_i - x_j\|_2^2)|)\|v\|_1 \\
&\leq \epsilon\|v\|_1,
\end{aligned}$$

using Hölder's inequality. This provides an $\ell_\infty$-guarantee of the error vector $(\widetilde{K} - K)v$, useful for Batch Gaussian KDE. Transforming this $\ell_\infty$-guarantee into a spectral approximator yields

$$(1 - \epsilon n) \cdot K \preceq \widetilde{K} \preceq (1 + \epsilon n) \cdot K.$$

Setting $\epsilon = 1/n^2$, the low-rank factorization offers an adequate spectral approximation to the exact kernel matrix $K$.

Given $\widetilde{K} = UV^\top$ for $U, V \in \mathbb{R}^{n \times n^{o(1)}}$, we can solve program (3) with $\widetilde{K}$ using our low-rank QP algorithm in time $O(n^{1+o(1)} \log(1/\epsilon))$.[6] This is the first almost-linear time algorithm for Gaussian kernel SVM, even in low-precision settings, as prior works either lack machinery to approximately form the kernel matrix efficiently, or do not possess faster convex optimization solvers for solving a structured quadratic program associated with a kernel SVM.

The requirements $d = O(\log n)$ and $B = o(\frac{\log n}{\log \log n})$ may seem restrictive, but they are necessary, as no sub-quadratic time algorithm exists for Gaussian kernel SVM without bias when $d = \Omega(\log n)$ and $B = \Omega(\log^2 n)$, and with bias when $B = \Omega(\log^6 n)$, assuming SETH. This is based on a reduction from Bichromatic Closet Pair to Gaussian kernel SVM, as established by Backurs et al. (2017). Our assumptions on $d$ and $B$ are therefore justified for seeking almost-linear time algorithms.

We note that in other variants of definitions for Gaussian kernels, one requires an additional parameter called the *kernel width*, and the kernel function is defined as $\exp(-\frac{\|x_i-x_j\|_2^2}{2\sigma^2})$. In commonly used heuristics (Ramdas et al., 2015), $\sigma = O(\sqrt{d})$, hence we could without loss of generality assuming $\sigma = 1$ by requiring the squared radius to be $B/d$.

## 3 CONCLUSION

On the algorithmic front, we introduce the first nearly-linear time algorithms for low-rank convex quadratic programming, leading to nearly-linear time algorithms for linear SVMs. For Gaussian kernel

---

[6]Additional requirement: $B = o(\frac{\log n}{\log \log n})$. See Section G for further discussion.

SVMs, we utilize a low-rank approximation from Aggarwal & Alman (2022) when $d = O(\log n)$ and the squared dataset radius is small, enabling an almost-linear time algorithm. On the hardness aspect, we establish that when $d = \Omega(\log n)$, if the squared dataset radius is sufficiently large ($\Omega(\log^2 n)$ without bias and $\Omega(\log^6 n)$ with bias), then assuming SETH, no sub-quadratic algorithm exists. As our work is theoretical in nature, we do not foresee any potential negative societal impact. Several open problems arise from our work:

**Better dependence on $k$ for low-rank QPs.**    Our low-rank QP solver exhibits a dependence of $k^{(\omega+1)/2}$ on the rank $k$. Given the precomputed factorization, can we improve the exponents on $k$? Ideally, an algorithm with nearly-linear dependence on $k$ would align more closely with input size.

**Better dependence on $m$ for general QPs.**    Focusing on SVMs with a few equality constraints, our QP solvers do not exhibit strong dependence on the number of equality constraints $m$. Without structural assumptions on the constraint matrix $A$, this is expected. However, many QPs, particularly in graph contexts, involve large $m$. Is there a pathway to an algorithm with better dependence on $m$? More broadly, can we achieve a result akin to that of Lee & Sidford (2019), where the number of iterations depends on the square root of the rank of $A$, with minimal per iteration cost?

**Stronger lower bound in terms of $B$ for Gaussian kernel SVMs.**    We establish hardness results for Gaussian kernel SVM when $B = \Omega(\log^2 n)$ without bias and $B = \Omega(\log^6 n)$ with bias. This contrasts with our algorithm, which requires $B$ to have sub-logarithmic dependence on $n$. For Batch Gaussian KDE, Aggarwal & Alman (2022) demonstrated that fast algorithms are feasible for $B = o(\log n)$, with no sub-quadratic time algorithms for $B = \omega(\log n)$ assuming SETH. Can a stronger lower bound be shown for SVM programs with a bias term, reflecting a more natural setting?

## ACKNOWLEDGEMENT

Part of the work was done while Yuzhou Gu was supported by the National Science Foundation under Grant No. DMS-1926686. Lichen Zhang was supported in part by NSF CCF-1955217 and NSF DMS-2022448.

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

## A  PRELIMINARY

### A.1  NOTATIONS

For a positive integer $n$, we use $[n]$ to denote the set $\{1, 2, \cdots, n\}$. For a matrix $A$, we use $A^\top$ to denote its transpose. For a matrix $A$, we define $\|A\|_{p \to q} := \sup_x \|Ax\|_q / \|x\|_p$. When $p = q = 2$, we recover the spectral norm.

We define the entrywise $\ell_p$-norm of a matrix $A$ as $\|A\|_p := (\sum_{i,j} |A_{i,j}|^p)^{1/p}$.

For any function $f : \mathbb{N} \to \mathbb{N}$ and $n \in \mathbb{N}$, we use $\widetilde{O}(f(n))$ to denote $O(f(n) \operatorname{poly} \log f(n))$. We use $\mathbb{1}\{E\}$ to denote the indicator for event $E$, i.e., if $E$ happens, $\mathbb{1}\{E\} = 1$ and otherwise it's 0.

### A.2  TREEWIDTH

Treewidth captures the sparsity and tree-like structures of a graph.

**Definition A.1** (Tree Decomposition and Treewidth). *Let $G = (V, E)$ be a graph, a tree decomposition of $G$ is a tree $T$ with $b$ vertices, and $b$ sets $J_1, \ldots, J_b \subseteq V$ (called bags), satisfying the following properties:*

- *For every edge $(u, v) \in E$, there exists $j \in [b]$ such that $u, v \in J_j$;*

- *For every vertex $v \in V$, $\{j \in [b] : v \in J_j\}$ is a non-empty subtree of $T$.*

*The treewidth of $G$ is defined as the minimum value of $\max\{|J_j| : j \in [b]\} - 1$ over all tree decompositions.*

A near-optimal tree decomposition of a graph can be computed in almost linear time.

**Theorem A.2** (Bernstein et al. (2022)). *Given a graph $G$, there is an $O(m^{1+o(1)})$ time algorithm that produces a tree decomposition of $G$ of maximum bag size $O(\tau \log^3 n)$, where $\tau$ is the actual (unknown) treewidth of $G$.*

Therefore, when $\tau = m^{\Theta(1)}$, we can compute an $\widetilde{O}(\tau)$-size tree decomposition in time $O(m \tau^{o(1)})$, which is negligible in the final running time of Theorem D.1.

### A.3  SPARSE CHOLESKY DECOMPOSITION

In this section we state a few results on sparse Cholesky decomposition. Fast sparse Cholesky decomposition algorithms are based on the concept of elimination tree, introduced in Schreiber (1982).

**Definition A.3** (Elimination tree). *Let $G$ be an undirected graph on $n$ vertices. An elimination tree $\mathcal{T}$ is a rooted tree on $V(G)$ together with an ordering $\pi$ of $V(G)$ such that for any vertex $v$, its parent is the smallest (under $\pi$) element $u$ such that there exists a path $P$ from $v$ to $u$, such that $\pi(w) \leq \pi(v)$ for all $w \in P - u$.*

The following lemma relates the elimination tree and the structure of Cholesky factors.

**Lemma A.4** (Schreiber (1982)). *Let $M$ be a PSD matrix and $\mathcal{T}$ be an elimination tree of the adjacency graph of $M$ (i.e., $(i, j) \in E(G)$ iff $M_{i,j} \neq 0$) together with an elimination ordering $\pi$.*

Let $P$ be the permutation matrix $P_{i,v} = \mathbb{1}\{v = \pi(i)\}$. Then the Cholesky factor $L$ of $PMP^\top$ (i.e., $PMP^\top = LL^\top$) satisfies $L_{i,j} \neq 0$ only if $\pi(i)$ is an ancestor of $\pi(j)$.

The following result is the current best result for computing a sparse Cholesky decomposition.

**Lemma A.5** ((Gu & Song, 2022, Lemma 8.4)). *Let $M \in \mathbb{R}^{n \times n}$ be a PSD matrix whose adjacency graph has treewidth $\tau$. Then we can compute the Cholesky factorization $M = LL^\top$ in $\widetilde{O}(n\tau^{\omega-1})$ time.*

The following result is the current best result for updating a sparse Cholesky decomposition.

**Lemma A.6** (Davis & Hager (1999)). *Let $M \in \mathbb{R}^{n \times n}$ be a PSD matrix whose adjacency graph has treewidth $\tau$. Assume that we are given the Cholseky factorization $M = LL^\top$. Let $w \in \mathbb{R}^n$ be a vector such that $M + ww^\top$ has the same adjacency graph as $M$. Then we can compute $\Delta_L \in \mathbb{R}^{n \times n}$ such that $L + \Delta_L$ is the Cholesky factor of $M + ww^\top$ in $O(\tau^2)$ time.*

Throughout our algorithm, we need to compute matrix-vector multiplications involving Cholesky factors. We use the following results from Gu & Song (2022).

**Lemma A.7** ((Gu & Song, 2022, Lemma 4.7)). *Let $M \in \mathbb{R}^{n \times n}$ be a PSD matrix whose adjacency graph has treewidth $\tau$. Assume that we are given the Cholseky factorization $M = LL^\top$. Then we have the following running time for matrix-vector multiplications.*

(i) *For $v \in \mathbb{R}^n$, computing $Lv$, $L^\top v$, $L^{-1}v$, $L^{-\top}v$ takes $O(n\tau)$ time.*

(ii) *For $v \in \mathbb{R}^n$, computing $Lv$ takes $O(\|v\|_0 \tau)$ time.*

(iii) *For $v \in \mathbb{R}^n$, computing $L^{-1}v$ takes $O(\|L^{-1}v\|_0 \tau)$ time.*

(iv) *For $v \in \mathbb{R}^n$, if $v$ is supported on a path in the elimination tree, then computing $L^{-1}v$ takes $O(\tau^2)$ time.*

(v) *For $v \in \mathbb{R}^n$, computing $\mathcal{W}^\top v$ takes $O(n\tau)$ time, where $\mathcal{W} = L^{-1}H^{1/2}$ with $H \in \mathbb{R}^{n \times n}$ is a non-negative diagonal matrix.*

**Lemma A.8** ((Gu & Song, 2022, Lemma 4.8)). *Let $M \in \mathbb{R}^{n \times n}$ be a PSD matrix whose adjacency graph has treewidth $\tau$. Assume that we are given the Cholseky factorization $M = LL^\top$. Then we have the following running time for matrix-vector multiplications, when we only need result for a subset of coordinates.*

(i) *Let $S$ be a path in the elimination tree whose one endpoint is the root. For $v \in \mathbb{R}^n$, computing $(L^{-\top}v)_S$ takes $O(\tau^2)$ time.*

(ii) *For $v \in \mathbb{R}^n$, for $i \in [n]$, computing $(\mathcal{W}^\top v)_i$ takes $O(\tau^2)$ time, where $\mathcal{W} = L^{-1}H^{1/2}$ with $H \in \mathbb{R}^{n \times n}$ be a non-negative diagonal matrix.*

### A.4 JOHNSON-LINDENTRAUSS LEMMA

We recall the Johnson-Lindenstrauss lemma, a powerful algorithmic primitive that reduces dimension while preserving $\ell_2$ norms.

**Lemma A.9** (Johnson & Lindenstrauss (1984)). *Let $\epsilon \in (0, 1)$ be the precision parameter. Let $\delta \in (0, 1)$ be the failure probability. Let $A \in \mathbb{R}^{m \times n}$ be a real matrix. Let $r = \epsilon^{-2}\log(mn/\delta)$. For $R \in \mathbb{R}^{r \times n}$ whose entries are i.i.d $\mathcal{N}(0, \frac{1}{r})$, the following holds with probability at least $1 - \delta$:*

$$(1 - \epsilon)\|a_i\|_2 \leq \|Ra_i\|_2 \leq (1 + \epsilon)\|a_i\|_2, \ \forall i \in [m],$$

*where for a matrix $A$, $a_i^\top$ denotes the $i$-th row of matrix $A \in \mathbb{R}^{m \times n}$.*

### A.5 HEAVY-LIGHT DECOMPOSITION

Heavy-light decomposition is useful when one wants to re-balance a binary tree with height $O(\log n)$.

**Lemma A.10** (Sleator & Tarjan (1981)). *Given a rooted tree $\mathcal{T}$ with $n$ vertices, we can construct in $O(n)$ time an ordering $\pi$ of the vertices such that (1) every path in $\mathcal{T}$ can be decomposed into $O(\log n)$ contiguous subseqeuences under $\pi$, and (2) every subtree in $\mathcal{T}$ is a single contiguous subsequence under $\pi$.*

# B  SVM FORMULATIONS

In this section, we review a list of formulations of SVM. These formulations have been implemented in the `LIBSVM` library Chang & Lin (2011).

Throughout this section, we use $\phi : \mathbb{R}^d \to \mathbb{R}^s$ to denote the feature mapping, $\mathsf{K}$ to denote the associated kernel function and $K \in \mathbb{R}^{n \times n}$ to denote the kernel matrix. For linear SVM, $\phi$ is just the identity mapping. We will focus on the dual quadratic program formulation as usual. We will also assume for each problem, a dataset $X \in \mathbb{R}^{n \times d}$ is given together with binary labels $y \in \mathbb{R}^n$. Let $Q := (yy^\top) \circ K$.

## B.1  $C$-SUPPORT VECTOR CLASSIFICATION

This formulation is also referred as the *soft-margin SVM*. It can be viewed as imposing a regularization on the primal program to allow mis-classification.

**Definition B.1** ($C$-Support Vector Classification). Given a parameter $C > 0$, the $C$-support vector classification ($C$-SVC) is defined as

$$\max_{\alpha \in \mathbb{R}^n} \quad \mathbf{1}_n^\top \alpha - \frac{1}{2} \alpha^\top Q \alpha$$
$$\text{s.t.} \quad \alpha^\top y = 0,$$
$$0 \le \alpha \le C \cdot \mathbf{1}_n.$$

## B.2  $\nu$-SUPPORT VECTOR CLASSIFICATION

The $C$-SVC (Definition B.1) penalizes large values of $\alpha$ by limiting the magnitude of it. The $\nu$-SVC (Definition B.2) turns $\mathbf{1}_n^\top \alpha$ from an objective into a constraint on $\ell_1$ norm.

**Definition B.2** ($\nu$-Support Vector Classification). Given a parameter $\nu > 0$, the $\nu$-support vector classification ($\nu$-SVC) is defined as

$$\min_{\alpha \in \mathbb{R}^n} \quad \frac{1}{2} \alpha^\top Q \alpha$$
$$\text{s.t.} \quad \alpha^\top y = 0,$$
$$\mathbf{1}_n^\top \alpha = \nu,$$
$$0 \le \alpha \le \frac{1}{n} \cdot \mathbf{1}_n.$$

One can interpret this formulation as to find a vector that lives in the orthogonal complement of $y$ that is non-negative, each entry is at most $\frac{1}{n}$ and its $\ell_1$ norm is $\nu$. Clearly, we must have $\nu \in (0, 1]$. More specifically, let $k_+$ be the number of positive labels and $k_-$ be the number of negative labels. It is shown by Chang & Lin (2001) that the above problem is feasible if and only if

$$\nu \le \frac{2 \min\{k_-, k_+\}}{n}.$$

## B.3  DISTRIBUTION ESTIMATION

SVM is widely-used for predicting binary labels. It can also be used to estimate the support of a high-dimensional distribution. The formulation is similar to $\nu$-SVC, except the PSD matrix $Q$ is *label-less*.

**Definition B.3** (Distribution Estimation). Given a parameter $\nu > 0$, the $\nu$-distribution estimation problem is defined as

$$\min_{\alpha \in \mathbb{R}^n} \quad \frac{1}{2} \alpha^\top K \alpha$$
$$\text{s.t.} \quad 0 \le \alpha \le \frac{1}{n} \cdot \mathbf{1}_n,$$
$$\mathbf{1}_n^\top \alpha = \nu.$$

### B.4 $\epsilon$-SUPPORT VECTOR REGRESSION

In addition to classification, support vector framework can also be adapted for regression.

**Definition B.4** ($\epsilon$-Support Vector Regression). Given parameters $\epsilon, C > 0$, the $\epsilon$-support vector regression ($\epsilon$-SVR) is defined as

$$\min_{\alpha, \alpha^* \in \mathbb{R}^n} \quad \frac{1}{2}(\alpha - \alpha^*)^\top K(\alpha - \alpha^*) + \epsilon \mathbf{1}_n^\top (\alpha + \alpha^*) + y^\top (\alpha - \alpha^*)$$
$$\text{s.t.} \quad \mathbf{1}_n^\top (\alpha - \alpha^*) = 0,$$
$$0 \le \alpha \le C \cdot \mathbf{1}_n,$$
$$0 \le \alpha^* \le C \cdot \mathbf{1}_n.$$

### B.5 $\nu$-SUPPORT VECTOR REGRESSION

One can similar adapt the parameter $\nu$ to control the $\ell_1$ norm of the regression.

**Definition B.5** ($\nu$-Support Vector Regression). Given parameters $\nu, C > 0$, the $\nu$-support vector regression ($\nu$-SVR) is defined as

$$\min_{\alpha, \alpha^* \in \mathbb{R}^n} \quad \frac{1}{2}(\alpha - \alpha^*)^\top K(\alpha - \alpha^*) + y^\top (\alpha - \alpha^*)$$
$$\text{s.t.} \quad \mathbf{1}_n^\top (\alpha - \alpha^*) = 0,$$
$$\mathbf{1}_n^\top (\alpha + \alpha^*) = C\nu,$$
$$0 \le \alpha \le \frac{C}{n} \cdot \mathbf{1}_n,$$
$$0 \le \alpha^* \le \frac{C}{n} \cdot \mathbf{1}_n.$$

### B.6 ONE EQUALITY CONSTRAINT

We classify $C$-SVC (Definition B.1), $\epsilon$-SVR (Definition B.4) and $\nu$-distribution estimation (Definition B.3) into the following generic form:

$$\min_{\alpha \in \mathbb{R}^n} \quad \frac{1}{2}\alpha^\top Q\alpha + p^\top \alpha$$
$$\text{s.t.} \quad \alpha^\top y = \Delta$$
$$0 \le \alpha \le C \cdot \mathbf{1}_n.$$

Note that $C$-SVC (Definition B.1) and distribution estimation (Definition B.3) are readily in this form. For $\epsilon$-SVR (Definition B.4), we need to perform a simple transformation:

Set $\widehat{\alpha} = \begin{bmatrix} \alpha \\ \alpha^* \end{bmatrix} \in \mathbb{R}^{2n}$, then it can be written as

$$\min_{\widehat{\alpha} \in \mathbb{R}^{2n}} \quad \frac{1}{2}\widehat{\alpha}^\top \begin{bmatrix} Q & -Q \\ -Q & Q \end{bmatrix} \widehat{\alpha} + \begin{bmatrix} \epsilon \mathbf{1}_n + y \\ \epsilon \mathbf{1}_n - y \end{bmatrix}^\top \widehat{\alpha}$$
$$\text{s.t.} \quad \begin{bmatrix} \mathbf{1}_n \\ -\mathbf{1}_n \end{bmatrix}^\top \widehat{\alpha} = 0$$
$$0 \le \widehat{\alpha} \le C \cdot \mathbf{1}_{2n}.$$

### B.7 TWO EQUALITY CONSTRAINTS

Both $\nu$-SVC (Definition B.2) and $\nu$-SVR (Definition B.5) require one extra constraint. They can be formulated as follows:

$$\min_{\alpha \in \mathbb{R}^n} \quad \frac{1}{2}\alpha^\top Q\alpha + p^\top \alpha$$

$$\text{s.t. } \mathbf{1}_n^\top \alpha = \Delta_1,$$
$$y^\top \alpha = \Delta_2,$$
$$0 \le \alpha \le C \cdot \mathbf{1}_n.$$

For $\nu$-SVR (Definition B.5), one can leverage a similar transformation as $\epsilon$-SVR (Definition B.4).

**Remark B.6.** All variants of SVM-related quadratic programs can all be solved using our QP solvers for three cases:

- Linear SVM with $n \gg d$, we can solve it in $\widetilde{O}(nd^{(\omega+1)/2} \log(1/\epsilon))$ time;

- Linear SVM with a small treewidth decomposition with width $\tau$ on $XX^\top$, we can solve it in $\widetilde{O}(n\tau^{(\omega+1)/2} \log(1/\epsilon))$ time;

- Gaussian kernel SVM with $d = \Theta(\log n)$ and $B = o(\frac{\log n}{\log \log n})$, we can solve it in $O(n^{1+o(1)} \log(1/\epsilon))$ time.

Even though our solvers have relatively bad dependence on the number of equality constraints, for all these SVM formulations, at most 2 equality constraints are presented and thus can be solved very fast.

## C  ALGORITHMS FOR GENERAL QP

In this section, we discuss algorithms for general (convex) quadratic programming. We show that they can be solved in the current matrix multiplication time via reduction to linear programming with convex constraints Lee et al. (2019).

### C.1  LCQP IN THE CURRENT MATRIX MULTIPLICATION TIME

**Proposition C.1.** *There is an algorithm that solves LCQP (Definition 1.1) up to $\epsilon$ error in $\widetilde{O}((n^\omega + n^{2.5-\alpha/2} + n^{2+1/6}) \log(1/\epsilon))$ time, where $\omega \le 2.373$ is the matrix multiplication constant and $\alpha \ge 0.32$ is the dual matrix multiplication constant.*

*Proof.* Let $Q = PDP^\top$ be an eigen-decomposition of $Q$ where $D$ is diagonal and $P$ is orthogonal. Let $\widetilde{x} := P^{-1}x$. Then it suffices to solve

$$\min \; \frac{1}{2}\widetilde{x}^\top D\widetilde{x} + c^\top P\widetilde{x}$$
$$\text{s.t. } AP\widetilde{x} = b$$
$$P\widetilde{x} \ge 0.$$

By adding $n$ non-negative variables and $n$ constraints $x = P\widetilde{x}$ we can make all constraints equality constraints. There are $n$ non-negative variables and $n$ unconstrained variables. If we want to ensure all variables are non-negative, we need to split every coordinate of $\widetilde{x}$ into two. In this way the coefficient matrix $Q$ will be block diagonal with block size 2.

We perform the above reduction, and assume that we have a program of form (1) where $Q$ is diagonal. Let $q_i := Q_{i,i}$ be the $i$-th element on the diagonal. Then the QP is equivalent to the following program

$$\min \; c^\top x + q^\top t$$
$$\text{s.t. } Ax = b$$
$$t_i \ge \frac{1}{2}x_i^2 \qquad \forall i \in [n]$$
$$x \ge 0$$

Note that the set $\{(x_i, t_i) \in \mathbb{R}^2 : t_i \ge \frac{1}{2}x_i^2\}$ is a convex set. So we can apply Lee et al. (2019) here with $n$ variables, each in the convex set $\{(a, b) \in \mathbb{R}^2 : a \ge 0, b \ge \frac{1}{2}a^2\}$. $\qquad \square$

## C.2 ALGORITHM FOR QCQP

Our algorithm for LCQP in the previous section can be generalized to quadratically constrained quadratic programs (QCQP). QCQP is defined as follows.

**Definition C.2** (QCQP). Let $Q_0, \ldots, Q_m \in \mathbb{R}^{n \times n}$ be PSD matrices. Let $q_0, \ldots, q_m \in \mathbb{R}^n$. Let $r \in \mathbb{R}^m$. Let $A \in \mathbb{R}^{d \times n}$, $b \in \mathbb{R}^d$. The quadratically constrained quadratic programming (QCQP) problem asks the solve the following program.

$$\min_{x \in \mathbb{R}^n} \quad \frac{1}{2} x^\top Q_0 x + q_0^\top x$$
$$\text{s.t.} \quad \frac{1}{2} x^\top Q_i x + q_i^\top x + r_i \leq 0 \qquad \forall i \in [m]$$
$$Ax = b$$
$$x \geq 0$$

**Proposition C.3.** *There is an algorithm that solves QCQP (Definition C.2) up to $\epsilon$ error in $\widetilde{O}(((mn)^\omega + (mn)^{2.5-\alpha/2} + (mn)^{2+1/6}) \log(1/\epsilon))$ time, where $\omega \leq 2.373$ is the matrix multiplication constant and $\alpha \geq 0.32$ is the dual matrix multiplication constant.*

*Proof.* We first rewrite the program as following.

$$\min \quad -r_0$$
$$\text{s.t.} \quad \frac{1}{2} x^\top Q_i x + q_i^\top x + r_i \leq 0 \qquad \forall 0 \leq i \leq m$$
$$Ax = b$$
$$x \geq 0$$

Write $Q_i = P_i D_i P_i^\top$ be an eigen-decomposition of $Q_i$ where $D_i$ is diagonal and $P_i$ is orthogonal. Let $\widetilde{x}_i \in \mathbb{R}^n$ be defined as $\widetilde{x}_i := P_i^{-1} x$. Then we can rewrite the program as

$$\min \quad -r_0$$
$$\text{s.t.} \quad \frac{1}{2} \widetilde{x}_i^\top D_i \widetilde{x}_i + q_i^\top P_i \widetilde{x}_i + r_i \leq 0 \qquad \forall 0 \leq i \leq m$$
$$Ax = b$$
$$\widetilde{x}_i = P_i^{-1} x$$
$$x \geq 0$$

For $0 \leq i \leq m$ and $j \in [n]$, define variable $t_{i,j} \in \mathbb{R}$. Then we can rewrite the program as

$$\min \quad -r_0$$
$$\text{s.t.} \quad \sum_{j \in [n]} D_{i,(j,j)} t_{i,j} + q_i^\top P_i \widetilde{x}_i + r_i \leq 0 \qquad \forall 0 \leq i \leq m$$
$$Ax = b$$
$$\widetilde{x}_i = P_i^{-1} x$$
$$t_{i,j} \geq \widetilde{x}_{i,j}^2$$
$$x \geq 0$$

We can consider $(\widetilde{x}_{i,j}, t_{i,j})_{0 \leq i \leq m, j \in [n]}$ as $(m+1)n$ variables in the convex set $\{(a,b) : b \geq \frac{1}{2} a^2\}$. Then we can apply Lee et al. (2019). $\square$

## D ALGORITHM FOR LOW-TREEWIDTH QP

In this section we present a nearly-linear time algorithm for solving low-treewidth QP with small number of linear constraints. We briefly describe the outline of this section.

- In Section D.1, we present the main statement of Section D.
- In Section D.2, we present the main data structure CENTRALPATHMAINTENANCE.
- In Section D.3, we present several data structures used in CENTRALPATHMAINTENANCE, including EXACTDS (Section D.3.1), APPROXDS (Section D.3.2), BATCHSKETCH (Section D.3.3), VECTORSKETCH (Section D.3.4), BALANCEDSKETCH (Section D.3.5).
- In Section D.4, we prove correctness and running time of CENTRALPATHMAINTENANCE data structure.
- In Section D.5, we prove the main result (Theorem D.1).

## D.1 MAIN STATEMENT

We consider programs of the form (16), i.e.,

$$\min_{x \in \mathbb{R}^n} \quad \frac{1}{2} x^\top Q x + c^\top x$$
$$\text{s.t.} \quad A x = b$$
$$x_i \in \mathcal{K}_i \qquad \forall i \in [n]$$

where $Q \in \mathcal{S}^{n_{\text{tot}}}$, $c \in \mathbb{R}^{n_{\text{tot}}}$, $A \in \mathbb{R}^{m \times n_{\text{tot}}}$, $b \in \mathbb{R}^m$, $\mathcal{K}_i \subset \mathbb{R}^{n_i}$ is a convex set. For simplicity, we assume that $n_i = O(1)$ for all $i \in [n]$.

**Theorem D.1.** *Consider the convex program* (16). *Let* $\phi_i : \mathcal{K}_i \to \mathbb{R}$ *be a* $\nu_i$-*self-concordant barrier for all* $i \in [n]$. *Suppose the program satisfies the following properties:*

- *Inner radius* $r$: *There exists* $z \in \mathbb{R}^{n_{\text{tot}}}$ *such that* $Az = b$ *and* $B(z, r) \in \mathcal{K}$.

- *Outer radius* $R$: $\mathcal{K} \subseteq B(0, R)$ *where* $0 \in \mathbb{R}^{n_{\text{tot}}}$.

- *Lipschitz constant* $L$: $\|Q\|_{2 \to 2} \le L$, $\|c\|_2 \le L$.

- *Treewidth* $\tau$: *Treewidth (Definition A.1) of the adjacency graph of* $Q$ *is at most* $\tau$.

*Let* $(w_i)_{i \in [n]} \in \mathbb{R}^n_{\ge 1}$ *and* $\kappa = \sum_{i \in [n]} w_i \nu_i$. *Given any* $0 < \epsilon \le \frac{1}{2}$, *we can find an approximate solution* $x \in \mathcal{K}$ *satisfying*

$$\frac{1}{2} x^\top Q x + c^\top x \le \min_{Ax=b, x \in \mathcal{K}} \left( \frac{1}{2} x^\top Q x + c^\top x \right) + \epsilon L R (R + 1),$$
$$\|Ax - b\|_1 \le 3\epsilon (R \|A\|_1 + \|b\|_1),$$

*in expected time*

$$\widetilde{O}((\sqrt{\kappa} n^{-1/2} + \log(R/(r\epsilon))) \cdot n (\tau^2 m + \tau m^2)^{1/2} (\tau^{\omega-1} + \tau m + m^{\omega-1})^{1/2}).$$

*When* $\max_{i \in [n]} \nu_i = \widetilde{O}(1)$, $w_i = 1$, $m = \widetilde{O}(\tau^{\omega-2})$, *the running time simplifies to*

$$\widetilde{O}(n \tau^{(\omega+1)/2} m^{1/2} \log(R/(r\epsilon))).$$

## D.2 ALGORITHM STRUCTURE AND CENTRAL PATH MAINTENANCE

Our algorithm is based on the robust Interior Point Method (robust IPM). Details of the robust IPM will be given in Section F. During the algorithm, we maintain a primal-dual solution pair $(x, s) \in \mathbb{R}^{n_{\text{tot}}} \times \mathbb{R}^{n_{\text{tot}}}$ on the robust central path. In addition, we maintain a sparsely-changing approximation $(\overline{x}, \overline{s}) \in \mathbb{R}^{n_{\text{tot}}} \times \mathbb{R}^{n_{\text{tot}}}$ to $(x, s)$. In each iteration, we implicitly perform update

$$x \leftarrow x + \overline{t} B_{w,\overline{x},\overline{t}}^{-1/2} (I - P_{w,\overline{x},\overline{t}}) B_{w,\overline{x},\overline{t}}^{-1/2} \delta_\mu$$
$$s \leftarrow s + \overline{t} \delta_\mu - \overline{t}^2 H_{w,\overline{x}} B_{w,\overline{x},\overline{t}}^{-1/2} (I - P_{w,\overline{x},\overline{t}}) B_{w,\overline{x},\overline{t}}^{-1/2} \delta_\mu$$

where

$$H_{w,\overline{x}} = \nabla^2 \phi_w(\overline{x}) \qquad \qquad \text{(see Eq. (24))}$$

$$B_{w,\overline{x},\overline{t}} = Q + \overline{t}H_{w,\overline{x}} \qquad \text{(see Eq. (25))}$$

$$P_{w,\overline{x},\overline{t}} = B_{w,\overline{x},\overline{t}}^{-1/2} A^\top (AB_{w,\overline{x},\overline{t}}^{-1} A^\top)^{-1} AB_{w,\overline{x},\overline{t}}^{-1/2} \qquad \text{(see Eq. (26))}$$

and explicitly maintain $(\overline{x}, \overline{s})$ such that they remain close to $(x, s)$ in $\ell_\infty$-distance.

This task is handled by the CENTRALPATHMAINTENANCE data structure, which is our main data structure. The robust IPM algorithm (Algorithm 19, 20) directly calls it in every iteration.

The CENTRALPATHMAINTENANCE data structure (Algorithm 1) has two main sub data structures, EXACTDS (Algorithm 2, 3) and APPROXDS (Algorithm 4, 5). EXACTDS is used to maintain $(x, s)$, and APPROXDS is used to maintain $(\overline{x}, \overline{s})$.

---

**Algorithm 1** Our main data structure for low-treewidth QP solver.

1: **data structure** CENTRALPATHMAINTENANCE                            ▷ Theorem D.2
2:   **private : member**
3:     EXACTDS exact                                           ▷ Algorithm 2, 3
4:     APPROXDS approx                                   ▷ Algorithm 4
5:     $\ell \in \mathbb{N}$
6:   **end members**
7:   **procedure** INITIALIZE($x, s \in \mathbb{R}^{n_{\text{tot}}}, t \in \mathbb{R}_+, \overline{\epsilon} \in (0, 1)$)
8:     exact.INITIALIZE($x, s, x, s, t$)                            ▷ Algorithm 2
9:     $\ell \leftarrow 0$
10:     $w \leftarrow \nu_{\max}, N \leftarrow \sqrt{\kappa} \log n \log \frac{n\kappa R}{\overline{\epsilon} r}$
11:     $q \leftarrow n^{1/2}(\tau^2 m + \tau m^2)^{-1/2}(\tau^{\omega-1} + \tau m + m^{\omega-1})^{1/2}$
12:     $\epsilon_{\text{apx},x} \leftarrow \overline{\epsilon}, \zeta_x \leftarrow 2\alpha, \delta_{\text{apx}} \leftarrow \frac{1}{N}$
13:     $\epsilon_{\text{apx},s} \leftarrow \overline{\epsilon} \cdot \overline{t}, \zeta_s \leftarrow 3\alpha\overline{t}$
14:

        approx.INITIALIZE($x, s, h, \widetilde{h}, \epsilon_x, \epsilon_s, H_{w,\overline{x}}^{1/2}\widehat{x}, H_{w,\overline{x}}^{-1/2}\widehat{s}, c_s, \beta_x, \beta_s, \beta_{c_s},$

        $\widetilde{\beta}_x, \widetilde{\beta}_s, q, \&\text{exact}, \epsilon_{\text{apx},x}, \epsilon_{\text{apx},s}, \delta_{\text{apx}})$

15:                ▷ Algorithm 4.Parameters from $x$ to $\widetilde{\beta}_s$ come from exact. $\&$exact is pointer to exact
16: **end procedure**
17: **procedure** MULTIPLYANDMOVE($t \in \mathbb{R}_+$)
18:     $\ell \leftarrow \ell + 1$
19:     **if** $|\overline{t} - t| > \overline{t} \cdot \epsilon_t$ or $\ell > q$ **then**
20:         $x, s \leftarrow \text{exact.OUTPUT}()$                               ▷ Algorithm 2
21:         INITIALIZE($x, s, t, \overline{\epsilon}$)
22:     **end if**
23:     $\beta_x, \beta_s, \beta_{c_s}, \widetilde{\beta}_x, \widetilde{\beta}_s \leftarrow \text{exact.MOVE}()$               ▷ Algorithm 2
24:     $\delta_{\overline{x}}, \delta_{\overline{s}} \leftarrow \text{approx.MOVEANDQUERY}(\beta_x, \beta_s, \beta_{c_s}, \widetilde{\beta}_x, \widetilde{\beta}_s)$     ▷ Algorithm 4
25:     $\delta_h, \delta_{\widetilde{h}}, \delta_{\epsilon_x}, \delta_{\epsilon_s}, \delta_{H_{w,\overline{x}}^{1/2}\widehat{x}}, \delta_{H_{w,\overline{x}}^{-1/2}\widehat{s}}, \delta_{c_s} \leftarrow \text{exact.UPDATE}(\delta_{\overline{x}}, \delta_{\overline{s}})$   ▷ Algorithm 3
26:     approx.UPDATE($\delta_{\overline{x}}, \delta_h, \delta_{\widetilde{h}}, \delta_{\epsilon_x}, \delta_{\epsilon_s}, \delta_{H_{w,\overline{x}}^{1/2}\widehat{x}}, \delta_{H_{w,\overline{x}}^{-1/2}\widehat{s}}, \delta_{c_s}$)    ▷ Algorithm 4
27: **end procedure**
28: **procedure** OUTPUT()
29:     **return** exact.OUTPUT()                                   ▷ Algorithm 2
30: **end procedure**
31: **end data structure**

---

**Theorem D.2.** *Data structure* CENTRALPATHMAINTENANCE *(Algorithm 1) implicitly maintains the central path primal-dual solution pair* $(x, s) \in \mathbb{R}^{n_{\text{tot}}} \times \mathbb{R}^{n_{\text{tot}}}$ *and explicitly maintains its approximation* $(\overline{x}, \overline{s}) \in \mathbb{R}^{n_{\text{tot}}} \times \mathbb{R}^{n_{\text{tot}}}$ *using the following functions:*

- INITIALIZE($x \in \mathbb{R}^{n_{\text{tot}}}, s \in \mathbb{R}^{n_{\text{tot}}}, t_0 \in \mathbb{R}_{>0}, \epsilon \in (0, 1)$)*: Initializes the data structure with initial primal-dual solution pair* $(x, s) \in \mathbb{R}^{n_{\text{tot}}} \times \mathbb{R}^{n_{\text{tot}}}$*, initial central path timestamp* $t_0 \in \mathbb{R}_{>0}$ *in* $\widetilde{O}(n(\tau^{\omega-1} + \tau m + m^{\omega-1}))$ *time.*

- MULTIPLYANDMOVE($t \in \mathbb{R}_{>0}$): *It implicitly maintains*

$$x \leftarrow x + \bar{t} B_{w,\overline{x},\bar{t}}^{-1/2}(I - P_{w,\overline{x},\bar{t}}) B_{w,\overline{x},\bar{t}}^{-1/2} \delta_\mu(\overline{x}, \overline{s}, \bar{t})$$

$$s \leftarrow s + \bar{t}\delta_\mu - \bar{t}^2 H_{w,\overline{x}} B_{w,\overline{x},\bar{t}}^{-1/2}(I - P_{w,\overline{x},\bar{t}}) B_{w,\overline{x},\bar{t}}^{-1/2} \delta_\mu(\overline{x}, \overline{s}, \bar{t})$$

  *where $H_{w,\overline{x}}$, $B_{w,\overline{x},\bar{t}}$, $P_{w,\overline{x},\bar{t}}$ are defined in Eq. (24)(25)(26) respectively, and $\bar{t}$ is some timestamp satisfying $|\bar{t} - t| \le \epsilon_t \cdot \bar{t}$.*

  *It also explicitly maintains $(\overline{x}, \overline{s}) \in \mathbb{R}^{n_{\text{tot}} \times n_{\text{tot}}}$ such that $\|\overline{x}_i - x_i\|_{\overline{x}_i} \le \overline{\epsilon}$ and $\|\overline{s}_i - s_i\|_{\overline{x}_i}^* \le t\overline{\epsilon}w_i$ for all $i \in [n]$ with probability at least $0.9$.*

  *Assuming the function is called at most $N$ times and $t$ decreases from $t_{\max}$ to $t_{\min}$, the total running time is*

$$\widetilde{O}((Nn^{-1/2} + \log(t_{\max}/t_{\min})) \cdot n(\tau^2 m + \tau m^2)^{1/2}(\tau^{\omega-1} + \tau m + m^{\omega-1})^{1/2}).$$

- OUTPUT: *Computes $(x, s) \in \mathbb{R}^{n_{\text{tot}}} \times \mathbb{R}^{n_{\text{tot}}}$ exactly and outputs them in $\widetilde{O}(n\tau m)$ time.*

## D.3    DATA STRUCTURES USED IN CENTRALPATHMAINTENANCE

In this section we present several data structures used in CENTRALPATHMAINTENANCE, including:

- EXACTDS (Section D.3.1): This data structure maintains an implicit representation of the primal-dual solution pair $(x, s)$. This is directly used by CENTRALPATHMAINTENANCE.
- APPROXDS (Section D.3.2): This data structure explicitly maintains an approximation $(\overline{x}, \overline{s})$ of $(x, s)$. This data structure is directly used by CENTRALPATHMAINTENANCE.
- BATCHSKETCH (Section D.3.3): This data structure maintains a sketch of $(x, s)$. This data structure is used by APPROXDS.
- VECTORSKETCH (Section D.3.4): This data structure maintains a sketch of sparsely-changing vectors. This data structure is used by BATCHSKETCH.
- BALANCEDSKETCH (Section D.3.5): This data structure maintains a sketch of vectors of form $\mathcal{W}^\top v$, where $v$ is sparsely-changing. This data structure is used by BATCHSKETCH.

Notation: In this section, for simplicity, we write $B_{\overline{x}}$ for $B_{w,\overline{x},\bar{t}}$, and $L_{\overline{x}}$ for the Cholesky factor of $B_{\overline{x}}$, i.e., $B_{\overline{x}} = L_{\overline{x}} L_{\overline{x}}^\top$.

### D.3.1    EXACTDS

In this section we present the data structure EXACTDS. It maintains an implicit representation of the primal-dual solution pair $(x, s)$ by maintaining several sparsely-changing vectors (see Eq. (8)(9)). This data structure has a similar spirit as EXACTDS in Gu & Song (2022), but we have a different representation from the previous works because we are working with quadratic programming rather than linear programming.

**Theorem D.3.** *Data structure EXACTDS (Algorithm 2, 3) implicitly maintains the primal-dual pair $(x, s) \in \mathbb{R}^{n_{\text{tot}}} \times \mathbb{R}^{n_{\text{tot}}}$, computable via the expression*

$$x = \widehat{x} + H_{w,\overline{x}}^{-1/2} \mathcal{W}^\top (h\beta_x - \widetilde{h}\widetilde{\beta}_x + \epsilon_x), \tag{8}$$

$$s = \widehat{s} + H_{w,\overline{x}}^{1/2} c_s \beta_{c_s} - H_{w,\overline{x}}^{1/2} \mathcal{W}^\top (h\beta_s - \widetilde{h}\widetilde{\beta}_s + \epsilon_s), \tag{9}$$

*where $\widehat{x}, \widehat{s} \in \mathbb{R}^{n_{\text{tot}}}$, $\mathcal{W} = L_{\overline{x}}^{-1} H_{w,\overline{x}}^{1/2} \in \mathbb{R}^{n_{\text{tot}} \times n_{\text{tot}}}$, $h = L_{\overline{x}}^{-1} \delta_\mu \in \mathbb{R}^{n_{\text{tot}}}$, $c_s = H_{w,\overline{x}}^{-1/2} \delta_\mu \in \mathbb{R}^{n_{\text{tot}}}$ $\beta_x, \beta_s, \beta_{c_s} \in \mathbb{R}$, $\widetilde{h} = L_{\overline{x}}^{-1} A^\top \in \mathbb{R}^{n_{\text{tot}} \times m}$, $\widetilde{\beta}_x, \widetilde{\beta}_s \in \mathbb{R}^m$, $\epsilon_x, \epsilon_s \in \mathbb{R}^{n_{\text{tot}}}$.*

*The data structure supports the following functions:*

- INITIALIZE($x, s, \overline{x}, \overline{s} \in \mathbb{R}^{n_{\text{tot}}}, \bar{t} \in \mathbb{R}_{>0}$): *Initializes the data structure in $\widetilde{O}(n\tau^{\omega-1} + n\tau m + nm^{\omega-1})$ time, with initial value of the primal-dual pair $(x, s)$, its initial approximation $(\overline{x}, \overline{s})$, and initial approximate timestamp $\bar{t}$.*

- MOVE(): *Performs robust central path step*

$$x \leftarrow x + \bar{t} B_{\overline{x}}^{-1} \delta_\mu - \bar{t} B_{\overline{x}}^{-1} A^\top (A B_{\overline{x}}^{-1} A^\top)^{-1} A B_{\overline{x}}^{-1} \delta_\mu, \tag{10}$$

$$s \leftarrow s + \bar{t} \delta_\mu - \bar{t}^2 B_{\overline{x}}^{-1} \delta_\mu + \bar{t}^2 B_{\overline{x}}^{-1} A^\top (A B_{\overline{x}}^{-1} A^\top)^{-1} A B_{\overline{x}}^{-1} \delta_\mu \tag{11}$$

  *in $O(m^\omega)$ time by updating its implicit representation.*

- UPDATE($\delta_{\overline{x}}, \delta_{\overline{s}} \in \mathbb{R}^{n_{\text{tot}}}$): *Updates the approximation pair $(\overline{x}, \overline{s})$ to $(\overline{x}^{\text{new}} = \overline{x} + \delta_{\overline{x}} \in \mathbb{R}^{n_{\text{tot}}}, \overline{s}^{\text{new}} = \overline{s} + \delta_{\overline{s}} \in \mathbb{R}^{n_{\text{tot}}})$ in $\widetilde{O}((\tau^2 m + \tau m^2)(\|\delta_{\overline{x}}\|_0 + \|\delta_{\overline{s}}\|_0))$ time, and output the changes in variables $\delta_{H_{w,\overline{x}}^{1/2}\widehat{x}}, \delta_h, \delta_{\beta_x}, \delta_{\widetilde{h}}, \delta_{\widetilde{\beta}_x}, \delta_{\epsilon_x}, \delta_{H_{w,\overline{x}}^{-1/2}\widehat{s}}, \delta_{\beta_s}, \delta_{\widetilde{\beta}_s}, \delta_{\epsilon_s}$.*

  *Furthermore, $h, \epsilon_x, \epsilon_s$ change in $O(\tau(\|\delta_{\overline{x}}\|_0 + \|\delta_{\overline{s}}\|_0))$ coordinates, $\widetilde{h}$ changes in $\widetilde{O}(\tau m (\|\delta_{\overline{x}}\|_0 + \|\delta_{\overline{s}}\|_0))$ coordinates, and $H_{\overline{x}}^{1/2}\widehat{x}, H_{\overline{x}}^{-1/2}\widehat{s}, c_s$ change in $O(\|\delta_{\overline{x}}\|_0 + \|\delta_{\overline{s}}\|_0)$ coordinates.*

- OUTPUT(): *Output $x$ and $s$ in $\widetilde{O}(n\tau m)$ time.*

- QUERYx($i \in [n]$): *Output $x_i$ in $\widetilde{O}(\tau^2 m)$ time. This function is used by APPROXDS.*

- QUERYs($i \in [n]$): *Output $s_i$ in $\widetilde{O}(\tau^2 m)$ time. This function is used by APPROXDS.*

*Proof of Theorem D.3.* By combining Lemma D.4 and D.5. $\qquad\square$

**Lemma D.4.** EXACTDS *correctly maintains an implicit representation of $(x, s)$, i.e., invariant*

$$x = \widehat{x} + H_{w,\overline{x}}^{-1/2} \mathcal{W}^\top (h\beta_x - \widetilde{h}\widetilde{\beta}_x + \epsilon_x),$$

$$s = \widehat{s} + H_{w,\overline{x}}^{1/2} c_s \beta_{c_s} - H_{w,\overline{x}}^{1/2} \mathcal{W}^\top (h\beta_s - \widetilde{h}\widetilde{\beta}_s + \epsilon_s),$$

$$h = L_{\overline{x}}^{-1} \overline{\delta}_\mu, \qquad c_s = H_{w,\overline{x}}^{-1/2} \overline{\delta}_\mu, \qquad \widetilde{h} = L_{\overline{x}}^{-1} A^\top,$$

$$\widetilde{u} = \widetilde{h}^\top \widetilde{h}, \qquad u = \widetilde{h}^\top h,$$

$$\overline{\alpha} = \sum_{i \in [n]} w_i^{-1} \cosh^2 \left( \frac{\lambda}{w_i} \gamma_i(\overline{x}, \overline{s}, \overline{t}) \right),$$

$$\overline{\delta}_\mu = \overline{\alpha}^{1/2} \delta_\mu(\overline{x}, \overline{s}, \overline{t})$$

*always holds after every external call, and return values of the queries are correct.*

*Proof.* INITIALIZE: By checking the definitions we see that all invariants are satisfied after INITIAL-IZE.

MOVE: By comparing the implicit representation (8)(9) and the robust central path step (10)(11), we see that MOVE updates $(x, s)$ correctly.

UPDATE: We would like to prove that UPDATE correctly updates the values of $h, c_s, \widetilde{h}, \widetilde{u}, u, \overline{\alpha}, \overline{\delta}_\mu$, while preserving the values of $(x, s)$.

First note that $H_{w,\overline{x}}, B_{\overline{x}}, L_{\overline{x}}$ are updated correctly. The remaining updates are separated into two steps: UPDATEh and UPDATEh.

**Step** UPDATEh: The first few lines of UPDATEh updates $\overline{\alpha}$ and $\overline{\delta}_\mu$ correctly.

We define $H_{w,\overline{x}}^{\text{new}} := H_{w,\overline{x}} + \Delta_{H_{w,\overline{x}}}, B_{\overline{x}}^{\text{new}} := B_{\overline{x}} + \Delta_{B_{\overline{x}}}, L_{\overline{x}}^{\text{new}} := L_{\overline{x}} + \Delta_{L_{\overline{x}}}, \overline{\delta}_\mu^{\text{new}} := \overline{\delta}_\mu + \delta_{\overline{\delta}_\mu}$, and so on. Immediately after Algorithm 3, Line 26, we have

$$h + \delta_h = L_{\overline{x}}^{-1} \overline{\delta}_\mu + L_{\overline{x}}^{-1} \delta_{\overline{\delta}_\mu} - (L_{\overline{x}} + \Delta_{L_{\overline{x}}})^{-1} \Delta_{L_{\overline{x}}} (L_{\overline{x}}^{-1} \overline{\delta}_\mu + L_{\overline{x}}^{-1} \delta_{\overline{\delta}_\mu})$$

$$= (L_{\overline{x}}^{-1} - (L_{\overline{x}} + \Delta_{L_{\overline{x}}})^{-1} \Delta_{L_{\overline{x}}} L_{\overline{x}}^{-1}) \overline{\delta}_\mu^{\text{new}}$$

$$= L_{\overline{x}}^{\text{new}} \overline{\delta}_\mu^{\text{new}},$$

$$c_s + \delta_{c_s} = H_{w,\overline{x}}^{-1/2} \overline{\delta}_\mu + \Delta_{H_{w,\overline{x}}^{-1/2}} (\overline{\delta}_\mu + \delta_{\overline{\delta}_\mu}) + H_{w,\overline{x}}^{-1/2} \delta_{\overline{\delta}_\mu}$$

---

**Algorithm 2** The EXACTDS data structure used in Algorithm 1.

---

1: **data structure** EXACTDS                                                                                  ▷ Theorem D.3
2: **members**
3:     $\overline{x}, \overline{s} \in \mathbb{R}^{n_{\text{tot}}}, \overline{t} \in \mathbb{R}_+, H_{w,\overline{x}}, B_{\overline{x}}, L_{\overline{x}} \in \mathbb{R}^{n_{\text{tot}} \times n_{\text{tot}}}$
4:     $\widehat{x}, \widehat{s}, h, \epsilon_x, \epsilon_s, c_s \in \mathbb{R}^{n_{\text{tot}}}, \widetilde{h} \in \mathbb{R}^{n_{\text{tot}} \times m}, \beta_x, \beta_s, \beta_{c_s} \in \mathbb{R}, \widetilde{\beta}_x, \widetilde{\beta}_s \in \mathbb{R}^m$
5:     $\widetilde{u} \in \mathbb{R}^{m \times m}, u \in \mathbb{R}^m, \overline{\alpha} \in \mathbb{R}, \overline{\delta}_\mu \in \mathbb{R}^n$
6:     $k \in \mathbb{N}$
7: **end members**
8: **procedure** INITIALIZE($x, s, \overline{x}, \overline{s} \in \mathbb{R}^{n_{\text{tot}}}, \overline{t} \in \mathbb{R}_+$)
9:     $\overline{x} \leftarrow \overline{x}, \overline{x} \leftarrow \overline{s}, \overline{t} \leftarrow \overline{t}$
10:     $\widehat{x} \leftarrow x, \widehat{s} \leftarrow s, \epsilon_x \leftarrow 0, \epsilon_s \leftarrow 0, \beta_x \leftarrow 0, \beta_s \leftarrow 0, \widetilde{\beta}_x \leftarrow 0, \widetilde{\beta}_s \leftarrow 0, \beta_{c_s} \leftarrow 0$
11:     $H_{w,\overline{x}} \leftarrow \nabla^2\phi_w(\overline{x}), B_{\overline{x}} \leftarrow Q + \overline{t}H_{w,\overline{x}}$
12:     Compute lower Cholesky factor $L_{\overline{x}}$ where $L_{\overline{x}}L_{\overline{x}}^\top = B_{\overline{x}}$
13:     INITIALIZE$h(\overline{x}, \overline{s}, H_{w,\overline{x}}, L_{\overline{x}})$
14: **end procedure**
15: **procedure** INITIALIZE$h(\overline{x}, \overline{s} \in \mathbb{R}^{n_{\text{tot}}}, H_{w,\overline{x}}, L_{\overline{x}} \in \mathbb{R}^{n_{\text{tot}} \times n_{\text{tot}}})$
16:     **for** $i \in [n]$ **do**
17:         $(\overline{\delta}_\mu)_i \leftarrow -\frac{\alpha \sinh(\frac{\lambda}{w_i}\gamma_i(\overline{x},\overline{s},\overline{t}))}{\gamma_i(\overline{x},\overline{s},\overline{t})} \cdot \mu_i(\overline{x},\overline{s},\overline{t})$
18:         $\overline{\alpha} \leftarrow \overline{\alpha} + w_i^{-1}\cosh^2(\frac{\lambda}{w_i}\gamma_i(\overline{x},\overline{s},\overline{t}))$
19:     **end for**
20:     $h \leftarrow L_{\overline{x}}^{-1}\overline{\delta}_\mu, \widetilde{h} \leftarrow L_{\overline{x}}^{-1}A^\top, c_s \leftarrow H_{w,\overline{x}}^{-1/2}\overline{\delta}_\mu$
21:     $\widetilde{u} \leftarrow \widetilde{h}^\top\widetilde{h}, u \leftarrow \widetilde{h}^\top h$
22: **end procedure**
23: **procedure** MOVE()
24:     $\beta_x \leftarrow \beta_x + \overline{t} \cdot (\overline{\alpha})^{-1/2}$
25:     $\widetilde{\beta}_x \leftarrow \widetilde{\beta}_x + \overline{t} \cdot (\overline{\alpha})^{-1/2} \cdot \widetilde{u}^{-1}u$
26:     $\beta_{c_s} \leftarrow \beta_s + \overline{t} \cdot (\overline{\alpha})^{-1/2}$
27:     $\beta_s \leftarrow \beta_s + \overline{t}^2 \cdot (\overline{\alpha})^{-1/2}$
28:     $\widetilde{\beta}_s \leftarrow \widetilde{\beta}_s + \overline{t}^2 \cdot (\overline{\alpha})^{-1/2} \cdot \widetilde{u}^{-1}u$
29:     **return** $\beta_x, \beta_s, \beta_{c_s}, \widetilde{\beta}_x, \widetilde{\beta}_s$
30: **end procedure**
31: **procedure** OUTPUT()
32:     **return** $\widehat{x} + H_{w,\overline{x}}^{-1/2}\mathcal{W}^\top(h\beta_x - \widetilde{h}\widetilde{\beta}_x + \epsilon_x), \widehat{s} + H_{w,\overline{x}}^{1/2}c_s\beta_{c_s} - H_{w,\overline{x}}^{1/2}\mathcal{W}^\top(h\beta_s - \widetilde{h}\widetilde{\beta}_s + \epsilon_s)$
33: **end procedure**
34: **procedure** QUERY$x(i \in [n])$
35:     **return** $\widehat{x}_i + H_{w,\overline{x},(i,i)}^{-1/2}(\mathcal{W}^\top(h\beta_x - \widetilde{h}\widetilde{\beta}_x + \epsilon_x))_i$
36: **end procedure**
37: **procedure** QUERY$s(i \in [n])$
38:     **return** $\widehat{s}_i + H_{w,\overline{x},(i,i)}^{1/2}c_{s,i}\beta_{c_s} + H_{w,\overline{x},(i,i)}^{1/2}(\mathcal{W}^\top(h\beta_s - \widetilde{h}\widetilde{\beta}_s + \epsilon_s))_i$
39: **end procedure**
40: **end data structure**

---

$$\begin{aligned}
&= (H_{w,\overline{x}}^{\text{new}})^{-1/2}\overline{\delta}_\mu^{\text{new}}, \\
\widetilde{h} + \delta_{\widetilde{h}} &= L_{\overline{x}}^{-1}A^\top - (L_{\overline{x}} + \Delta_{L_{\overline{x}}})^{-1}\Delta_{L_{\overline{x}}}A^\top \\
&= (L_{\overline{x}}^{-1} - (L_{\overline{x}} + \Delta_{L_{\overline{x}}})^{-1}\Delta_{L_{\overline{x}}}L_{\overline{x}}^{-1})A^\top \\
&= L_{\overline{x}}^{\text{new}}A^\top.
\end{aligned}$$

So $h, c_s, \widetilde{h}$ are updated correctly. Also

$$\begin{aligned}
\widetilde{u} + \delta_{\widetilde{u}} &= \widetilde{h}^\top\widetilde{h} + \delta_{\widetilde{h}}^\top(\widetilde{h} + \delta_{\widetilde{h}}) + \widetilde{h}^\top\delta_{\widetilde{h}} = (\widetilde{h} + \delta_{\widetilde{h}})^\top(\widetilde{h} + \delta_{\widetilde{h}}), \\
u + \delta_u &= \widetilde{h}^\top h + \delta_{\widetilde{h}}^\top(h + \delta_h) + \widetilde{h}^\top\delta_h = (\widetilde{h} + \delta_{\widetilde{h}})^\top(h + \delta_h).
\end{aligned}$$

---

**Algorithm 3** Algorithm 2 continued.

1: **data structure** EXACTDS                                                              ▷ Theorem D.3
2: **procedure** UPDATE($\delta_{\overline{x}}, \delta_{\overline{s}} \in \mathbb{R}^{n_{\text{tot}}}$)
3:     $\Delta_{H_{w,\overline{x}}} \leftarrow \nabla^2 \phi_w(\overline{x} + \delta_{\overline{x}}) - H_{w,\overline{x}}$   ▷ $\Delta_{H_{w,\overline{x}}}$ is non-zero only for diagonal blocks $(i,i)$ for which $\delta_{\overline{x},i} \neq 0$
4:     Compute $\Delta_{L_{\overline{x}}}$ where $(L_{\overline{x}} + \Delta_{L_{\overline{x}}})(L_{\overline{x}} + \Delta_{L_{\overline{x}}})^{\top} = B_{\overline{x}} + \overline{t}\Delta_{H_{w,\overline{x}}}$
5:     UPDATE$h(\delta_{\overline{x}}, \delta_{\overline{s}}, \Delta_{H_{w,\overline{x}}}, \Delta_{L_{\overline{x}}})$
6:     UPDATE$\mathcal{W}(\Delta_{H_{w,\overline{x}}}, \Delta_{L_{\overline{x}}})$
7:     $\overline{x} \leftarrow \overline{x} + \delta_{\overline{x}}, \overline{s} \leftarrow \overline{s} + \delta_{\overline{s}}$
8:     $H_{w,\overline{x}} \leftarrow H_{w,\overline{x}} + \Delta_{H_{w,\overline{x}}}, B_{\overline{x}} \leftarrow B_{\overline{x}} + \overline{t}\Delta_{H_{w,\overline{x}}}, L_{\overline{x}} \leftarrow L_{\overline{x}} + \Delta_{L_{\overline{x}}}$
9:     **return** $\delta_h, \delta_{\widetilde{h}}, \delta_{\epsilon_x}, \delta_{\epsilon_s}, \delta_{H_{w,\overline{x}}^{1/2}\widehat{x}}, \delta_{H_{w,\overline{x}}^{-1/2}\widehat{s}}, \delta_{c_s}$
10: **end procedure**
11: **procedure** UPDATE$h(\delta_{\overline{x}}, \delta_{\overline{s}} \in \mathbb{R}^{n_{\text{tot}}}, \Delta_{H_{w,\overline{x}}}, \Delta_{L_{\overline{x}}} \in \mathbb{R}^{n_{\text{tot}} \times n_{\text{tot}}})$
12:     $S \leftarrow \{i \in [n] \mid \delta_{\overline{x},i} \neq 0 \text{ or } \delta_{\overline{s},i} \neq 0\}$
13:     $\delta_{\overline{\delta}_{\mu}} \leftarrow 0$
14:     **for** $i \in S$ **do**
15:         Let $\gamma_i = \gamma_i(\overline{x}, \overline{s}, \overline{t}), \gamma_i^{\text{new}} = \gamma_i(\overline{x} + \delta_{\overline{x}}, \overline{s} + \delta_{\overline{s}}, \overline{t}), \mu_i^{\text{new}} = \mu_i(\overline{x} + \delta_{\overline{x}}, \overline{s} + \delta_{\overline{s}}, \overline{t})$
16:         $\overline{\alpha} \leftarrow \overline{\alpha} - w_i^{-1}\cosh^2(\frac{\lambda}{w_i}\gamma_i) + w_i^{-1}\cosh^2(\frac{\lambda}{w_i}\gamma_i^{\text{new}})$
17:         $\delta_{\overline{\delta}_{\mu},i} \leftarrow -\alpha\sinh(\frac{\lambda}{w_i}\gamma_i^{\text{new}}) \cdot \frac{1}{\gamma_i^{\text{new}}} \cdot \mu_i^{\text{new}} - \overline{\delta}_{\mu,i}$
18:     **end for**
19:     $\delta_h \leftarrow L_{\overline{x}}^{-1}\delta_{\overline{\delta}_{\mu}} - (L_{\overline{x}} + \Delta_{L_{\overline{x}}})^{-1}\Delta_{L_{\overline{x}}}(h + L_{\overline{x}}^{-1}\delta_{\overline{\delta}_{\mu}})$
20:     $\delta_{c_s} \leftarrow \Delta_{H_{w,\overline{x}}^{-1/2}}(\overline{\delta}_{\mu} + \delta_{\overline{\delta}_{\mu}}) + H_{w,\overline{x}}^{-1/2}\delta_{\overline{\delta}_{\mu}}$
21:     $\delta_{\widetilde{h}} \leftarrow -(L_{\overline{x}} + \Delta_{L_{\overline{x}}})^{-1}\Delta_{L_{\overline{x}}}\widetilde{h}$
22:     $\delta_{\widehat{s}} \leftarrow -\delta_{\overline{\delta}_{\mu}}\beta_{c_s}$
23:     $\delta_{\epsilon_x} \leftarrow -\delta_h\beta_x + \delta_{\widetilde{h}}\widetilde{\beta}_x$
24:     $\delta_{\epsilon_s} \leftarrow -\delta_h\beta_s + \delta_{\widetilde{h}}\widetilde{\beta}_s$
25:     $\delta_{\widetilde{u}} \leftarrow \delta_{\widetilde{h}}^{\top}(\widetilde{h} + \delta_{\widetilde{h}}) + \widetilde{h}^{\top}\delta_{\widetilde{h}}$
26:     $\delta_u \leftarrow \delta_{\widetilde{h}}^{\top}(h + \delta_h) + \widetilde{h}^{\top}\delta_h$
27:     $\overline{\delta}_{\mu} \leftarrow \overline{\delta}_{\mu} + \delta_{\overline{\delta}_{\mu}}, h \leftarrow h + \delta_h, \widetilde{h} \leftarrow \widetilde{h} + \delta_{\widetilde{h}}, \epsilon_x \leftarrow \epsilon_x + \delta_{\epsilon_x}, \epsilon_s \leftarrow \epsilon_s + \delta_{\epsilon_s}, \widetilde{u} \leftarrow \widetilde{u} + \delta_{\widetilde{u}}, u \leftarrow u + \delta_u$
28: **end procedure**
29: **procedure** UPDATE$\mathcal{W}(\Delta_{H_{w,\overline{x}}}, \Delta_{L_{\overline{x}}} \in \mathbb{R}^{n_{\text{tot}}})$
30:     $\delta_{\epsilon_x} \leftarrow \Delta_{L_{\overline{x}}}^{\top}L_{\overline{x}}^{-\top}(h\beta_x - \widetilde{h}\widetilde{\beta}_x + \epsilon_x)$
31:     $\delta_{\epsilon_s} \leftarrow \Delta_{L_{\overline{x}}}^{\top}L_{\overline{x}}^{-\top}(h\beta_s - \widetilde{h}\widetilde{\beta}_s + \epsilon_s)$
32:     $\epsilon_x \leftarrow \epsilon_x + \delta_{\epsilon_x}, \epsilon_s \leftarrow \epsilon_s + \delta_{\epsilon_s}$
33: **end procedure**
34: **end data structure**

---

So $\widetilde{u}$ and $u$ are maintained correctly. Furthermore, immediately after Algorithm 3, Line 26, we have

$$(\widehat{x} + L_{\overline{x}}^{-\top}(h^{\text{new}}\beta_x - \widetilde{h}^{\text{new}}\widetilde{\beta}_x + \epsilon_x^{\text{new}})) - (\widehat{x} + L_{\overline{x}}^{-\top}(h\beta_x - \widetilde{h}\widetilde{\beta}_x + \epsilon_x))$$
$$= L_{\overline{x}}^{-\top}(\delta_h\beta_x - \delta_{\widetilde{h}}\widetilde{\beta}_s + \delta_{\epsilon_x})$$
$$= 0.$$

Therefore, after UPDATE$h$ finishes, we have

$$x = \widehat{x} + L_{\overline{x}}^{-\top}(h\beta_x - \widetilde{h}\widetilde{\beta}_x + \epsilon_x).$$

For $s$, we have

$$(\widehat{s}^{\text{new}} + (H_{w,\overline{x}}^{\text{new}})^{1/2}c_s^{\text{new}}\beta_{c_s} - L_{\overline{x}}^{-\top}(h^{\text{new}}\beta_s - \widetilde{h}^{\text{new}}\widetilde{\beta}_s + \epsilon_s^{\text{new}}))$$
$$- (\widehat{s} + H_{w,\overline{x}}^{1/2}c_s\beta_{c_s} - L_{\overline{x}}^{-\top}(h\beta_s - \widetilde{h}\widetilde{\beta}_s + \epsilon_s))$$

$$= \delta_{\widehat{s}} + \delta_{\overline{\delta}}\beta_{c_s} - L_{\overline{x}}^{-\top}(\delta_h\beta_s - \delta_{\widetilde{h}}\widetilde{\beta}_s + \delta_{\epsilon_s})$$
$$= 0.$$

Therefore, after UPDATE$h$ finishes, we have

$$s = \widehat{s} + (H_{w,\overline{x}}^{\mathrm{new}})^{1/2}c_s\beta_{c_s} - L_{\overline{x}}^{-\top}(h\beta_s - \widetilde{h}\widetilde{\beta}_s + \epsilon_s).$$

So $x$ and $s$ are both updated correctly. This proves the correctness of UPDATE$h$.

**Step** UPDATE$\mathcal{W}$: Define $\epsilon_x^{\mathrm{new}} := \epsilon_x + \delta_{\epsilon_x}$, $\epsilon_s^{\mathrm{new}} := \epsilon_s + \delta_{\epsilon_s}$. Immediately after Algorithm 3, Line 31, we have

$$(\widehat{x} + (L_{\overline{x}}^{\mathrm{new}})^{-\top}(h\beta_x - \widetilde{h}\widetilde{\beta}_x + \epsilon_x^{\mathrm{new}})) - (\widehat{x} + L_{\overline{x}}^{-\top}(h\beta_x - \widetilde{h}\widetilde{\beta}_x + \epsilon_x))$$
$$= ((L_{\overline{x}}^{\mathrm{new}})^{-\top} - L_{\overline{x}}^{-\top})(h\beta_x - \widetilde{h}\widetilde{\beta}_x + \epsilon_x) + (L_{\overline{x}}^{\mathrm{new}})^{-\top}\delta_{\epsilon_x}$$
$$= 0,$$
$$(\widehat{s} + (H_{w,\overline{x}}^{\mathrm{new}})^{1/2}c_s\beta_{c_s} - (L_{\overline{x}}^{\mathrm{new}})^{-\top}(h\beta_s - \widetilde{h}\widetilde{\beta}_s + \epsilon_s^{\mathrm{new}}))$$
$$\quad - (\widehat{s} + (H_{w,\overline{x}}^{\mathrm{new}})^{1/2}c_s\beta_{c_s} - L_{\overline{x}}^{-\top}(h\beta_s - \widetilde{h}\widetilde{\beta}_s + \epsilon_s))$$
$$= (-(L_{\overline{x}}^{\mathrm{new}})^{-\top} + L_{\overline{x}}^{-\top})(h\beta_s - \widetilde{h}\widetilde{\beta}_s + \epsilon_s) - (L_{\overline{x}}^{\mathrm{new}})^{-\top}\delta_{\epsilon_s}$$
$$= 0.$$

Therefore, after UPDATE$\mathcal{W}$ finishes, we have

$$x = \widehat{x} + (L_{\overline{x}}^{\mathrm{new}})^{-\top}(h\beta_x - \widetilde{h}\widetilde{\beta}_x + \epsilon_x),$$
$$s = \widehat{s} + (H_{w,\overline{x}}^{\mathrm{new}})^{1/2}c_s\beta_{c_s} - (L_{\overline{x}}^{\mathrm{new}})^{-\top}(h\beta_s - \widetilde{h}\widetilde{\beta}_s + \epsilon_s).$$

So $x$ and $s$ are both updated correctly. This proves the correctness of UPDATE$\mathcal{W}$. $\qquad\square$

**Lemma D.5.** *We bound the running time of* EXACTDS *as following.*

  *(i)* EXACTDS.INITIALIZE *(Algorithm 2) runs in* $\widetilde{O}(n\tau^{\omega-1} + n\tau m + nm^{\omega-1})$ *time.*

  *(ii)* EXACTDS.MOVE *(Algorithm 2) runs in* $\widetilde{O}(m^\omega)$ *time.*

  *(iii)* EXACTDS.OUTPUT *(Algorithm 2) runs in* $\widetilde{O}(n\tau m)$ *time and correctly outputs* $(x, s)$.

  *(iv)* EXACTDS.QUERY$x$ *and* EXACTDS.QUERY$s$ *(Algorithm 2) runs in* $\widetilde{O}(\tau^2 m)$ *time and returns the correct answer.*

  *(v)* EXACTDS.UPDATE *(Algorithm 2) runs in* $\widetilde{O}((\tau^2 m + \tau m^2)(\|\delta_{\overline{x}}\|_0 + \|\delta_{\overline{s}}\|_0))$ *time. Furthermore,* $h, \epsilon_x, \epsilon_s$ *change in* $O(\tau(\|\delta_{\overline{x}}\|_0 + \|\delta_{\overline{s}}\|_0))$ *coordinates,* $\widetilde{h}$ *changes in* $\widetilde{O}(\tau m(\|\delta_{\overline{x}}\|_0 + \|\delta_{\overline{s}}\|_0))$ *coordinates, and* $H_{\overline{x}}^{1/2}\widehat{x}, H_{\overline{x}}^{-1/2}\widehat{s}, c_s$ *change in* $O(\|\delta_{\overline{x}}\|_0 + \|\delta_{\overline{s}}\|_0)$ *coordinates.*

*Proof.*    (i) Computing $L_{\overline{x}}$ takes $\widetilde{O}(n\tau^{\omega-1})$ time by Lemma A.5. Computing $h$ and $\widetilde{h}$ takes $\widetilde{O}(n\tau m)$ by Lemma A.7(i).[7] Computing $\widetilde{u}$ and $u$ takes $\mathcal{T}_{\mathrm{mat}}(m, n, m) = \widetilde{O}(nm^{\omega-1})$ time. All other operations are cheap.

  (ii) Computing $\widetilde{u}^{-1}$ takes $\widetilde{O}(m^\omega)$ time. All other operations take $O(m^2)$ time.

  (iii) Running time is by Lemma A.7(v). Correctness is by Lemma D.4.

  (iv) Running time is by Lemma A.8(ii). Correctness is by Lemma D.4.

  (v) Computing $\Delta_{L_{\overline{x}}}$ takes $\widetilde{O}(\tau^2\|\delta_{\overline{x}}\|_0)$ time by Lemma A.6. It is easy to see that $\mathrm{nnz}(\Delta_{H_{w,\overline{x}}}) = O(\|\delta_{\overline{x}}\|_0)$ and $\mathrm{nnz}(\Delta_{L_{\overline{x}}}) = \widetilde{O}(\tau^2\|\delta_{\overline{x}}\|_0)$. It remains to analyze UPDATE$h$ and UPDATE$\mathcal{W}$. For simplicity, we write $k = \|\delta_{\overline{x}}\|_0 + \|\delta_{\overline{s}}\|_0$ in this proof only.

---

[7]Here we compute $\widetilde{h}$ by computing $\widetilde{h}_{*,i} = L_{\overline{x}}^{-1}(A_{i,*})^\top$ for $i \in [m]$ independently. Using fast rectangular matrix multiplication is possible to improve this running time and other similar places. We keep the current bounds for simplicity.

- UPDATE$h$: Updating $\overline{\alpha}$ and computing $\delta_{\overline{\delta}_\mu}$ takes $O(k)$ time. Also, $\|\delta_{\overline{\delta}_\mu}\|_0 = O(k)$.

  Computing $\delta_h$ takes $\widetilde{O}(\tau^2 k)$ time by Lemma A.7(i). Also, $\delta_h$ is supported on $O(k)$ paths in the elimination tree, so $\|\delta_h\|_0 = \widetilde{O}(\tau k)$. Similarly we see that computing $\delta_{\widetilde{h}}$ take $\widetilde{O}(\tau^2 m k)$ time and $\mathrm{nnz}(\delta_{\widetilde{h}}) = \widetilde{O}(\tau m k)$.

  Computing $\delta_{c_s}$ and $\delta_{\widehat{s}}$ takes $O(\tau^2 k)$ time and $\|\delta_{c_s}\|_0, \|\delta_{\widehat{s}}\|_0 = O(k)$.

  Computing $\delta_{\epsilon_x}$ and $\delta_{\epsilon_s}$ takes $O(\tau m k)$ time after computing $\delta_h$ and $\delta_{\widetilde{h}}$. Furthermore, $\|\delta_{\epsilon_x}\|_0, \|\delta_{\epsilon_s}\|_0 = O(\tau k)$.

  Computing $\delta_{\widetilde{u}}$ takes $\mathcal{T}_{\mathrm{mat}}(m, \tau k, m) = \widetilde{O}(\tau m^2 k)$ time. Computing $\delta_u$ takes $\widetilde{O}(\tau m k)$ time.

- UPDATE$\mathcal{W}$: To compute $\delta_{\epsilon_x}$ and $\delta_{\epsilon_s}$, we first compute $L_{\overline{x}}^{-\top}(h\beta_x - \widetilde{h}\widetilde{\beta}_x + \epsilon_x)$ and $L_{\overline{x}}^{-\top}(h\beta_s - \widetilde{h}\widetilde{\beta}_s + \epsilon_s)$, where $S \subseteq [n_{\mathrm{tot}}]$ is the row support of $\Delta_{L_{\overline{x}}}$, which can be decomposed into at most $O(\|\delta_{\overline{x}}\|_0)$ paths. This takes $\widetilde{O}(\tau^2 m \|\delta_{\overline{x}}\|_0)$ time by Lemma A.8(i) (the extra $m$ factor is due to $\widetilde{h}$).

Combining everything we finish the proof of running time of EXACTDS.UPDATE. $\qquad \square$

### D.3.2 APPROXDS

In this section we present the data structure APPROXDS. Given BATCHSKETCH, a data structure maintaining a sketch of the primal-dual pair $(x, s) \in \mathbb{R}^{n_{\mathrm{tot}}} \times \mathbb{R}^{n_{\mathrm{tot}}}$, APPROXDS maintains a sparsely-changing $\ell_\infty$-approximation of $(x, s)$. This data structure is a slight variation of APPROXDS in Gu & Song (2022).

---

**Algorithm 4** The APPROXDS data structure used in Algorithm 1.

1: **data structure** APPROXDS ▷ Theorem D.6
2: **private : members**
3:     $\epsilon_{\mathrm{apx},x}, \epsilon_{\mathrm{apx},s} \in \mathbb{R}$
4:     $\ell \in \mathbb{N}$
5:     BATCHSKETCH bs ▷ This maintains a sketch of $H_{w,\overline{x}}^{1/2} x$ and $H_{w,\overline{x}}^{-1/2} s$. See Algorithm 6, 7, 8.
6:     EXACTDS* exact ▷ This is a pointer to the EXACTDS (Algorithm 2, 3) we maintain in parallel to APPROXDS.
7:     $\widetilde{x}, \widetilde{s} \in \mathbb{R}^{n_{\mathrm{tot}}}$ ▷ $(\widetilde{x}, \widetilde{s})$ is a sparsely-changing approximation of $(x, s)$. They have the same value as $(\overline{x}, \overline{s})$, but for these local variables we use $(\widetilde{x}, \widetilde{s})$ to avoid confusion.
8: **end members**
9: **procedure** INITIALIZE($x, s \in \mathbb{R}^{n_{\mathrm{tot}}}, h \in \mathbb{R}^{n_{\mathrm{tot}}}, \widetilde{h} \in \mathbb{R}^{n_{\mathrm{tot}} \times m}, \epsilon_x, \epsilon_s, H_{w,\overline{x}}^{1/2}\widehat{x}, H_{w,\overline{x}}^{-1/2}\widehat{s}, c_s \in \mathbb{R}^{n_{\mathrm{tot}}}, \beta_x, \beta_s, \beta_{c_s} \in \mathbb{R}, \widetilde{\beta}_x, \widetilde{\beta}_s \in \mathbb{R}^m, q \in \mathbb{N},$ EXACTDS* exact, $\epsilon_{\mathrm{apx},x}, \epsilon_{\mathrm{apx},s}, \delta_{\mathrm{apx}} \in \mathbb{R}$)
10:     $\ell \leftarrow 0, q \leftarrow q$
11:     $\epsilon_{\mathrm{apx},x} \leftarrow \epsilon_{\mathrm{apx},x}, \epsilon_{\mathrm{apx},s} \leftarrow \epsilon_{\mathrm{apx},s}$
12:     bs.INITIALIZE($x, h, \widetilde{h}, \epsilon_x, \epsilon_s, H_{w,\overline{x}}^{1/2}\widehat{x}, H_{w,\overline{x}}^{-1/2}\widehat{s}, c_s, \beta_x, \beta_s, \beta_{c_s}, \widetilde{\beta}_x, \widetilde{\beta}_s, \delta_{\mathrm{apx}}/q$) ▷ Algorithm 6
13:     $\widetilde{x} \leftarrow x, \widetilde{s} \leftarrow s$
14:     exact $\leftarrow$ exact
15: **end procedure**
16: **procedure** UPDATE($\delta_{\overline{x}} \in \mathbb{R}^{n_{\mathrm{tot}}}, \delta_h \in \mathbb{R}^{n_{\mathrm{tot}}}, \delta_{\widetilde{h}} \in \mathbb{R}^{n_{\mathrm{tot}} \times m}, \delta_{\epsilon_x}, \delta_{\epsilon_s}, \delta_{H_{w,\overline{x}}^{1/2}\widehat{x}}, \delta_{H_{w,\overline{x}}^{-1/2}\widehat{s}}, \delta_{c_s} \in \mathbb{R}^{n_{\mathrm{tot}}}$)
17:     bs.UPDATE($\delta_{\overline{x}}, \delta_h, \delta_{\widetilde{h}}, \delta_{\epsilon_x}, \delta_{\epsilon_s}, \delta_{H_{w,\overline{x}}^{1/2}\widehat{x}}, \delta_{H_{w,\overline{x}}^{-1/2}\widehat{s}}, \delta_{c_s}$) ▷ Algorithm 7
18:     $\ell \leftarrow \ell + 1$
19: **end procedure**
20: **procedure** MOVEANDQUERY($\beta_x, \beta_s, \beta_{c_s} \in \mathbb{R}, \widetilde{\beta}_x, \widetilde{\beta}_s \in \mathbb{R}^m$)
21:     bs.MOVE($\beta_x, \beta_s, \beta_{c_s}, \widetilde{\beta}_x, \widetilde{\beta}_s$) ▷ Algorithm 7. Do not update $\ell$ yet
22:     $\delta_{\widetilde{x}} \leftarrow$ QUERY$x(\epsilon_{\mathrm{apx},x}/(2\log q + 1))$ ▷ Algorithm 5
23:     $\delta_{\widetilde{s}} \leftarrow$ QUERY$s(\epsilon_{\mathrm{apx},s}/(2\log q + 1))$ ▷ Algorithm 5
24:     $\widetilde{x} \leftarrow \widetilde{x} + \delta_{\widetilde{x}}, \widetilde{s} \leftarrow \widetilde{s} + \delta_{\widetilde{s}}$
25:     **return** $(\delta_{\widetilde{x}}, \delta_{\widetilde{s}})$
26: **end procedure**
27: **end data structure**

---

**Theorem D.6.** *Given parameters* $\epsilon_{\mathrm{apx},x}, \epsilon_{\mathrm{apx},s} \in (0, 1), \delta_{\mathrm{apx}} \in (0, 1), \zeta_x, \zeta_s \in \mathbb{R}$ *such that*

$$\|H_{w,\overline{x}^{(\ell)}}^{1/2} x^{(\ell)} - H_{w,\overline{x}^{(\ell)}}^{1/2} x^{(\ell+1)}\|_2 \le \zeta_x, \quad \|H_{w,\overline{x}^{(\ell)}}^{-1/2} s^{(\ell)} - H_{w,\overline{x}^{(\ell)}}^{-1/2} s^{(\ell+1)}\|_2 \le \zeta_s$$

---

**Algorithm 5** APPROXDS Algorithm 4 continued.

---

1: **data structure** APPROXDS                                                   ▷ Theorem D.6
2:    **private:**
3:    **procedure** QUERY$x(\epsilon \in \mathbb{R})$
4:       $\mathcal{I} \leftarrow 0$
5:       **for** $j = 0 \rightarrow \lfloor \log_2 \ell \rfloor$ **do**
6:           **if** $\ell \bmod 2^j = 0$ **then**
7:               $\mathcal{I} \leftarrow \mathcal{I} \cup \mathsf{bs.QUERY}x(\ell - 2^j + 1, \epsilon)$                             ▷ Algorithm 8
8:           **end if**
9:       **end for**
10:       $\delta_{\widetilde{x}} \leftarrow 0$
11:       **for all** $i \in \mathcal{I}$ **do**
12:           $x_i \leftarrow \mathsf{exact.QUERY}x(i)$                                           ▷ Algorithm 2
13:           **if** $\|\widetilde{x}_i - x_i\|_{\widetilde{x}_i} > \epsilon$ **then**
14:               $\delta_{\widetilde{x},i} \leftarrow x_i - \widetilde{x}_i$
15:           **end if**
16:       **end for**
17:       **return** $\delta_{\widetilde{x}}$
18: **end procedure**
19: **procedure** QUERY$s(\epsilon \in \mathbb{R})$
20:       Same as QUERY$x$, except for replacing $x, \widetilde{x}, \cdots$ with $s, \widetilde{s}, \cdots$, and replacing "$\|\widetilde{x}_i - x_i\|_{\widetilde{x}_i}$" in Line 13 with "$\|\widetilde{s}_i - s_i\|_{\widetilde{x}_i}^*$".
21: **end procedure**
22: **end data structure**

---

*for all $\ell \in \{0, \ldots, q-1\}$, data structure* APPROXDS *(Algorithm 4 and Algorithm 5) supports the following operations:*

- INITIALIZE($x, s \in \mathbb{R}^{n_{\mathrm{tot}}}, h \in \mathbb{R}^{n_{\mathrm{tot}}}, \widetilde{h} \in \mathbb{R}^{n_{\mathrm{tot}} \times m}, \epsilon_x, \epsilon_s, H_{w,\overline{x}}^{1/2}\widehat{x}, H_{w,\overline{x}}^{-1/2}\widehat{s}, c_s \in \mathbb{R}^{n_{\mathrm{tot}}}, \beta_x, \beta_s, \beta_{c_s} \in \mathbb{R}, \widetilde{\beta}_x, \widetilde{\beta}_s \in \mathbb{R}^m, q \in \mathbb{N}, \mathrm{EXACTDS}^*$ exact, $\epsilon_{\mathrm{apx},x}, \epsilon_{\mathrm{apx},s}, \delta_{\mathrm{apx}} \in \mathbb{R}$): *Initialize the data structure in* $\widetilde{O}(n\tau^{\omega-1} + n\tau m)$ *time.*

- MOVEANDQUERY($\beta_x, \beta_s, \beta_{c_s} \in \mathbb{R}, \widetilde{\beta}_x, \widetilde{\beta}_s \in \mathbb{R}^m$): *Update values of* $\beta_x, \beta_s, \beta_{c_s}, \widetilde{\beta}_x, \widetilde{\beta}_s$ *by calling* BATCHSKETCH.MOVE. *This effectively moves* $(x^{(\ell)}, s^{(\ell)})$ *to* $(x^{(\ell+1)}, s^{(\ell+1)})$ *while keeping* $\overline{x}^{(\ell)}$ *unchanged.*

  *Then return two sets* $L_x^{(\ell)}, L_s^{(\ell)} \subset [n]$ *where*
  $$L_x^{(\ell)} \supseteq \{i \in [n] : \|H_{w,\overline{x}^{(\ell)}}^{1/2} x_i^{(\ell)} - H_{w,\overline{x}^{(\ell)}}^{1/2} x_i^{(\ell+1)}\|_2 \geq \epsilon_{\mathrm{apx},x}\},$$
  $$L_s^{(\ell)} \supseteq \{i \in [n] : \|H_{w,\overline{x}^{(\ell)}}^{-1/2} s_i^{(\ell)} - H_{w,\overline{x}^{(\ell)}}^{-1/2} s_i^{(\ell+1)}\|_2 \geq \epsilon_{\mathrm{apx},s}\},$$
  *satisfying*
  $$\sum_{0 \leq \ell \leq q-1} |L_x^{(\ell)}| = \widetilde{O}(\epsilon_{\mathrm{apx},x}^{-2} \zeta_x^2 q^2),$$
  $$\sum_{0 \leq \ell \leq q-1} |L_s^{(\ell)}| = \widetilde{O}(\epsilon_{\mathrm{apx},s}^{-2} \zeta_s^2 q^2).$$

  *For every query, with probability at least* $1 - \delta_{\mathrm{apx}}/q$, *the return values are correct.*

  *Furthermore, total time cost over all queries is at most*
  $$\widetilde{O}\left((\epsilon_{\mathrm{apx},x}^{-2} \zeta_x^2 + \epsilon_{\mathrm{apx},s}^{-2} \zeta_s^2) q^2 \tau^2 m\right).$$

- UPDATE($\delta_{\overline{x}} \in \mathbb{R}^{n_{\mathrm{tot}}}, \delta_h \in \mathbb{R}^{n_{\mathrm{tot}}}, \delta_{\widetilde{h}} \in \mathbb{R}^{n_{\mathrm{tot}} \times m}, \delta_{\epsilon_x}, \delta_{\epsilon_s}, \delta_{H_{w,\overline{x}}^{1/2}\widehat{x}}, \delta_{H_{w,\overline{x}}^{-1/2}\widehat{s}}, \delta_{c_s} \in \mathbb{R}^{n_{\mathrm{tot}}}$):

  *Update sketches of* $H_{w,\overline{x}^{(\ell)}}^{1/2} x^{(\ell+1)}$ *and* $H_{w,\overline{x}^{(\ell)}}^{-1/2} s^{(\ell+1)}$ *by calling* BATCHSKETCH.UPDATE.

*This effectively moves $\overline{x}^{(\ell)}$ to $\overline{x}^{(\ell+1)}$ while keeping $(x^{(\ell+1)}, s^{(\ell+1)})$ unchanged. Then advance timestamp $\ell$.*

*Each update costs*

$$\widetilde{O}(\tau^2(\|\delta_{\overline{x}}\|_0 + \|\delta_h\|_0 + \|\delta_{\widetilde{h}}\|_0 + \|\delta_{\epsilon_x}\|_0 + \|\delta_{\epsilon_s}\|_0) + \|\delta_{H_{w,\overline{x}}^{1/2}\widehat{x}}\|_0 + \|\delta_{H_{w,\overline{x}}^{-1/2}\widehat{s}}\|_0 + \|\delta_{c_s}\|_0)$$

*time.*

*Proof.* The proof is essentially the same as proof of (Gu & Song, 2022, Theorem 4.18). For the running time claims, we plug in Theorem D.8 when necessary. □

### D.3.3 BATCHSKETCH

In this section we present the data structure BATCHSKETCH. It maintains a sketch of $H_{\overline{x}}^{1/2} x$ and $H_{\overline{x}}^{-1/2} s$. It is a variation of BATCHSKETCH in Gu & Song (2022).

We recall the following definition from Gu & Song (2022).

**Definition D.7** (Partition tree). A partition tree $(\mathcal{S}, \chi)$ of $\mathbb{R}^n$ is a constant degree rooted tree $\mathcal{S} = (V, E)$ and a labeling of the vertices $\chi : V \to 2^{[n]}$, such that

- $\chi(\text{root}) = [n]$;

- if $v$ is a leaf of $\mathcal{S}$, then $|\chi(v)| = 1$;

- for any non-leaf node $v \in V$, the set $\{\chi(c) : c \text{ is a child of } v\}$ is a partition of $\chi(v)$.

---

**Algorithm 6** The BATCHSKETCH data structure used by Algorithm 4 and 5.

1: **data structure** BATCHSKETCH                                      ▷ Theorem D.8
2: **members**
3:    $\Phi \in \mathbb{R}^{r \times n_{\text{tot}}}$                      ▷ All sketches need to share the same sketching matrix
4:    $\mathcal{S}, \chi$ partition tree
5:    $\ell \in \mathbb{N}$                                              ▷ Current timestamp
6:    BALANCEDSKETCH sketch$\mathcal{W}^\top h$, sketch$\mathcal{W}^\top \widetilde{h}$, sketch$\mathcal{W}^\top \epsilon_x$, sketch$\mathcal{W}^\top \epsilon_s$ ▷ Algorithm 10
7:    VECTORSKETCH sketch$H_{w,\overline{x}}^{1/2}\widehat{x}$, sketch$H_{w,\overline{x}}^{-1/2}\widehat{s}$, sketch$c_s$    ▷ Algorithm 9
8:    $\beta_x, \beta_s, \beta_{c_s} \in \mathbb{R}, \widetilde{\beta}_x, \widetilde{\beta}_s \in \mathbb{R}^m$
9:    $(\text{history}[t])_{t \geq 0}$                                   ▷ Snapshot of data at timestamp $t$. See Remark D.9.
10: **end members**
11: **procedure** INITIALIZE($\overline{x} \in \mathbb{R}^{n_{\text{tot}}}, h \in \mathbb{R}^{n_{\text{tot}}}, \widetilde{h} \in \mathbb{R}^{n_{\text{tot}} \times m}, \epsilon_x, \epsilon_s, H_{w,\overline{x}}^{1/2}\widehat{x}, H_{w,\overline{x}}^{-1/2}\widehat{s}, c_s \in$
    $\mathbb{R}^{n_{\text{tot}}}, \beta_x, \beta_s, \beta_{c_s} \in \mathbb{R}, \widetilde{\beta}_x, \widetilde{\beta}_s \in \mathbb{R}^m, \delta_{\text{apx}} \in \mathbb{R}$)
12:    Construct partition tree $(\mathcal{S}, \chi)$ as in Definition D.11
13:    $r \leftarrow \Theta(\log^3(n_{\text{tot}}) \log(1/\delta_{\text{apx}}))$
14:    Initialize $\Phi \in \mathbb{R}^{r \times n_{\text{tot}}}$ with iid $\mathcal{N}(0, \frac{1}{r})$
15:    $\beta_x \leftarrow \beta_x, \beta_s \leftarrow \beta_s, \beta_{c_s} \leftarrow \beta_{c_s}, \widetilde{\beta}_x \leftarrow \widetilde{\beta}_x, \widetilde{\beta}_s \leftarrow \widetilde{\beta}_s$
16:    sketch$\mathcal{W}^\top h$.INITIALIZE($\mathcal{S}, \chi, \Phi, \overline{x}, h$)             ▷ Algorithm 10
17:    sketch$\mathcal{W}^\top \widetilde{h}$.INITIALIZE($\mathcal{S}, \chi, \Phi, \overline{x}, \widetilde{h}$)             ▷ Algorithm 10
18:    sketch$\mathcal{W}^\top \epsilon_x$.INITIALIZE($\mathcal{S}, \chi, \Phi, \overline{x}, \epsilon_x$)             ▷ Algorithm 10
19:    sketch$\mathcal{W}^\top \epsilon_s$.INITIALIZE($\mathcal{S}, \chi, \Phi, \overline{x}, \epsilon_s$)             ▷ Algorithm 10
20:    sketch$H_{w,\overline{x}}^{1/2}\widehat{x}$.INITIALIZE($\mathcal{S}, \chi, \Phi, H_{w,\overline{x}}^{1/2}\widehat{x}$)             ▷ Algorithm 9
21:    sketch$H_{w,\overline{x}}^{-1/2}\widehat{s}$.INITIALIZE($\mathcal{S}, \chi, \Phi, H_{w,\overline{x}}^{-1/2}\widehat{s}$)             ▷ Algorithm 9
22:    sketch$c_s$.INITIALIZE($\mathcal{S}, \chi, \Phi, c_s$)             ▷ Algorithm 9
23:    $\ell \leftarrow 0$. Make snapshot history$[\ell]$             ▷ Remark D.9
24: **end procedure**
25: **end data structure**

---

**Theorem D.8.** *Data structure BATCHSKETCH (Algorithm 6, 8) supports the following operations:*

---

**Algorithm 7** BATCHSKETCH Algorithm 6 continued.

---

1: **data structure** BATCHSKETCH ▷ Theorem D.8
2: **procedure** MOVE($\beta_x, \beta_s, \beta_{c_s} \in \mathbb{R}, \widetilde{\beta}_x, \widetilde{\beta}_s \in \mathbb{R}^m$)
3:     $\beta_x \leftarrow \beta_x, \beta_s \leftarrow \beta_s, \beta_{c_s} \leftarrow \beta_{c_s}, \widetilde{\beta}_x \leftarrow \widetilde{\beta}_x, \widetilde{\beta}_s \leftarrow \widetilde{\beta}_s$ ▷ Do not update $\ell$ yet
4: **end procedure**
5: **procedure** UPDATE($\delta_{\overline{x}} \in \mathbb{R}^{n_{\text{tot}}}, \delta_h \in \mathbb{R}^{n_{\text{tot}}}, \delta_{\widetilde{h}} \in \mathbb{R}^{n_{\text{tot}} \times m}, \delta_{\epsilon_x}, \delta_{\epsilon_s}, \delta_{H_{w,\overline{x}}^{1/2}\widehat{x}}, \delta_{H_{w,\overline{x}}^{-1/2}\widehat{s}}, \delta_{c_s} \in \mathbb{R}^{n_{\text{tot}}}$)
6:     sketch$\mathcal{W}^\top h$.UPDATE($\delta_{\overline{x}}, \delta_h$) ▷ Algorithm 11
7:     sketch$\mathcal{W}^\top \widetilde{h}$.UPDATE($\delta_{\overline{x}}, \delta_{\widetilde{h}}$) ▷ Algorithm 11
8:     sketch$\mathcal{W}^\top \epsilon_x$.UPDATE($\delta_{\overline{x}}, \delta_{\epsilon_x}$) ▷ Algorithm 11
9:     sketch$\mathcal{W}^\top \epsilon_s$.UPDATE($\delta_{\overline{x}}, \delta_{\epsilon_s}$) ▷ Algorithm 11
10:     sketch$H_{w,\overline{x}}^{1/2}\widehat{x}$.UPDATE($\delta_{H_{w,\overline{x}}^{1/2}\widehat{x}}$) ▷ Algorithm 9
11:     sketch$H_{w,\overline{x}}^{-1/2}\widehat{s}$.UPDATE($\delta_{H_{w,\overline{x}}^{-1/2}\widehat{s}}$) ▷ Algorithm 9
12:     sketch$c_s$.UPDATE($\delta_{c_s}$) ▷ Algorithm 9
13:     $\ell \leftarrow \ell + 1$
14:     Make snapshot history$[\ell]$ ▷ Remark D.9
15: **end procedure**
16: **end data structure**

---

- INITIALIZE($\overline{x} \in \mathbb{R}^{n_{\text{tot}}}, h \in \mathbb{R}^{n_{\text{tot}}}, \widetilde{h} \in \mathbb{R}^{n_{\text{tot}} \times m}, \epsilon_x, \epsilon_s, H_{w,\overline{x}}^{1/2}\widehat{x}, H_{w,\overline{x}}^{-1/2}\widehat{s}, c_s \in \mathbb{R}^{n_{\text{tot}}}, \beta_x, \beta_s, \beta_{c_s} \in \mathbb{R}, \widetilde{\beta}_x, \widetilde{\beta}_s \in \mathbb{R}^m, \delta_{\text{apx}} \in \mathbb{R}$): *Initialize the data structure in* $\widetilde{O}(n\tau^{\omega-1} + n\tau m)$ *time.*

- MOVE($\beta_x, \beta_s, \beta_{c_s} \in \mathbb{R}, \widetilde{\beta}_x, \widetilde{\beta}_s \in \mathbb{R}^m$): *Update values of* $\beta_x, \beta_s, \beta_{c_s}, \widetilde{\beta}_x, \widetilde{\beta}_s$ *in* $O(m)$ *time.* *This effectively moves* $(x^{(\ell)}, s^{(\ell)})$ *to* $(x^{(\ell+1)}, s^{(\ell+1)})$ *while keeping* $\overline{x}^{(\ell)}$ *unchanged.*

- UPDATE($\delta_{\overline{x}} \in \mathbb{R}^{n_{\text{tot}}}, \delta_h \in \mathbb{R}^{n_{\text{tot}}}, \delta_{\widetilde{h}} \in \mathbb{R}^{n_{\text{tot}} \times m}, \delta_{\epsilon_x}, \delta_{\epsilon_s}, \delta_{H_{w,\overline{x}}^{1/2}\widehat{x}}, \delta_{H_{w,\overline{x}}^{-1/2}\widehat{s}}, \delta_{c_s} \in \mathbb{R}^{n_{\text{tot}}}$): *Update sketches of* $H_{w,\overline{x}^{(\ell)}}^{1/2} x^{(\ell+1)}$ *and* $H_{w,\overline{x}^{(\ell)}}^{-1/2} s^{(\ell+1)}$. *This effectively moves* $\overline{x}^{(\ell)}$ *to* $\overline{x}^{(\ell+1)}$ *while keeping* $(x^{(\ell+1)}, s^{(\ell+1)})$ *unchanged. Then advance timestamp* $\ell$.

  *Each update costs*

  $$\widetilde{O}(\tau^2(\|\delta_{\overline{x}}\|_0 + \|\delta_h\|_0 + \|\delta_{\widetilde{h}}\|_0 + \|\delta_{\epsilon_x}\|_0 + \|\delta_{\epsilon_s}\|_0) + \|\delta_{H_{w,\overline{x}}^{1/2}\widehat{x}}\|_0 + \|\delta_{H_{w,\overline{x}}^{-1/2}\widehat{s}}\|_0 + \|\delta_{c_s}\|_0)$$

  *time.*

- QUERY$x(\ell' \in \mathbb{N}, \epsilon \in \mathbb{R})$: *Given timestamp* $\ell'$, *return a set* $S \subseteq [n]$ *where*

  $$S \supseteq \{i \in [n] : \|H_{w,\overline{x}^{(\ell')}}^{1/2} x_i^{(\ell')} - H_{w,\overline{x}^{(\ell)}}^{1/2} x_i^{(\ell+1)}\|_2 \geq \epsilon\},$$

  *and*

  $$|S| = O(\epsilon^{-2}(\ell - \ell' + 1) \sum_{\ell' \leq t \leq \ell} \|H_{w,\overline{x}^{(t)}}^{1/2} x^{(t)} - H_{w,\overline{x}^{(t)}}^{1/2} x^{(t+1)}\|_2^2 + \sum_{\ell' \leq t \leq \ell-1} \|\overline{x}^{(t)} - \overline{x}^{(t+1)}\|_{2,0})$$

  *where* $\ell$ *is the current timestamp.*

  *For every query, with probability at least* $1 - \delta$, *the return values are correct, and costs at most*

  $$\widetilde{O}(\tau^2 \cdot (\epsilon^{-2}(\ell - \ell' + 1) \sum_{\ell' \leq t \leq \ell} \|H_{\overline{x}^{(t)}}^{1/2} x^{(t)} - H_{\overline{x}^{(t)}}^{1/2} x^{(t+1)}\|_2^2 + \sum_{\ell' \leq t \leq \ell-1} \|\overline{x}^{(t)} - \overline{x}^{(t+1)}\|_{2,0}))$$

  *running time.*

- QUERY$s(\ell' \in \mathbb{N}, \epsilon \in \mathbb{R})$: *Given timestamp* $\ell'$, *return a set* $S \subseteq [n]$ *where*

  $$S \supseteq \{i \in [n] : \|H_{w,\overline{x}^{(\ell')}}^{-1/2} s_i^{(\ell')} - H_{w,\overline{x}^{(\ell)}}^{-1/2} s_i^{(\ell+1)}\|_2 \geq \epsilon\}$$

---

**Algorithm 8** BATCHSKETCH Algorithm 6, 7 continued.

1: **data structure** BATCHSKETCH               ▷ Theorem D.8
2: **private:**
3:   **procedure** QUERY$x$SKETCH($v \in \mathcal{S}$)          ▷ Return the value of $\Phi_{\chi(v)}(H^{1/2}_{w,\overline{x}} x)_{\chi(v)}$
4:     **return** $\mathsf{sketch} H^{1/2}_{w,\overline{x}} \widehat{x}.\text{QUERY}(v) + \mathsf{sketch} \mathcal{W}^{\top} h.\text{QUERY}(v) \cdot \beta_x - \mathsf{sketch} \mathcal{W}^{\top} \widetilde{h}.\text{QUERY}(v) \cdot$
    $\widetilde{\beta}_x + \mathsf{sketch} \mathcal{W}^{\top} \epsilon_x.\text{QUERY}(v)$                ▷ Algorithm 9, 10
5:   **end procedure**
6:   **procedure** QUERY$s$SKETCH($v \in \mathcal{S}$)          ▷ Return the value of $\Phi_{\chi(v)}(H^{-1/2}_{w,\overline{x}} s)_{\chi(v)}$
7:     **return** $\mathsf{sketch} H^{-1/2}_{w,\overline{x}} \widehat{s}.\text{QUERY}(v) + \mathsf{sketch} c_s.\text{QUERY}(v) \cdot \beta_{c_s} - \mathsf{sketch} \mathcal{W}^{\top} h.\text{QUERY}(v) \cdot$
    $\beta_s + \mathsf{sketch} \mathcal{W}^{\top} \widetilde{h}.\text{QUERY}(v) \cdot \widetilde{\beta}_s - \mathsf{sketch} \mathcal{W}^{\top} \epsilon_s.\text{QUERY}(v)$       ▷ Algorithm 9, 10
8:   **end procedure**
9: **public:**
10: **procedure** QUERY$x$($\ell' \in \mathbb{N}, \epsilon \in \mathbb{R}$)
11:     $L_0 = \{\text{root}(\mathcal{S})\}$
12:     $S \leftarrow \emptyset$
13:     **for** $d = 0 \to \infty$ **do**
14:         **if** $L_d = \emptyset$ **then**
15:             **return** $S$
16:         **end if**
17:         $L_{d+1} \leftarrow \emptyset$
18:         **for** $v \in L_d$ **do**
19:             **if** $v$ is a leaf node **then**
20:                 $S \leftarrow S \cup \{v\}$
21:             **else**
22:                 **for** $u$ child of $v$ **do**
23:                     **if** $\|\text{QUERY}x\text{SKETCH}(u) - \text{history}[\ell'].\text{QUERY}x\text{SKETCH}(u)\|_2 > 0.9\epsilon$ **then**
24:                         $L_{d+1} \leftarrow L_{d+1} \cup \{u\}$
25:                     **end if**
26:                 **end for**
27:             **end if**
28:         **end for**
29:     **end for**
30: **end procedure**
31: **procedure** QUERY$s$($\ell' \in \mathbb{N}, \epsilon \in \mathbb{R}$)
32:   Same as QUERY$x$, except for replacing QUERY$x$SKETCH in Line 23 with QUERY$s$SKETCH.
33: **end procedure**
34: **end structure**

---

*and*

$$|S| = O(\epsilon^{-2}(\ell - \ell' + 1) \sum_{\ell' \leq t \leq \ell} \|H^{-1/2}_{w,\overline{x}^{(t)}} s^{(t)} - H^{-1/2}_{w,\overline{x}^{(t)}} s^{(t+1)}\|_2^2 + \sum_{\ell' \leq t \leq \ell-1} \|\overline{x}^{(t)} - \overline{x}^{(t+1)}\|_{2,0})$$

*where $\ell$ is the current timestamp.*

*For every query, with probability at least $1 - \delta$, the return values are correct, and costs at most*

$$\widetilde{O}(\tau^2 \cdot (\epsilon^{-2}(\ell - \ell' + 1) \sum_{\ell' \leq t \leq \ell} \|H^{1/2}_{\overline{x}^{(t)}} s^{(t)} - H^{1/2}_{\overline{x}^{(t)}} x^{(t+1)}\|_2^2 + \sum_{\ell' \leq t \leq \ell-1} \|\overline{x}^{(t)} - \overline{x}^{(t+1)}\|_{2,0}))$$

*running time.*

*Proof.* The proof is essentially the same as proof of (Gu & Song, 2022, Theorem 4.21). For the running time claims, we plug in Lemma D.10 and D.12 when necessary.     □

**Remark D.9** (Snapshot). As in previous works, we use persistent data structures (e.g., Driscoll et al. (1989)) to keep a snapshot of the data structure after every update. This allows us to support query to

historical data. This incurs an $O(\log n_{\text{tot}}) = \widetilde{O}(1)$ multiplicative factor in all running times, which we ignore in our analysis.

### D.3.4 VECTORSKETCH

VECTORSKETCH is a data structure used to maintain sketches of sparsely-changing vectors. It is a direct application of segment trees. For completeness, we include code (Algorithm 9) from (Gu & Song, 2022, Algorithm 10).

---

**Algorithm 9** (Gu & Song, 2022, Algorithm 10). Used in Algorithm 6, 7, 8.

1: **data structure** VECTORSKETCH $\hspace{4cm}$ ▷ Lemma D.10
2:   **private: members**
3:     $\Phi \in \mathbb{R}^{r \times n_{\text{tot}}}$
4:     Partition tree $(\mathcal{S}, \chi)$
5:     $x \in \mathbb{R}^{n_{\text{tot}}}$
6:     Segment tree $\mathcal{T}$ on $[n]$ with values in $\mathbb{R}^r$
7:   **end members**
8:   **procedure** INITIALIZE($\mathcal{S}, \chi$ : partition tree, $\Phi \in \mathbb{R}^{r \times n_{\text{tot}}}, x \in \mathbb{R}^{n_{\text{tot}}}$)
9:     $(\mathcal{S}, \chi) \leftarrow (\mathcal{S}, \chi), \Phi \leftarrow \Phi$
10:     $x \leftarrow x$
11:     Order leaves of $\mathcal{S}$ (variable blocks) such that every node $\chi(v)$ corresponds to a contiguous interval $\subseteq [n]$.
12:     Build a segment tree $\mathcal{T}$ on $[n]$ such that each segment tree interval $I \subseteq [n]$ maintains $\Phi_I x_I \in \mathbb{R}^r$.
13:   **end procedure**
14:   **procedure** UPDATE($\delta_x \in \mathbb{R}^{n_{\text{tot}}}$)
15:     **for** all $i \in [n_{\text{tot}}]$ such that $\delta_{x,i} \neq 0$ **do**
16:       Let $j \in [n]$ be such that $i$ is in $j$-th block
17:       Update $\mathcal{T}$ at $j$-th coordinate $\Phi_j x_j \leftarrow \Phi_j x_j + \Phi_i \cdot \delta_{x,i}$.
18:       $x_i \leftarrow x_i + \delta_{x,i}$
19:     **end for**
20:   **end procedure**
21:   **procedure** QUERY($v \in V(\mathcal{S})$)
22:     Find interval $I$ corresponding to $\chi(v)$
23:     **return** range sum of $\mathcal{T}$ on interval $I$
24:   **end procedure**
25: **end data structure**

---

**Lemma D.10** ((Gu & Song, 2022, Lemma 4.23)). *Given a partition tree $(\mathcal{S}, \chi)$ of $\mathbb{R}^n$, and a JL sketching matrix $\Phi \in \mathbb{R}^{r \times n_{\text{tot}}}$, the data structure* VECTORSKETCH *(Algorithm 9) maintains $\Phi_{\chi(v)} x_{\chi(v)}$ for all nodes $v$ in the partition tree implicitly through the following functions:*

- INITIALIZE($\mathcal{S}, \chi, \Phi$): *Initializes the data structure in $O(r n_{\text{tot}})$ time.*

- UPDATE($\delta_x \in \mathbb{R}^{n_{\text{tot}}}$): *Maintains the data structure for $x \leftarrow x + \delta_x$ in $O(r \|\delta_x\|_0 \log n)$ time.*

- QUERY($v \in V(\mathcal{S})$): *Outputs $\Phi_{\chi(v)} x_{\chi(v)}$ in $O(r \log n)$ time.*

### D.3.5 BALANCEDSKETCH

In this section, we present data structure BALANCEDSKETCH. It is a data structure for maintaining a sketch of a vector of form $\mathcal{W}^\top h$, where $\mathcal{W} = L_{\overline{x}}^{-1} H_{w,\overline{x}}^{1/2}$ and $h \in \mathbb{R}^{n_{\text{tot}}}$ is a sparsely-changing vector. This is a variation of BLOCKBALANCEDSKETCH in Gu & Song (2022).

We use the following construction of a partition tree.

**Definition D.11** (Construction of Partition Tree). We fix an ordering $\pi$ of $[n]$ using the heavy-light decomposition (Lemma A.10). Let $\mathcal{S}$ be a complete binary tree with leaf set $[n]$ and ordering $\pi$. Let $\chi$ map a node to the set of leaves in its subtree. Then $(\mathcal{S}, \chi)$ is a valid partition tree.

---

**Algorithm 10** The BALANCEDSKETCH data structure is used in Algorithm 6, 7, 8.

1: **data structure** BALANCEDSKETCH              ▷ Lemma D.12
2:  **private: members**
3:      $\Phi \in \mathbb{R}^{r \times n_{\text{tot}}}$
4:      Partition tree $(\mathcal{S}, \chi)$ with balanced binary tree $\mathcal{B}$
5:      $t \in \mathbb{N}$
6:      $h \in \mathbb{R}^{n_{\text{tot}}}, \overline{x} \in \mathbb{R}^{n_{\text{tot}}}, H_{w,\overline{x}} \in \mathbb{R}^{n_{\text{tot}} \times n_{\text{tot}}}$
7:      $\{L[t] \in \mathbb{R}^{n_{\text{tot}} \times n_{\text{tot}}}\}_{t \geq 0}$
8:      $\{J_v \in \mathbb{R}^{r \times n_{\text{tot}}}\}_{v \in \mathcal{S}}$
9:      $\{Z_v \in \mathbb{R}^{r \times n_{\text{tot}}}\}_{v \in \mathcal{B}}$
10:     $\{y_v^{\triangledown} \in \mathbb{R}^r\}_{v \in \mathcal{B}}$
11:     $\{t_v \in \mathbb{N}\}_{v \in \mathcal{B}}$
12: **end members**
13: **procedure** INITIALIZE($\mathcal{S}, \chi$ : partition tree, $\Phi \in \mathbb{R}^{r \times n_{\text{tot}}}, \overline{x} \in \mathbb{R}^{n_{\text{tot}}}, h \in \mathbb{R}^{n_{\text{tot}} \times k}$)
14:      $(\mathcal{S}, \chi) \leftarrow (\mathcal{S}, \chi), \Phi \leftarrow \Phi$
15:      $t \leftarrow 0, h \leftarrow h$
16:      $H_{w,\overline{x}} \leftarrow \nabla^2 \phi(\overline{x}), B_{\overline{x}} \leftarrow Q + \overline{t} H_{w,\overline{x}}$
17:      Compute lower Cholesky factor $L_{\overline{x}}[t]$ of $B_{\overline{x}}$
18:      **for all** $v \in \mathcal{S}$ **do**
19:          $J_v \leftarrow \Phi_{\chi(v)} H_{w,\overline{x}}^{1/2}$
20:      **end for**
21:      **for all** $v \in \mathcal{B}$ **do**
22:          $Z_v \leftarrow J_v L_{\overline{x}}[t]^{-\top}$
23:          $y_v^{\triangledown} \leftarrow Z_v (I - I_{\Lambda(v)}) h$
24:          $t_v \leftarrow t$
25:      **end for**
26: **end procedure**
27: **procedure** QUERY($v \in \mathcal{S}$)
28:      **if** $v \in \mathcal{S} \backslash \mathcal{B}$ **then**
29:          **return** $J_v \cdot L_{\overline{x}}[t]^{-\top} h$
30:      **end if**
31:      $\Delta_{L_{\overline{x}}} \leftarrow (L_{\overline{x}}[t] - L_{\overline{x}}[t_v]) \cdot I_{\Lambda(v)}$
32:      $\delta_{Z_v} \leftarrow -(L_{\overline{x}}[t]^{-1} \cdot \Delta_{L_{\overline{x}}} \cdot Z_v^{\top})^{\top}$
33:      $Z_v \leftarrow Z_v + \delta_{Z_v}$
34:      $\delta_{y_v^{\triangledown}} \leftarrow \delta_{Z_v} \cdot (I - I_{\Lambda(v)}) h$
35:      $y_v^{\triangledown} \leftarrow y_v^{\triangledown} + \delta_{y_v^{\triangledown}}$
36:      $t_v \leftarrow t$
37:      $y_v^{\triangle} \leftarrow Z_v \cdot I_{\Lambda(v)} \cdot h$
38:      **return** $y_v^{\triangle} + y_v^{\triangledown}$
39: **end procedure**
40: **end data structure**

---

**Lemma D.12.** *Given an elimination tree $\mathcal{T}$ with height $\eta$, a JL matrix $\Phi \in \mathbb{R}^{r \times n_{\text{tot}}}$, and a partition tree $(\mathcal{S}, \chi)$ constructed as in Definition D.11 with height $\widetilde{O}(1)$, the data structure BALANCEDSKETCH (Algorithm 10, 11, 12), maintains $\Phi_{\chi(v)}(\mathcal{W}^\top h)_{\chi(v)}$ for each $v \in V(\mathcal{S})$ through the following operations*

- INITIALIZE($(\mathcal{S}, \chi)$ : *partition tree*, $\Phi \in \mathbb{R}^{n_{\text{tot}}}, \overline{x} \in \mathbb{R}^{n_{\text{tot}}}, h \in \mathbb{R}^{n_{\text{tot}} \times k}$): *Initializes the data structure in $\widetilde{O}(r(n\tau^{\omega-1} + n\tau k))$ time.*

- UPDATE($\delta_{\overline{x}} \in \mathbb{R}^{n_{\text{tot}}}, \delta_h \in \mathbb{R}^{n_{\text{tot}} \times k}$): *Updates all sketches in $\mathcal{S}$ implicitly to reflect $(\mathcal{W}, h)$ updating to $(\mathcal{W}^{\text{new}}, h^{\text{new}})$ in $\widetilde{O}(r\tau^2 k)$ time.*

- QUERY($v \in \mathcal{S}$): *Outputs $\Phi_{\chi(v)}(\mathcal{W}^\top h)_{\chi(v)}$ in $\widetilde{O}(r\tau^2 k)$ time.*

*Proof.* The proof is almost same as the proof of (Gu & Song, 2022, Lemma 4.24). (In fact, our $\mathcal{W}$ is simpler than the one used in Gu & Song (2022).)

---

**Algorithm 11** BALANCEDSKETCH Algorithm 10 continued. This is used in Algorithm 6, 7, 8.

1: **data structure** BALANCEDSKETCH
2: **procedure** UPDATE($\delta_{\overline{x}} \in \mathbb{R}^{n_{\text{tot}}}, \delta_h \in \mathbb{R}^{n_{\text{tot}} \times k}$)
3:      **for** $i \in [n]$ where $\delta_{\overline{x},i} \neq 0$ **do**
4:          UPDATE$\overline{x}(\delta_{\overline{x},i})$
5:      **end for**
6:      **for** all $\delta_{h,i} \neq 0$ **do**
7:          $v \leftarrow \Lambda^{\circ}(i)$
8:          **for** all $u \in \mathcal{P}^{\mathcal{B}}(v)$ **do**
9:              $y_u^{\triangledown} \leftarrow y_v^{\triangledown} + Z_u \cdot I_{\{i\}} \cdot \delta_h$
10:          **end for**
11:      **end for**
12:      $h \leftarrow h + \delta_h$
13: **end procedure**
14: **procedure** UPDATE$\overline{x}(\delta_{\overline{x},i} \in \mathbb{R}^{n_i})$
15:      $t \leftarrow t + 1$
16:      $\overline{x}_i \leftarrow \overline{x}_i + \delta_{\overline{x},i}$
17:      $\Delta_{H_{w,\overline{x}},(i,i)} \leftarrow \nabla^2 \phi_i(\overline{x}_i) - H_{w,\overline{x},(i,i)}$
18:      Compute $\Delta_{L_{\overline{x}}}$ such that $L_{\overline{x}}[t] \leftarrow L_{\overline{x}}[t-1] + \Delta_{L_{\overline{x}}}$ is the lower Cholesky factor of $A(H_{w,\overline{x}} + \Delta_{H_{w,\overline{x}}})^{-1} A^{\top}$
19:      $S \leftarrow \mathcal{P}^{\mathcal{B}}(\Lambda^{\circ}(\mathsf{low}^{\mathcal{T}}(i)))$
20:      UPDATE$L(S, \Delta_{L_{\overline{x}}})$
21:      UPDATE$H(i, \Delta_{H_{w,\overline{x}},(i,i)})$
22: **end procedure**
23: **end data structure**

---

For INITIALIZE running time, we note that computing $Z_v$ for all $v \in \mathcal{B}$ takes $\widetilde{O}(rn\tau^{\omega-1})$ time by (Gu & Song, 2022, Lemma 8.3). Because $Z_v$ is supported on the path from $v$ to the root in $\mathcal{T}$, we know that $\mathrm{nnz}(Z) = O(rn\tau)$. Therefore computing $y_v^{\triangledown}$ for all $v \in \mathcal{B}$ takes $\widetilde{O}(rn\tau k)$ time.

Remaining claims follow from combining proof of (Gu & Song, 2022, Lemma 4.24) and (Gu & Song, 2022, Lemma 8.3). $\qquad\square$

### D.4 ANALYSIS OF CENTRALPATHMAINTENANCE

**Lemma D.13** (Correctness of CENTRALPATHMAINTENANCE)**.** *Algorithm 1 implicitly maintains the primal-dual solution pair $(x, s)$ via representation Eq. (8)(9). It also explicitly maintains $(\overline{x}, \overline{s}) \in \mathbb{R}^{n_{\text{tot}}} \times \mathbb{R}^{n_{\text{tot}}}$ such that $\|\overline{x}_i - x_i\|_{\overline{x}_i} \leq \overline{\epsilon}$ and $\|\overline{s}_i - s_i\|_{\overline{x}_i}^* \leq t\overline{\epsilon}w_i$ for all $i \in [n]$ with probability at least $0.9$.*

*Proof.* We correctly maintain the implicit representation because of correctness of exact.UPDATE (Theorem D.3).

We show that $\|\overline{x}_i - x_i\|_{\overline{x}_i} \leq \overline{\epsilon}$ and $\|\overline{s}_i - s_i\|_{\overline{x}_i}^* \leq t\overline{\epsilon}w_i$ for all $i \in [n]$ (c.f. Algorithm 20, Line 16). approx maintains an $\ell_{\infty}$ approximation of $H_{w,\overline{x}}^{1/2}x$. For $\ell \leq q$, we have

$$\|H_{w,\overline{x}}^{1/2}x^{(\ell+1)} - H_{w,\overline{x}}^{1/2}x^{(\ell)}\|_2 = \|\delta_x\|_{w,\overline{x}} \leq \frac{9}{8}\alpha \leq \zeta_x$$

where the first step from definition of $\|\cdot\|_{w,\overline{x}}$, the second step follows from Lemma F.11, the third step follows from definition of $\zeta_x$.

By Theorem D.6, with probability at least $1 - \delta_{\text{apx}}$, approx correctly maintains $\overline{x}$ such that $\|H_{w,\overline{x}}^{1/2}\overline{x} - H_{w,\overline{x}}^{1/2}x\|_{\infty} \leq \epsilon_{\text{apx},x} \leq \overline{\epsilon}$. Then

$$\|\overline{x}_i - x_i\|_{\overline{x}_i} \leq w_i^{-1/2}\|H_{w,\overline{x}}^{1/2}\overline{x} - H_{w,\overline{x}}^{1/2}x\|_{\infty} \leq w_i^{-1/2}\overline{\epsilon} \leq \overline{\epsilon}.$$

---

**Algorithm 12** BALANCEDSKETCH Algorithm 10, 11 continued. This is used in Algorithm 6, 7, 8.

1: **data structure** BALANCEDSKETCH $\qquad\qquad\qquad\qquad\qquad$ ▷ Lemma D.12
2: **private:**
3: **procedure** UPDATE$L(S \subset \mathcal{B}, \Delta_{L_{\overline{x}}} \in \mathbb{R}^{n_{\mathrm{tot}} \times n_{\mathrm{tot}}})$
4: $\qquad$ **for** all $v \in S$ **do**
5: $\qquad\qquad \delta_{Z_v} \leftarrow -(L_{\overline{x}}[t-1]^{-1}(L_{\overline{x}}[t-1] - L_{\overline{x}}[t_v]) \cdot I_{\Lambda(v)} \cdot Z_v^{\top})^{\top}$
6: $\qquad\qquad \delta'_{Z_v} \leftarrow -(L_{\overline{x}}[t]^{-1} \cdot \Delta_{L_{\overline{x}}} \cdot (Z_v + \delta_{Z_v})^{\top})^{\top}$
7: $\qquad\qquad Z_v \leftarrow Z_v + \delta_{Z_v} + \delta'_{Z_v}$
8: $\qquad\qquad \delta_{y_v^{\triangledown}} \leftarrow (\delta_{Z_v} + \delta'_{Z_v})(I - I_{\Lambda(v)})h$
9: $\qquad\qquad y_v^{\triangledown} \leftarrow y_v^{\triangledown} + \delta_{y_v^{\triangledown}}$
10: $\qquad\qquad t_v \leftarrow t$
11: $\qquad$ **end for**
12: **end procedure**
13: **private:**
14: **procedure** UPDATE$H(i \in [n], \Delta_{H_{w,\overline{x}},(i,i)} \in \mathbb{R}^{n_i \times n_i})$
15: $\qquad$ Find $u$ such that $\chi(u) = \{i\}$
16: $\qquad \Delta_{H_{w,\overline{x}}^{1/2},(i,i)} \leftarrow (H_{w,\overline{x},(i,i)} + \Delta_{H_{w,\overline{x}},(i,i)})^{1/2} - H_{w,\overline{x},(i,i)}^{1/2}$
17: $\qquad \delta_{J_u} \leftarrow \Phi_i \cdot \Delta_{H_{w,\overline{x}}^{1/2},(i,i)}$
18: $\qquad$ **for** all $v \in \mathcal{P}^{\mathcal{S}}(u)$ **do**
19: $\qquad\qquad J_v \leftarrow J_v + \delta_{J_u}$
20: $\qquad\qquad$ **if** $v \in \mathcal{B}$ **then**
21: $\qquad\qquad\qquad \delta_{Z_v} \leftarrow \delta_{J_v} \cdot L_{\overline{x}}[t_v]^{-\top}$
22: $\qquad\qquad\qquad Z_v \leftarrow Z_v + \delta_{Z_v}$
23: $\qquad\qquad\qquad \delta_{y_v^{\triangledown}} \leftarrow \delta_{Z_v} \cdot (I - I_{\Lambda(v)}) \cdot h$
24: $\qquad\qquad\qquad y_v^{\triangledown} \leftarrow y_v^{\triangledown} + \delta_{y_v^{\triangledown}}$
25: $\qquad\qquad$ **end if**
26: $\qquad$ **end for**
27: $\qquad H_{w,\overline{x}} \leftarrow H_{w,\overline{x}} + \Delta_{H_{w,\overline{x}},(i,i)}$
28: **end procedure**
29: **end data structure**

---

Note that the last step is loose by a factor of $w_i^{1/2}$. When $w_i$s are large, we could improve running time by using a tighter choice of $\epsilon_{\mathrm{apx},x}$, as did in Gu & Song (2022). Here we use a loose bound for simplicity of presentation. Same remark applies to $s$.

The proof for $s$ is similar. We have

$$\|H_{w,\overline{x}}^{-1/2}\delta_s\|_2 = \|\delta_s\|_{w,\overline{x}}^* \leq \frac{17}{8}\alpha \cdot t \leq \zeta_s$$

and

$$\|\overline{s}_i - s_i\|_{\overline{x}_i}^* \leq w_i^{1/2}\|H_{w,\overline{x}}^{-1/2}\overline{s} - H_{w,\overline{x}}^{-1/2}s\|_\infty \leq w_i^{1/2}\epsilon_{\mathrm{apx},s} \leq \overline{\epsilon} \cdot \overline{t} \cdot w_i. \qquad \square$$

**Lemma D.14.** *We bound the running time of* CENTRALPATHMAINTENANCE *as following.*

- CENTRALPATHMAINTENANCE.INITIALIZE *takes* $\widetilde{O}(n\tau^{\omega-1} + n\tau m + nm^{\omega-1})$ *time.*

- *If* CENTRALPATHMAINTENANCE.MULTIPLYANDMOVE *is called $N$ times, then it has total running time*

$$\widetilde{O}((Nn^{-1/2} + \log(t_{\max}/t_{\min})) \cdot n(\tau^2 m + \tau m^2)^{1/2}(\tau^{\omega-1} + \tau m + m^{\omega-1})^{1/2}).$$

- CENTRALPATHMAINTENANCE.OUTPUT *takes* $\widetilde{O}(n\tau m)$ *time.*

*Proof.* INITIALIZE part: By Theorem D.3 and D.6.

OUTPUT part: By Theorem D.3.

MULTIPLYANDMOVE part: Between two restarts, the total size of $|L_x|$ returned by approx.QUERY is bounded by $\widetilde{O}(q^2 \zeta_x^2 / \epsilon_{\mathrm{apx},x}^2)$ by Theorem D.6. By plugging in $\zeta_x = 2\alpha$, $\epsilon_{\mathrm{apx},x} = \bar{\epsilon}$, we have $\sum_{\ell \in [q]} |L_x^{(\ell)}| = \widetilde{O}(q^2)$. Similarly, for $s$ we have $\sum_{\ell \in [q]} |L_s^{(\ell)}| = \widetilde{O}(q^2)$.

**Update time:** By Theorem D.3 and D.6, in a sequence of $q$ updates, total cost for update is $\widetilde{O}(q^2(\tau^2 m + \tau m^2))$. So the amortized update cost per iteration is $\widetilde{O}(q(\tau^2 m + \tau m^2))$. The total update cost is

$$\text{number of iterations} \cdot \text{time per iteration} = \widetilde{O}(Nq(\tau^2 m + \tau m^2)).$$

**Init/restart time:** We restart the data structure whenever $k > q$ or $|\bar{t} - t| > \bar{t}\epsilon_t$, so there are $O(N/q + \log(t_{\max}/t_{\min})\epsilon_t^{-1})$ restarts in total. By Theorem D.3 and D.6, time cost per restart is $\widetilde{O}(n(\tau^{\omega-1} + \tau m + m^{\omega-1}))$. So the total initialization time is

$$\text{number of restarts} \cdot \text{time per restart} = \widetilde{O}((N/q + \log(t_{\max}/t_{\min})\epsilon_t^{-1}) \cdot n(\tau^{\omega-1} + \tau m + m^{\omega-1})).$$

**Combine everything:** Overall running time is

$$\widetilde{O}(Nq(\tau^2 m + \tau m^2) + (N/q + \log(t_{\max}/t_{\min})\epsilon_t^{-1}) \cdot n(\tau^{\omega-1} + \tau m + m^{\omega-1})).$$

Taking $\epsilon_t = \frac{1}{2}\bar{\epsilon}$, the optimal choice for $q$ is

$$q = n^{1/2}(\tau^2 m + \tau m^2)^{-1/2}(\tau^{\omega-1} + \tau m + m^{\omega-1})^{1/2},$$

achieving overall running time

$$\widetilde{O}((Nn^{-1/2} + \log(t_{\max}/t_{\min})) \cdot n(\tau^2 m + \tau m^2)^{1/2}(\tau^{\omega-1} + \tau m + m^{\omega-1})^{1/2}). \qquad \square$$

*Proof of Theorem D.2.* Combining Lemma D.13 and D.14. $\qquad \square$

## D.5 PROOF OF MAIN STATEMENT

*Proof of Theorem D.1.* Use CENTRALPATHMAINTENANCE (Algorithm 1) as the maintenance data structure in Algorithm 20. Combining Theorem D.2 and Theorem F.1 finishes the proof. $\qquad \square$

## E ALGORITHM FOR LOW-RANK QP

In this section we present a nearly-linear time algorithm for solving low-rank QP with small number of linear constraints. We briefly describe the outline of this section.

- In Section E.1, we present the main statement of Section E.
- In Section E.2, we present the main data structure CENTRALPATHMAINTENANCE.
- In Section E.3, we present several data structures used in CENTRALPATHMAINTENANCE, including EXACTDS (Section E.3.1), APPROXDS (Section E.3.2), BATCHSKETCH (Section E.3.3).
- In Section E.4, we prove correctness and running time of CENTRALPATHMAINTENANCE data structure.
- In Section E.5, we prove the main result (Theorem E.1).

### E.1 MAIN STATEMENT

We consider programs of the form (16), i.e.,

$$\min_{x \in \mathbb{R}^n} \quad \frac{1}{2} x^\top Q x + c^\top x$$

$$\text{s.t. } Ax = b$$
$$x_i \in \mathcal{K}_i \qquad \forall i \in [n]$$

where $Q \in \mathcal{S}^{n_{\text{tot}}}, c \in \mathbb{R}^{n_{\text{tot}}}, A \in \mathbb{R}^{m \times n_{\text{tot}}}, b \in \mathbb{R}^m, \mathcal{K}_i \subset \mathbb{R}^{n_i}$ is a convex set. For simplicity, we assume that $n_i = O(1)$ for all $i \in [n]$.

**Theorem E.1.** *Consider the convex program* (16). *Let* $\phi_i : \mathcal{K}_i \to \mathbb{R}$ *be a* $\nu_i$-*self-concordant barrier for all* $i \in [n]$. *Suppose the program satisfies the following properties:*

- *Inner radius* $r$: *There exists* $z \in \mathbb{R}^{n_{\text{tot}}}$ *such that* $Az = b$ *and* $B(z, r) \in \mathcal{K}$.

- *Outer radius* $R$: $\mathcal{K} \subseteq B(0, R)$ *where* $0 \in \mathbb{R}^{n_{\text{tot}}}$.

- *Lipschitz constant* $L$: $\|Q\|_{2\to 2} \leq L, \|c\|_2 \leq L$.

- *Low rank: We are given a factorization* $Q = UV^\top$ *where* $U, V \in \mathbb{R}^{n_{\text{tot}} \times k}$.

*Let* $(w_i)_{i \in [n]} \in \mathbb{R}^n_{\geq 1}$ *and* $\kappa = \sum_{i \in [n]} w_i \nu_i$. *Given any* $0 < \epsilon \leq \frac{1}{2}$, *we can find an approximate solution* $x \in \mathcal{K}$ *satisfiying*

$$\frac{1}{2}x^\top Q x + c^\top x \leq \min_{Ax=b, x \in \mathcal{K}} \left( \frac{1}{2}x^\top Q x + c^\top x \right) + \epsilon LR(R+1),$$
$$\|Ax - b\|_1 \leq 3\epsilon(R\|A\|_1 + \|b\|_1),$$

*in expected time*

$$\widetilde{O}((\sqrt{\kappa}n^{-1/2} + \log(R/(r\epsilon))) \cdot n(k+m)^{(\omega+1)/2}).$$

*When* $\max_{i \in [n]} \nu_i = \widetilde{O}(1), w_i = 1$, *the running time simplifies to*

$$\widetilde{O}(n(k+m)^{(\omega+1)/2}) \log(R/(r\epsilon))).$$

## E.2 ALGORITHM STRUCTURE AND CENTRAL PATH MAINTENANCE

Similar to the low-treewidth case, our algorithm is based on the robust IPM. Details of the robust IPM will be given in Section F. During the algorithm, we maintain a primal-dual solution pair $(x, s) \in \mathbb{R}^{n_{\text{tot}}} \times \mathbb{R}^{n_{\text{tot}}}$ on the robust central path. In addition, we maintain a sparsely-changing approximation $(\overline{x}, \overline{s}) \in \mathbb{R}^{n_{\text{tot}}} \times \mathbb{R}^{n_{\text{tot}}}$ to $(x, s)$. In each iteration, we implicitly perform update

$$x \leftarrow x + \overline{t}B_{w,\overline{x},\overline{t}}^{-1/2}(I - P_{w,\overline{x},\overline{t}})B_{w,\overline{x},\overline{t}}^{-1/2}\delta_\mu$$
$$s \leftarrow s + \overline{t}\delta_\mu - \overline{t}^2 H_{w,\overline{x}}B_{w,\overline{x},\overline{t}}^{-1/2}(I - P_{w,\overline{x},\overline{t}})B_{w,\overline{x},\overline{t}}^{-1/2}\delta_\mu$$

where

$$H_{w,\overline{x}} = \nabla^2 \phi_w(\overline{x}) \hspace{4cm} \text{(see Eq. (24))}$$
$$B_{w,\overline{x},\overline{t}} = Q + \overline{t}H_{w,\overline{x}} \hspace{3.5cm} \text{(see Eq. (25))}$$
$$P_{w,\overline{x},\overline{t}} = B_{w,\overline{x},\overline{t}}^{-1/2}A^\top(AB_{w,\overline{x},\overline{t}}^{-1}A^\top)^{-1}AB_{w,\overline{x},\overline{t}}^{-1/2} \hspace{1cm} \text{(see Eq. (26))}$$

and explicitly maintain $(\overline{x}, \overline{s})$ such that they remain close to $(x, s)$ in $\ell_\infty$-distance.

This task is handled by the CENTRALPATHMAINTENANCE data structure, which is our main data structure. The robust IPM algorithm (Algorithm 19, 20) directly calls it in every iteration.

The CENTRALPATHMAINTENANCE data structure (Algorithm 13) has two main sub data structures, EXACTDS (Algorithm 14, 15) and APPROXDS (Algorithm 16). EXACTDS is used to maintain $(x, s)$, and APPROXDS is used to maintain $(\overline{x}, \overline{s})$.

**Theorem E.2.** *Data structure* CENTRALPATHMAINTENANCE *(Algorithm 13) implicitly maintains the central path primal-dual solution pair* $(x, s) \in \mathbb{R}^{n_{\text{tot}}} \times \mathbb{R}^{n_{\text{tot}}}$ *and explicitly maintains its approximation* $(\overline{x}, \overline{s}) \in \mathbb{R}^{n_{\text{tot}}} \times \mathbb{R}^{n_{\text{tot}}}$ *using the following functions:*

- INITIALIZE$(x \in \mathbb{R}^{n_{\text{tot}}}, s \in \mathbb{R}^{n_{\text{tot}}}, t_0 \in \mathbb{R}_{>0}, \epsilon \in (0, 1))$: *Initializes the data structure with initial primal-dual solution pair* $(x, s) \in \mathbb{R}^{n_{\text{tot}}} \times \mathbb{R}^{n_{\text{tot}}}$, *initial central path timestamp* $t_0 \in \mathbb{R}_{>0}$ *in* $\widetilde{O}(n(k^{\omega-1} + m^{\omega-1}))$ *time.*

---

**Algorithm 13** Main algorithm for low-rank QP.

---

1: **data structure** CENTRALPATHMAINTENANCE                   ▷ Theorem E.2
2:   **private : members**
3:     EXACTDS exact                                 ▷ Algorithm 14, 15
4:     APPROXDS approx                           ▷ Algorithm 16
5:     $\ell \in \mathbb{N}$
6:   **end members**
7: **procedure** INITIALIZE($x, s \in \mathbb{R}^{n_{\text{tot}}}, t \in \mathbb{R}_+, \overline{\epsilon} \in (0, 1)$)
8:     exact.INITIALIZE($x, s, x, s, t$)                           ▷ Algorithm 14
9:     $\ell \leftarrow 0$
10:     $w \leftarrow \nu_{\max}, N \leftarrow \sqrt{\kappa} \log n \log \frac{n\kappa R}{\overline{\epsilon} r}$
11:     $q \leftarrow n^{1/2}(k^2 + m^2)^{-1/2}(d^{\omega-1} + m^{\omega-1})^{1/2}$
12:     $\epsilon_{\text{apx},x} \leftarrow \overline{\epsilon}, \zeta_x \leftarrow 2\alpha, \delta_{\text{apx}} \leftarrow \frac{1}{N}$
13:     $\epsilon_{\text{apx},s} \leftarrow \overline{\epsilon} \cdot \overline{t}, \zeta_s \leftarrow 3\alpha\overline{t}$
14:

      approx.INITIALIZE($x, s, h, \widehat{h}, \widetilde{h}, H_{w,\overline{x}}^{1/2}\widehat{x}, H_{w,\overline{x}}^{-1/2}\widehat{s}, \beta_x, \beta_s, \widehat{\beta}_x, \widehat{\beta}_s, \widetilde{\beta}_x, \widetilde{\beta}_s, q, \&\text{exact},$

      $\epsilon_{\text{apx},x}, \epsilon_{\text{apx},s}, \delta_{\text{apx}}$)

15:             ▷ Algorithm 16.Parameters from $x$ to $\widetilde{\beta}_s$ come from exact. $\&$exact is pointer to exact
16: **end procedure**
17: **procedure** MULTIPLYANDMOVE($t \in \mathbb{R}_+$)
18:     $\ell \leftarrow \ell + 1$
19:     **if** $|\overline{t} - t| > \overline{t} \cdot \epsilon_t$ or $\ell > q$ **then**
20:         $x, s \leftarrow$ exact.OUTPUT()                           ▷ Algorithm 15
21:         INITIALIZE($x, s, t, \overline{\epsilon}$)
22:     **end if**
23:     $\beta_x, \beta_s, \widehat{\beta}_x, \widehat{\beta}_s, \widetilde{\beta}_x, \widetilde{\beta}_s \leftarrow$ exact.MOVE()               ▷ Algorithm 14
24:     $\delta_{\overline{x}}, \delta_{\overline{s}} \leftarrow$ approx.MOVEANDQUERY($\beta_x, \beta_s, \widehat{\beta}_x, \widehat{\beta}_s, \widetilde{\beta}_x, \widetilde{\beta}_s$)     ▷ Algorithm 16
25:     $\delta_h, \delta_{\widehat{h}}, \delta_{\widetilde{h}}, \delta_{H_{w,\overline{x}}^{1/2}\widehat{x}}, \delta_{H_{w,\overline{x}}^{-1/2}\widehat{s}} \leftarrow$ exact.UPDATE($\delta_{\overline{x}}, \delta_{\overline{s}}$)     ▷ Algorithm 15
26:     approx.UPDATE($\delta_{\overline{x}}, \delta_h, \delta_{\widehat{h}}, \delta_{\widetilde{h}}, \delta_{H_{w,\overline{x}}^{1/2}\widehat{x}}, \delta_{H_{w,\overline{x}}^{-1/2}\widehat{s}}$)     ▷ Algorithm 16
27: **end procedure**
28: **procedure** OUTPUT()
29:     **return** exact.OUTPUT()                                 ▷ Algorithm 15
30: **end procedure**
31: **end data structure**

---

- MULTIPLYANDMOVE($t \in \mathbb{R}_{>0}$)*: It implicitly maintains*

$$x \leftarrow x + \overline{t}B_{w,\overline{x},\overline{t}}^{-1/2}(I - P_{w,\overline{x},\overline{t}})B_{w,\overline{x},\overline{t}}^{-1/2}\delta_\mu(\overline{x}, \overline{s}, \overline{t})$$
$$s \leftarrow s + \overline{t}\delta_\mu - \overline{t}^2 H_{w,\overline{x}}B_{w,\overline{x},\overline{t}}^{-1/2}(I - P_{w,\overline{x},\overline{t}})B_{w,\overline{x},\overline{t}}^{-1/2}\delta_\mu(\overline{x}, \overline{s}, \overline{t})$$

*where* $H_{w,\overline{x}}$, $B_{w,\overline{x},\overline{t}}$, $P_{w,\overline{x},\overline{t}}$ *are defined in Eq.* (24)(25)(26) *respectively, and* $\overline{t}$ *is some timestamp satisfying* $|\overline{t} - t| \leq \epsilon_t \cdot \overline{t}$.

*It also explicitly maintains* $(\overline{x}, \overline{s}) \in \mathbb{R}^{n_{\text{tot}} \times n_{\text{tot}}}$ *such that* $\|\overline{x}_i - x_i\|_{\overline{x}_i} \leq \overline{\epsilon}$ *and* $\|\overline{s}_i - s_i\|_{\overline{x}_i}^* \leq t\overline{\epsilon}w_i$ *for all* $i \in [n]$ *with probability at least* 0.9.

*Assuming the function is called at most* $N$ *times and* $t$ *decreases from* $t_{\min}$ *to* $t_{\min}$, *the total running time is*

$$\widetilde{O}((Nn^{-1/2} + \log(t_{\max}/t_{\min})) \cdot n(k^{(\omega+1)/2} + m^{(\omega+1)/2})).$$

- OUTPUT*: Computes* $(x, s) \in \mathbb{R}^{n_{\text{tot}}} \times \mathbb{R}^{n_{\text{tot}}}$ *exactly and outputs them in* $\widetilde{O}(n(k+m))$ *time.*

### E.3 Data Structures Used in CentralPathMaintenance

In this section we present several data structures used in CentralPathMaintenance, including:

- ExactDS (Section E.3.1): This data structure maintains an implicit representation of the primal-dual solution pair $(x, s)$. This is directly used by CentralPathMaintenance.

- ApproxDS (Section E.3.2): This data structure explicitly maintains an approximation $(\overline{x}, \overline{s})$ of $(x, s)$. This data structure is directly used by CentralPathMaintenance.

- BatchSketch (Section E.3.3): This data structure maintains a sketch of $(x, s)$. This data structure is used by ApproxDS.

#### E.3.1 ExactDS

In this section we present the data structure ExactDS. It maintains an implicit representation of the primal-dual solution pair $(x, s)$ by maintaining several sparsely-changing vectors (see Eq. (12)(13)).

**Theorem E.3.** *Data structure* ExactDS *(Algorithm 14, 15) implicitly maintains the primal-dual pair* $(x, s) \in \mathbb{R}^{n_{\mathrm{tot}}} \times \mathbb{R}^{n_{\mathrm{tot}}}$, *computable via the expression*

$$x = \widehat{x} + H_{w,\overline{x}}^{-1/2} h \beta_x + H_{w,\overline{x}}^{-1/2} \widehat{h} \widehat{\beta}_x + H_{w,\overline{x}}^{-1/2} \widetilde{h} \widetilde{\beta}_x, \tag{12}$$

$$s = \widehat{s} + H_{w,\overline{x}}^{1/2} h \beta_s + H_{w,\overline{x}}^{1/2} \widehat{h} \widehat{\beta}_s + H_{w,\overline{x}}^{1/2} \widetilde{h} \widetilde{\beta}_s, \tag{13}$$

*where* $\widehat{x}, \widehat{s} \in \mathbb{R}^{n_{\mathrm{tot}}}$, $h = H_{w,\overline{x}}^{-1/2} \overline{\delta}_\mu \in \mathbb{R}^{n_{\mathrm{tot}}}$, $\widehat{h} = H_{w,\overline{x}}^{-1/2} U^\top \in \mathbb{R}^{n_{\mathrm{tot}} \times k}$, $\widetilde{h} = H_{w,\overline{x}}^{-1/2} A^\top \in \mathbb{R}^{n_{\mathrm{tot}} \times m}$, $\beta_x, \beta_s \in \mathbb{R}$, $\widehat{\beta}_x, \widehat{\beta}_s \in \mathbb{R}^k$, $\widetilde{\beta}_x, \widetilde{\beta}_s \in \mathbb{R}^m$.

*The data structure supports the following functions:*

- Initialize$(x, s, \overline{x}, \overline{s} \in \mathbb{R}^{n_{\mathrm{tot}}}, \overline{t} \in \mathbb{R}_{>0})$: *Initializes the data structure in* $\widetilde{O}(n(k^\omega + m^\omega))$ *time, with initial value of the primal-dual pair* $(x, s)$, *its initial approximation* $(\overline{x}, \overline{s})$, *and initial approximate timestamp* $\overline{t}$.

- Move()*: Performs robust central path step*

$$x \leftarrow x + \overline{t} B_{\overline{x}}^{-1} \delta_\mu - \overline{t} B_{\overline{x}}^{-1} A^\top (A B_{\overline{x}}^{-1} A^\top)^{-1} A B_{\overline{x}}^{-1} \delta_\mu, \tag{14}$$

$$s \leftarrow s + \overline{t} \delta_\mu - \overline{t}^2 B_{\overline{x}}^{-1} \delta_\mu + \overline{t}^2 B_{\overline{x}}^{-1} A^\top (A B_{\overline{x}}^{-1} A^\top)^{-1} A B_{\overline{x}}^{-1} \delta_\mu \tag{15}$$

  *in* $O(k^\omega + m^\omega)$ *time by updating its implicit representation.*

- Update$(\delta_{\overline{x}}, \delta_{\overline{s}} \in \mathbb{R}^{n_{\mathrm{tot}}})$: *Updates the approximation pair* $(\overline{x}, \overline{s})$ *to* $(\overline{x}^{\mathrm{new}} = \overline{x} + \delta_{\overline{x}} \in \mathbb{R}^{n_{\mathrm{tot}}}, \overline{s}^{\mathrm{new}} = \overline{s} + \delta_{\overline{s}} \in \mathbb{R}^{n_{\mathrm{tot}}})$ *in* $\widetilde{O}((k^2 + m^2)(\|\delta_{\overline{x}}\|_0 + \|\delta_{\overline{s}}\|_0))$ *time, and output the changes in variables* $h, \widehat{h}, \widetilde{h}, H_{w,\overline{x}}^{1/2} \widehat{x}, H_{w,\overline{x}}^{-1/2} \widehat{s}$.

  *Furthermore,* $h, H_{w,\overline{x}}^{1/2} \widehat{x}, H_{w,\overline{x}}^{-1/2} \widehat{s}$ *changes in* $O(\|\delta_{\overline{x}}\|_0 + \|\delta_{\overline{s}}\|_0)$ *coordinates,* $\widehat{h}$ *changes in* $O(k(\|\delta_{\overline{x}}\|_0 + \|\delta_{\overline{s}}\|_0))$ *coordinates,* $\widetilde{h}$ *changes in* $O(m(\|\delta_{\overline{x}}\|_0 + \|\delta_{\overline{s}}\|_0))$ *coordinates.*

- Output()*: Output* $x$ *and* $s$ *in* $\widetilde{O}(n(k + m))$ *time.*

- Query$x(i \in [n])$: *Output* $x_i$ *in* $\widetilde{O}(k + m)$ *time. This function is used by* ApproxDS.

- Query$s(i \in [n])$: *Output* $s_i$ *in* $\widetilde{O}(k + m)$ *time. This function is used by* ApproxDS.

*Proof of Theorem E.3.* By combining Lemma E.4 and E.5. $\qquad\square$

**Lemma E.4.** ExactDS *correctly maintains an implicit representation of* $(x, s)$, *i.e., invariant*

$$x = \widehat{x} + H_{w,\overline{x}}^{-1/2} h \beta_x + H_{w,\overline{x}}^{-1/2} \widehat{h} \widehat{\beta}_x + H_{w,\overline{x}}^{-1/2} \widetilde{h} \widetilde{\beta}_x,$$

$$s = \widehat{s} + H_{w,\overline{x}}^{1/2} h \beta_s + H_{w,\overline{x}}^{1/2} \widehat{h} \widehat{\beta}_s + H_{w,\overline{x}}^{1/2} \widetilde{h} \widetilde{\beta}_s,$$

$$h = H_{w,\overline{x}}^{-1/2} \overline{\delta}_\mu \in \mathbb{R}^{n_{\mathrm{tot}}}, \widehat{h} = H_{w,\overline{x}}^{-1/2} U^\top \in \mathbb{R}^{n_{\mathrm{tot}} \times d}, \widetilde{h} = H_{w,\overline{x}}^{-1/2} A^\top \in \mathbb{R}^{n_{\mathrm{tot}} \times m},$$

---

**Algorithm 14** This is used in Algorithm 13.

1: **data structure** EXACTDS        ▷ Theorem E.3
2:   **members**
3:     $\overline{x}, \overline{s} \in \mathbb{R}^{n_{\text{tot}}}, \overline{t} \in \mathbb{R}_+, H_{w,\overline{x}} \in \mathbb{R}^{n_{\text{tot}} \times n_{\text{tot}}}$
4:     $\widehat{x}, \widehat{s}, \in \mathbb{R}^{n_{\text{tot}}}, \widehat{h} \in \mathbb{R}^{n_{\text{tot}} \times k}, \widetilde{h} \in \mathbb{R}^{n_{\text{tot}} \times m}, \beta_x, \beta_s \in \mathbb{R}, \widehat{\beta}_x, \widehat{\beta}_s \in \mathbb{R}^d, \widetilde{\beta}_x, \widetilde{\beta}_s \in \mathbb{R}^m$
5:     $u_1, u_2 \in \mathbb{R}^{k \times m}, u_3 \in \mathbb{R}^{m \times m}, u_4 \in \mathbb{R}^m, u_5 \in \mathbb{R}^d, u_6 \in \mathbb{R}^{k \times k}$
6:     $\overline{\alpha} \in \mathbb{R}, \overline{\delta}_\mu \in \mathbb{R}^n$
7:     $K \in \mathbb{N}$
8:   **end members**
9:   **procedure** INITIALIZE$(x, s, \overline{x}, \overline{s} \in \mathbb{R}^{n_{\text{tot}}}, \overline{t} \in \mathbb{R}_+)$
10:     $\overline{x} \leftarrow \overline{x}, \overline{x} \leftarrow \overline{s}, \overline{t} \leftarrow \overline{t}$
11:     $\widehat{x} \leftarrow x, \widehat{s} \leftarrow s, \beta_x \leftarrow 0, \beta_s \leftarrow 0, \widehat{\beta}_x \leftarrow 0, \widehat{\beta}_s \leftarrow 0, \widetilde{\beta}_x \leftarrow 0, \widetilde{\beta}_s \leftarrow 0$
12:     $H_{w,\overline{x}} \leftarrow \nabla^2 \phi_w(\overline{x})$
13:     INITIALIZE$h(\overline{x}, \overline{s}, H_{w,\overline{x}})$
14:   **end procedure**
15:   **procedure** INITIALIZE$h(\overline{x}, \overline{s} \in \mathbb{R}^{n_{\text{tot}}}, H_{w,\overline{x}} \in \mathbb{R}^{n_{\text{tot}} \times n_{\text{tot}}})$
16:     **for** $i \in [n]$ **do**
17:         $(\overline{\delta}_\mu)_i \leftarrow -\frac{\alpha \sinh(\frac{\lambda}{w_i} \gamma_i(\overline{x},\overline{s},\overline{t}))}{\gamma_i(\overline{x},\overline{s},\overline{t})} \cdot \mu_i(\overline{x}, \overline{s}, \overline{t})$
18:         $\overline{\alpha} \leftarrow \overline{\alpha} + w_i^{-1} \cosh^2(\frac{\lambda}{w_i} \gamma_i(\overline{x}, \overline{s}, \overline{t}))$
19:     **end for**
20:     $h \leftarrow H_{w,\overline{x}}^{-1/2} \overline{\delta}_\mu, \widehat{h} \leftarrow H_{w,\overline{x}}^{-1/2} U^\top, \widetilde{h} \leftarrow H_{w,\overline{x}}^{-1/2} A^\top$
21:     $u_1 \leftarrow U H_{w,\overline{x}}^{-1} A^\top, u_2 \leftarrow V H_{w,\overline{x}}^{-1} A^\top, u_3 \leftarrow A H_{w,\overline{x}}^{-1} A^\top$
22:     $u_4 \leftarrow A H_{w,\overline{x}}^{-1} \overline{\delta}_\mu, u_5 \leftarrow V H_{w,\overline{x}}^{-1} \overline{\delta}_\mu, u_6 \leftarrow V H_{w,\overline{x}}^{-1} U^\top$
23:   **end procedure**
24:   **procedure** MOVE()
25:     $v_0 \leftarrow I + \overline{t}^{-1} u_6 \in \mathbb{R}^{k \times k}$
26:     $v_1 \leftarrow \overline{t}^{-1} u_3 - \overline{t}^{-2} u_1^\top v_0^{-1} u_2 \in \mathbb{R}^{m \times m}$
27:     $v_2 \leftarrow \overline{t}^{-1} u_4 - \overline{t}^{-2} u_1^\top v_0^{-1} u_5 \in \mathbb{R}^m$
28:     $\beta_x \leftarrow \beta_x + (\overline{\alpha})^{-1/2}$
29:     $\widehat{\beta}_x \leftarrow \widehat{\beta}_x - (\overline{\alpha})^{-1/2} \cdot \overline{t}^{-1} v_0^{-1} u_5 + (\overline{\alpha})^{-1/2} \cdot \overline{t}^{-1} v_0^{-1} u_2 v_1^{-1} v_2$
30:     $\widetilde{\beta}_x \leftarrow \widetilde{\beta}_x - (\overline{\alpha})^{-1/2} \cdot v_1^{-1} v_2$
31:     $\beta_s \leftarrow \beta_s$
32:     $\widehat{\beta}_s \leftarrow \widehat{\beta}_s + (\overline{\alpha})^{-1/2} \cdot v_0^{-1} u_5 - (\overline{\alpha})^{-1/2} \cdot v_0^{-1} u_2 v_1^{-1} v_2$
33:     $\widetilde{\beta}_s \leftarrow \widetilde{\beta}_s + (\overline{\alpha})^{-1/2} \cdot \overline{t} v_1^{-1} v_2$
34:     **return** $\beta_x, \beta_s, \widehat{\beta}_x, \widehat{\beta}_s, \widetilde{\beta}_x, \widetilde{\beta}_s$
35:   **end procedure**
36: **end data structure**

---

$$u_1 = U H_{w,\overline{x}}^{-1} A^\top \in \mathbb{R}^{d \times m}, u_2 = V H_{w,\overline{x}}^{-1} A^\top \in \mathbb{R}^{d \times m}, u_3 = A H_{w,\overline{x}}^{-1} A^\top \in \mathbb{R}^{m \times m},$$

$$u_4 = A H_{w,\overline{x}}^{-1} \overline{\delta}_\mu \in \mathbb{R}^m, u_5 = V H_{w,\overline{x}}^{-1} \overline{\delta}_\mu \in \mathbb{R}^d, u_6 = V H_{w,\overline{x}}^{-1} U^\top \in \mathbb{R}^{d \times d},$$

$$\overline{\alpha} = \sum_{i \in [n]} w_i^{-1} \cosh^2(\frac{\lambda}{w_i} \gamma_i(\overline{x}, \overline{s}, \overline{t})),$$

$$\overline{\delta}_\mu = \overline{\alpha}^{1/2} \delta_\mu(\overline{x}, \overline{s}, \overline{t})$$

*always holds after every external call, and return values of the queries are correct.*

*Proof.* INITIALIZE: By checking the definitions we see that all invariants are satisfied after INITIAL-IZE.

MOVE: By the invariants, we have

$$v_0 = I + \overline{t}^{-1} V H_{w,\overline{x}}^{-1} U^\top,$$

---

**Algorithm 15** Algorithm 14 continued.

1: **data structure** EXACTDS                                                                ▷ Theorem E.3
2:   **procedure** OUTPUT()
3:     **return** $\widehat{x} + H_{w,\overline{x}}^{-1/2} h \beta_x + H_{w,\overline{x}}^{-1/2} \widehat{h} \widehat{\beta}_x + H_{w,\overline{x}}^{-1/2} \widetilde{h} \widetilde{\beta}_x, \widehat{s} + H_{w,\overline{x}}^{1/2} h \beta_s + H_{w,\overline{x}}^{1/2} \widehat{h} \widehat{\beta}_s + H_{w,\overline{x}}^{1/2} \widetilde{h} \widetilde{\beta}_s$
4:   **end procedure**
5:   **procedure** QUERY$x(i \in [n])$
6:     **return** $\widehat{x}_i + H_{w,\overline{x}}^{-1/2} h_{i,*} \beta_x + H_{w,\overline{x}}^{-1/2} \widehat{h}_{i,*} \widehat{\beta}_x + H_{w,\overline{x}}^{-1/2} \widetilde{h}_{i,*} \widetilde{\beta}_x$
7:   **end procedure**
8:   **procedure** QUERY$s(i \in [n])$
9:     **return** $\widehat{s}_i + H_{w,\overline{x}}^{1/2} h_{i,*} \beta_s + H_{w,\overline{x}}^{1/2} \widehat{h}_{i,*} \widehat{\beta}_s + H_{w,\overline{x}}^{1/2} \widetilde{h}_{i,*} \widetilde{\beta}_s$
10: **end procedure**
11: **procedure** UPDATE($\delta_{\overline{x}}, \delta_{\overline{s}} \in \mathbb{R}^{n_{\text{tot}}}$)
12:   $\Delta_{H_{w,\overline{x}}} \leftarrow \nabla^2 \phi_w(\overline{x} + \delta_{\overline{x}}) - H_{w,\overline{x}}$  ▷ $\Delta_{H_{w,\overline{x}}}$ is non-zero only for diagonal blocks $(i,i)$ for which $\delta_{\overline{x},i} \neq 0$
13:   $S \leftarrow \{i \in [n] \mid \delta_{\overline{x},i} \neq 0 \text{ or } \delta_{\overline{s},i} \neq 0\}$
14:   $\delta_{\overline{\delta}_\mu} \leftarrow 0$
15:   **for** $i \in S$ **do**
16:     Let $\gamma_i = \gamma_i(\overline{x}, \overline{s}, \overline{t}), \gamma_i^{\text{new}} = \gamma_i(\overline{x} + \delta_{\overline{x}}, \overline{s} + \delta_{\overline{s}}, \overline{t}), \mu_i^{\text{new}} = \mu_i(\overline{x} + \delta_{\overline{x}}, \overline{s} + \delta_{\overline{s}}, \overline{t})$
17:     $\overline{\alpha} \leftarrow \overline{\alpha} - w_i^{-1} \cosh^2(\frac{\lambda}{w_i} \gamma_i) + w_i^{-1} \cosh^2(\frac{\lambda}{w_i} \gamma_i^{\text{new}})$
18:     $\delta_{\overline{\delta}_\mu, i} \leftarrow -\alpha \sinh(\frac{\lambda}{w_i} \gamma_i^{\text{new}}) \cdot \frac{1}{\gamma_i^{\text{new}}} \cdot \mu_i^{\text{new}} - \overline{\delta}_{\mu,i}$
19:   **end for**
20:   $\delta_h \leftarrow \Delta_{H_{w,\overline{x}}^{-1/2}}(\overline{\delta}_\mu + \delta_{\overline{\delta}_\mu}) + H_{w,\overline{x}}^{-1/2} \delta_{\overline{\delta}_\mu}$
21:   $\delta_{\widehat{h}} \leftarrow \Delta_{H_{w,\overline{x}}^{-1/2}} U^\top$
22:   $\delta_{\widetilde{h}} \leftarrow \Delta_{H_{w,\overline{x}}^{-1/2}} A^\top$
23:   $\delta_{\widehat{x}} \leftarrow -(\delta_h \beta_x + \delta_{\widehat{h}} \widehat{\beta}_x + \delta_{\widetilde{h}} \widetilde{\beta}_x)$
24:   $\delta_{\widehat{s}} \leftarrow -(\delta_h \beta_s + \delta_{\widehat{h}} \widehat{\beta}_s + \delta_{\widetilde{h}} \widetilde{\beta}_s)$
25:   $h \leftarrow h + \delta_h, \widehat{h} \leftarrow \widehat{h} + \delta_{\widehat{h}}, \widetilde{h} \leftarrow \widetilde{h} + \delta_{\widetilde{h}}, \widehat{x} \leftarrow \widehat{x} + \delta_{\widehat{x}}, \widehat{s} \leftarrow \widehat{s} + \delta_{\widehat{s}}$
26:   $u_1 \leftarrow u_1 + U \Delta_{H_{w,\overline{x}}^{-1}} A^\top$
27:   $u_2 \leftarrow u_2 + V \Delta_{H_{w,\overline{x}}^{-1}} A^\top$
28:   $u_3 \leftarrow u_3 + A \Delta_{H_{w,\overline{x}}^{-1}} A^\top$
29:   $u_4 \leftarrow u_4 + A(\Delta_{H_{w,\overline{x}}^{-1}}(\overline{\delta}_\mu + \delta_{\overline{\delta}_\mu}) + H_{w,\overline{x}}^{-1} \delta_{\overline{\delta}_\mu})$
30:   $u_5 \leftarrow u_5 + V(\Delta_{H_{w,\overline{x}}^{-1}}(\overline{\delta}_\mu + \delta_{\overline{\delta}_\mu}) + H_{w,\overline{x}}^{-1} \delta_{\overline{\delta}_\mu})$
31:   $u_6 \leftarrow u_6 + V \Delta_{H_{w,\overline{x}}^{-1}} U^\top$
32:   $\overline{x} \leftarrow \overline{x} + \delta_{\overline{x}}, \overline{s} \leftarrow \overline{s} + \delta_{\overline{s}}$
33:   $H_{w,\overline{x}} \leftarrow H_{w,\overline{x}} + \Delta_{H_{w,\overline{x}}}$
34:   **return** $\delta_h, \delta_{\widehat{h}}, \delta_{\widetilde{h}}, \delta_{H_{w,\overline{x}}^{1/2} \widehat{x}}, \delta_{H_{w,\overline{x}}^{-1/2} \widehat{s}}$
35: **end procedure**
36: **end data structure**

---

$$v_1 = \overline{t}^{-1} A H_{w,\overline{x}}^{-1} A^\top - \overline{t}^{-1} A H_{w,\overline{x}}^{-1} U^\top (I + \overline{t}^{-1} V H_{w,\overline{x}}^{-1} U^\top)^{-1} V H_{w,\overline{x}} A^\top$$
$$= A B_{\overline{x}}^{-1} A^\top$$
$$v_2 = \overline{t}^{-1} A H_{w,\overline{x}}^{-1} \overline{\delta}_\mu - \overline{t}^{-1} A H_{w,\overline{x}}^{-1} U^\top (I + \overline{t}^{-1} V H_{w,\overline{x}}^{-1} U^\top)^{-1} V H_{w,\overline{x}} \overline{\delta}_\mu$$
$$= A B_{\overline{x}}^{-1} \overline{\delta}_\mu.$$

By implicit representation (12),

$$\delta_x = H_{w,\overline{x}}^{-1/2} h \delta_{\beta_x} + H_{w,\overline{x}}^{-1/2} \widehat{h} \delta_{\widehat{\beta}_x} + H_{w,\overline{x}}^{-1/2} \widetilde{h} \delta_{\widetilde{\beta}_x}$$
$$= H_{w,\overline{x}}^{-1} \overline{\delta}_\mu \cdot (\overline{\alpha})^{-1/2}$$
$$\quad + H_{w,\overline{x}}^{-1} U^\top \cdot (\overline{\alpha})^{-1/2} \overline{t}^{-1} v_0^{-1} (-u_5 + u_2 v_1^{-1} v_2)$$

$$
\begin{aligned}
&\quad - H_{w,\overline{x}}^{-1} A^\top \cdot (\overline{\alpha})^{-1/2} v_1^{-1} v_2 \\
&= H_{w,\overline{x}}^{-1} \delta_\mu \\
&\qquad + H_{w,\overline{x}}^{-1} U^\top \overline{t}^{-1} (I + \overline{t}^{-1} V H_{w,\overline{x}}^{-1} U^\top)^{-1} (-V H_{w,\overline{x}}^{-1} \delta_\mu + V H_{w,\overline{x}}^{-1} A^\top (A B_{\overline{x}}^{-1} A^\top)^{-1} A B_{\overline{x}}^{-1} \delta_\mu) \\
&\qquad - H_{w,\overline{x}}^{-1} A^\top (A B_{\overline{x}}^{-1} A^\top)^{-1} A B_{\overline{x}}^{-1} \delta_\mu \\
&= \overline{t} \cdot (\overline{t}^{-1} H_{w,\overline{x}}^{-1} - \overline{t}^{-2} H_{w,\overline{x}}^{-1} U^\top (I + \overline{t}^{-1} V H_{w,\overline{x}}^{-1} U^\top)^{-1} V H_{w,\overline{x}}^{-1}) \delta_\mu \\
&\qquad - \overline{t} (\overline{t}^{-1} H_{w,\overline{x}}^{-1} - \overline{t}^{-2} \overline{t}^{-2} H_{w,\overline{x}}^{-1} U^\top (I + \overline{t}^{-1} V H_{w,\overline{x}}^{-1} U^\top)^{-1} V H_{w,\overline{x}}^{-1}) A^\top (A B_{\overline{x}}^{-1} A^\top)^{-1} A B_{\overline{x}}^{-1} \delta_\mu \\
&= \overline{t} B_{\overline{x}}^{-1} \delta_\mu - \overline{t} B_{\overline{x}}^{-1} A^\top (A B_{\overline{x}}^{-1} A^\top)^{-1} A B_{\overline{x}}^{-1} \delta_\mu.
\end{aligned}
$$

Comparing with the robust central path step (14), we see that $x$ is updated correctly.

For $s$, from implicit representation 13 we have

$$
\begin{aligned}
\delta_s &= H_{w,\overline{x}}^{1/2} h \delta_{\beta_x} + H_{w,\overline{x}}^{1/2} \widehat{h} \delta_{\widehat{\beta}_x} + H_{w,\overline{x}}^{1/2} \widetilde{h} \delta_{\widetilde{\beta}_x} \\
&= -U^\top \cdot (\overline{\alpha})^{-1/2} \cdot v_0^{-1} (-u_5 + u_2 v_1^{-1} v_2) + A^\top \cdot (\overline{\alpha})^{-1/2} \cdot \overline{t} v_1^{-1} v_2 \\
&= \overline{t} \delta_\mu - \overline{t}^2 B_{\overline{x}}^{-1} \delta_\mu + \overline{t}^2 B_{\overline{x}}^{-1} A^\top (A B_{\overline{x}}^{-1} A^\top)^{-1} A B_{\overline{x}}^{-1} \delta_\mu.
\end{aligned}
$$

Comparing with robust central path step (15), we see that $s$ is updated correctly.

UPDATE: We would like to prove that UPDATE correctly updates the values of $\widehat{x}, \widehat{s}, h, \widehat{h}, \widetilde{h}$, $u_1, u_2, u_3, u_4, u_5, u_6, \overline{\alpha}, \overline{\delta}_\mu$, while preserving the values of $(x, s)$. In fact, by checking the definitions, it is easy to see that $h, \widehat{h}, \widetilde{h}, u_1, u_2, u_3, u_4, u_5, u_6, \overline{\alpha}, \overline{\delta}_\mu$ are updated correctly. Furthermore

$$
\begin{aligned}
\delta_x &= \delta_{\widehat{x}} + \delta_h \beta_x + \delta_{\widehat{h}} \widehat{\beta}_x + \delta_{\widetilde{h}} \widetilde{\beta}_x = 0, \\
\delta_s &= \delta_{\widehat{s}} + \delta_h \beta_s + \delta_{\widehat{h}} \widehat{\beta}_s + \delta_{\widetilde{h}} \widetilde{\beta}_s = 0.
\end{aligned}
$$

So values of $(x, s)$ are preserved. $\qquad\square$

**Lemma E.5.** *We bound the running time of* EXACTDS *as following.*

(i) EXACTDS.INITIALIZE *(Algorithm 14) runs in* $\widetilde{O}(n(k^{\omega-1} + m^{\omega-1}))$ *time.*

(ii) EXACTDS.MOVE *(Algorithm 14) runs in* $\widetilde{O}(k^\omega + m^\omega)$ *time.*

(iii) EXACTDS.OUTPUT *(Algorithm 15) runs in* $\widetilde{O}(n(k + m))$ *time and correctly outputs* $(x, s)$.

(iv) EXACTDS.QUERY$x$ *and* EXACTDS.QUERY$s$ *(Algorithm 15) runs in* $\widetilde{O}(k + m)$ *time and returns the correct answer.*

(v) EXACTDS.UPDATE *(Algorithm 15) runs in* $\widetilde{O}((k^2 + m^2)(\|\delta_{\overline{x}}\|_0 + \|\delta_{\overline{s}}\|_0))$ *time. Furthermore,* $\|\delta_h\|_0, \|\delta_{\widehat{x}}\|_0, \|\delta_{\widehat{s}}\|_0 = O(\|\delta_{\overline{x}}\|_0 + \|\delta_{\overline{s}}\|_0)$, $\mathrm{nnz}(\widehat{h}) = O(d(\|\delta_{\overline{x}}\|_0 + \|\delta_{\overline{s}}\|_0))$, $\mathrm{nnz}(\widetilde{h}) = O(m(\|\delta_{\overline{x}}\|_0 + \|\delta_{\overline{s}}\|_0))$.

*Proof.* (i) EXACTDS.INITIALIZE: Computing $u_1$ and $u_2$ takes $\mathcal{T}_{\mathrm{mat}}(k, n, m) = \widetilde{O}(n(k^{\omega-1} + m^{\omega-1}))$ time. Computing $u_3$ takes $\mathcal{T}_{\mathrm{mat}}(m, n, m) = \widetilde{O}(nm^{\omega-1})$ time. Computing $u_4$ takes $O(nm)$ time. Computing $u_5$ takes $O(nk)$ time. Computing $u_6$ takes $\mathcal{T}_{\mathrm{mat}}(k, n, k) = \widetilde{O}(nk^{\omega-1})$ time. All other computations are cheaper.

(ii) EXACTDS.MOVE: Computing $v_0^{-1}$ takes $\widetilde{O}(k^\omega)$ time. Computing $v_1^{-1}$ takes $\widetilde{O}(m^\omega)$ time. All other computations are cheaper.

(iii) EXACTDS.OUTPUT: Takes $\widetilde{O}(n(k + m))$ time.

(iv) EXACTDS.QUERY$x$ and EXACTDS.QUERY$s$: Takes $\widetilde{O}(k + m)$ time.

(v) EXACTDS.UPDATE: For simplicity, write $t = \|\delta_{\overline{x}}\|_0 + \|\delta_{\overline{x}}\|_0$. Computing $\delta_h$ takes $\widetilde{O}(t)$ time. Computing $\delta_{\widehat{h}}$ takes $\widetilde{O}(tk)$ time. Computing $\delta_{\widetilde{h}}$ takes $\widetilde{O}(tm)$ time. Computing $\delta_{\widehat{s}}$ and $\delta_{\widehat{s}}$ takes $\widetilde{O}(t(k+m))$ time. The sparsity statements follow directly. Computing $u_1$ and $u_2$ takes $\widetilde{O}(tkm)$ time. Computing $u_3$ takes $\widetilde{O}(tm^2)$ time. Computing $u_4$ takes $\widetilde{O}(tm)$ time. Computing $u_5$ takes $\widetilde{O}(tk)$ time. Computing $u_6$ takes $\widetilde{O}(tk^2)$ time. □

### E.3.2 APPROXDS

In this section we present the data structure APPROXDS. Given BATCHSKETCH, a data structure maintaining a sketch of the primal-dual pair $(x, s) \in \mathbb{R}^{n_{\text{tot}}} \times \mathbb{R}^{n_{\text{tot}}}$, APPROXDS maintains a sparsely-changing $\ell_\infty$-approximation of $(x, s)$.

---

**Algorithm 16** This is used in Algorithm 13.

1: **data structure** APPROXDS              ▷ Theorem E.6
2: **private : members**
3:   $\epsilon_{\text{apx},x}, \epsilon_{\text{apx},s} \in \mathbb{R}$
4:   $\ell \in \mathbb{N}$
5:   BATCHSKETCH bs    ▷ This maintains a sketch of $H_{w,\overline{x}}^{1/2}x$ and $H_{w,\overline{x}}^{-1/2}s$. See Algorithm 17 and 18.
6:   EXACTDS* exact   ▷ This is a pointer to the EXACTDS (Algorithm 14, 15) we maintain in parallel to APPROXDS.
7:   $\widetilde{x}, \widetilde{s} \in \mathbb{R}^{n_{\text{tot}}}$    ▷ $(\widetilde{x}, \widetilde{s})$ is a sparsely-changing approximation of $(x, s)$. They have the same value as $(\overline{x}, \overline{s})$, but for these local variables we use $(\widetilde{x}, \widetilde{s})$ to avoid confusion.
8: **end members**
9: **procedure** INITIALIZE($x, s \in \mathbb{R}^{n_{\text{tot}}}, h \in \mathbb{R}^{n_{\text{tot}}}, \widehat{h} \in \mathbb{R}^{n_{\text{tot}} \times k}, \widetilde{h} \in \mathbb{R}^{n_{\text{tot}} \times m}, H_{w,\overline{x}}^{1/2}\widehat{x}, H_{w,\overline{x}}^{-1/2}\widehat{s} \in$
  $\mathbb{R}^{n_{\text{tot}}}, \beta_x, \beta_s \in \mathbb{R}, \widehat{\beta}_x, \widehat{\beta}_s \in \mathbb{R}^d, \widetilde{\beta}_x, \widetilde{\beta}_s \in \mathbb{R}^m, q \in \mathbb{N}, \text{EXACTDS* exact}, \epsilon_{\text{apx},x}, \epsilon_{\text{apx},s}, \delta_{\text{apx}} \in \mathbb{R}$)
10:   $\ell \leftarrow 0, q \leftarrow q$
11:   $\epsilon_{\text{apx},x} \leftarrow \epsilon_{\text{apx},x}, \epsilon_{\text{apx},s} \leftarrow \epsilon_{\text{apx},s}$
12:   bs.INITIALIZE($x, h, \widehat{h}, \widetilde{h}, H_{w,\overline{x}}^{1/2}\widehat{x}, H_{w,\overline{x}}^{-1/2}\widehat{s}, \beta_x, \beta_s, \widehat{\beta}_x, \widehat{\beta}_s, \widetilde{\beta}_x, \widetilde{\beta}_s, \delta_{\text{apx}}/q$)    ▷ Algorithm 17
13:   $\widetilde{x} \leftarrow x, \widetilde{s} \leftarrow s$
14:   exact $\leftarrow$ exact
15: **end procedure**
16: **procedure** UPDATE($\delta_{\overline{x}} \in \mathbb{R}^{n_{\text{tot}}}, \delta_h \in \mathbb{R}^{n_{\text{tot}}}, \delta_{\widehat{h}} \in \mathbb{R}^{n_{\text{tot}} \times k}, \delta_{\widetilde{h}} \in \mathbb{R}^{n_{\text{tot}} \times m}, \delta_{H_{w,\overline{x}}^{1/2}\widehat{x}}, \delta_{H_{w,\overline{x}}^{-1/2}\widehat{s}} \in \mathbb{R}^{n_{\text{tot}}}$)
17:   bs.UPDATE($\delta_{\overline{x}}, \delta_h, \delta_{\widehat{h}}, \delta_{\widetilde{h}}, \delta_{H_{w,\overline{x}}^{1/2}\widehat{x}}, \delta_{H_{w,\overline{x}}^{-1/2}\widehat{s}}$)       ▷ Algorithm 17
18:   $\ell \leftarrow \ell + 1$
19: **end procedure**
20: **procedure** MOVEANDQUERY($\beta_x, \beta_s \in \mathbb{R}, \widehat{\beta}_x, \widehat{\beta}_s \in \mathbb{R}^d, \widetilde{\beta}_x, \widetilde{\beta}_s \in \mathbb{R}^m$)
21:   bs.MOVE($\beta_x, \beta_s, \widehat{\beta}_x, \widehat{\beta}_s, \widetilde{\beta}_x, \widetilde{\beta}_s$)      ▷ Algorithm 17. Do not update $\ell$ yet
22:   $\delta_{\widetilde{x}} \leftarrow \text{QUERY}x(\epsilon_{\text{apx},x}/(2\log q + 1))$        ▷ Algorithm 16
23:   $\delta_{\widetilde{s}} \leftarrow \text{QUERY}s(\epsilon_{\text{apx},s}/(2\log q + 1))$        ▷ Algorithm 16
24:   $\widetilde{x} \leftarrow \widetilde{x} + \delta_{\widetilde{x}}, \widetilde{s} \leftarrow \widetilde{s} + \delta_{\widetilde{s}}$
25:   **return** $(\delta_{\widetilde{x}}, \delta_{\widetilde{s}})$
26: **end procedure**
27: **procedure** QUERY$x(\epsilon \in \mathbb{R})$
28:   Same as Algorithm 5, QUERY$x$.
29: **end procedure**
30: **procedure** QUERY$s(\epsilon \in \mathbb{R})$
31:   Same as Algorithm 5, QUERY$s$.
32: **end procedure**
33: **end data structure**

---

**Theorem E.6.** *Given parameters $\epsilon_{\text{apx},x}, \epsilon_{\text{apx},s} \in (0, 1), \delta_{\text{apx}} \in (0, 1), \zeta_x, \zeta_s \in \mathbb{R}$ such that*

$$\|H_{w,\overline{x}^{(\ell)}}^{1/2}x^{(\ell)} - H_{w,\overline{x}^{(\ell)}}^{1/2}x^{(\ell+1)}\|_2 \le \zeta_x, \quad \|H_{w,\overline{x}^{(\ell)}}^{-1/2}s^{(\ell)} - H_{w,\overline{x}^{(\ell)}}^{-1/2}s^{(\ell+1)}\|_2 \le \zeta_s$$

*for all $\ell \in \{0, \ldots, q-1\}$, data structure APPROXDS (Algorithm 16) supports the following operations:*

- INITIALIZE($x, s \in \mathbb{R}^{n_{\text{tot}}}, h \in \mathbb{R}^{n_{\text{tot}}}, \widehat{h} \in \mathbb{R}^{n_{\text{tot}} \times k}, \widetilde{h} \in$
  $\mathbb{R}^{n_{\text{tot}} \times m}, H_{w,\overline{x}}^{1/2}\widehat{x}, H_{w,\overline{x}}^{-1/2}\widehat{s} \in \mathbb{R}^{n_{\text{tot}}}, \beta_x, \beta_s \in \mathbb{R}, \widehat{\beta}_x, \widehat{\beta}_s \in \mathbb{R}^k, \widetilde{\beta}_x, \widetilde{\beta}_s \in \mathbb{R}^m, q \in$

$\mathbb{N}$, EXACTDS* exact, $\epsilon_{\mathrm{apx},x}, \epsilon_{\mathrm{apx},s}, \delta_{\mathrm{apx}} \in \mathbb{R}$): *Initialize the data structure in* $\widetilde{O}(n(k+m))$ *time.*

- MOVEANDQUERY($\beta_x, \beta_s \in \mathbb{R}, \widehat{\beta}_x, \widehat{\beta}_s \in \mathbb{R}^d, \widetilde{\beta}_x, \widetilde{\beta}_s \in \mathbb{R}^m$): *Update values of* $\beta_x, \beta_s, \widehat{\beta}_x, \widehat{\beta}_s, \widetilde{\beta}_x, \widetilde{\beta}_s$ *by calling* BATCHSKETCH.MOVE. *This effectively moves* $(x^{(\ell)}, s^{(\ell)})$ *to* $(x^{(\ell+1)}, s^{(\ell+1)})$ *while keeping* $\overline{x}^{(\ell)}$ *unchanged.*

  *Then return two sets* $L_x^{(\ell)}, L_s^{(\ell)} \subset [n]$ *where*
  $$L_x^{(\ell)} \supseteq \{i \in [n] : \|H_{w,\overline{x}^{(\ell)}}^{1/2} x_i^{(\ell)} - H_{w,\overline{x}^{(\ell)}}^{1/2} x_i^{(\ell+1)}\|_2 \geq \epsilon_{\mathrm{apx},x}\},$$
  $$L_s^{(\ell)} \supseteq \{i \in [n] : \|H_{w,\overline{x}^{(\ell)}}^{-1/2} s_i^{(\ell)} - H_{w,\overline{x}^{(\ell)}}^{-1/2} s_i^{(\ell+1)}\|_2 \geq \epsilon_{\mathrm{apx},s}\},$$
  *satisfying*
  $$\sum_{0 \leq \ell \leq q-1} |L_x^{(\ell)}| = \widetilde{O}(\epsilon_{\mathrm{apx},x}^{-2} \zeta_x^2 q^2),$$
  $$\sum_{0 \leq \ell \leq q-1} |L_s^{(\ell)}| = \widetilde{O}(\epsilon_{\mathrm{apx},s}^{-2} \zeta_s^2 q^2).$$

  *For every query, with probability at least* $1 - \delta_{\mathrm{apx}}/q$, *the return values are correct.*

  *Furthermore, total time cost over all queries is at most*
  $$\widetilde{O}\left((\epsilon_{\mathrm{apx},x}^{-2} \zeta_x^2 + \epsilon_{\mathrm{apx},s}^{-2} \zeta_s^2) q^2 (k+m)\right).$$

- UPDATE($\delta_{\overline{x}} \in \mathbb{R}^{n_{\mathrm{tot}}}, \delta_h \in \mathbb{R}^{n_{\mathrm{tot}}}, \delta_{\widehat{h}} \in \mathbb{R}^{n_{\mathrm{tot}} \times d}, \delta_{\widetilde{h}} \in \mathbb{R}^{n_{\mathrm{tot}} \times m}, \delta_{H_{w,\overline{x}}^{1/2} \widehat{x}}, \delta_{H_{w,\overline{x}}^{-1/2} \widehat{s}} \in \mathbb{R}^{n_{\mathrm{tot}}}$):

  *Update sketches of* $H_{w,\overline{x}^{(\ell)}}^{1/2} x^{(\ell+1)}$ *and* $H_{w,\overline{x}^{(\ell)}}^{-1/2} s^{(\ell+1)}$ *by calling* BATCHSKETCH.UPDATE. *This effectively moves* $\overline{x}^{(\ell)}$ *to* $\overline{x}^{(\ell+1)}$ *while keeping* $(x^{(\ell+1)}, s^{(\ell+1)})$ *unchanged. Then advance timestamp* $\ell$.

  *Each update costs*
  $$\widetilde{O}(\|\delta_h\|_0 + \mathrm{nnz}(\delta_{\widehat{h}}) + \mathrm{nnz}(\delta_{\widetilde{h}}) + \|H_{w,\overline{x}}^{1/2} \widehat{x}\|_0 + \|H_{w,\overline{x}}^{-1/2} \widehat{s}\|_0)$$
  *time.*

*Proof.* The proof is essentially the same as proof of (Gu & Song, 2022, Theorem 4.18). For the running time claims, we plug in Theorem E.7 when necessary. $\square$

### E.3.3 BATCHSKETCH

In this section we present the data structure BATCHSKETCH. It maintains a sketch of $H_{\overline{x}}^{1/2} x$ and $H_{\overline{x}}^{-1/2} s$. It is a variation of BATCHSKETCH in Gu & Song (2022).

**Theorem E.7.** *Data structure* BATCHSKETCH *(Algorithm 17, 18) supports the following operations:*

- INITIALIZE($\overline{x} \in \mathbb{R}^{n_{\mathrm{tot}}}, h \in \mathbb{R}^{n_{\mathrm{tot}}}, \widehat{h} \in \mathbb{R}^{n_{\mathrm{tot}} \times k}, \widetilde{h} \in \mathbb{R}^{n_{\mathrm{tot}} \times m}, H_{w,\overline{x}}^{1/2} \widehat{x}, H_{w,\overline{x}}^{-1/2} \widehat{s} \in \mathbb{R}^{n_{\mathrm{tot}}}, \beta_x, \beta_s \in \mathbb{R}, \widehat{\beta}_x, \widehat{\beta}_s \in \mathbb{R}^k, \widetilde{\beta}_x, \widetilde{\beta}_s \in \mathbb{R}^m, \delta_{\mathrm{apx}} \in \mathbb{R}$): *Initialize the data structure in* $\widetilde{O}(n(k+m))$ *time.*

- MOVE($\beta_x, \beta_s \in \mathbb{R}, \widehat{\beta}_x, \widehat{\beta}_s \in \mathbb{R}^k, \widetilde{\beta}_x, \widetilde{\beta}_s \in \mathbb{R}^m$): *Update values of* $\beta_x, \beta_s, \widehat{\beta}_x, \widehat{\beta}_s, \widetilde{\beta}_x, \widetilde{\beta}_s$ *in* $O(k+m)$ *time. This effectively moves* $(x^{(\ell)}, s^{(\ell)})$ *to* $(x^{(\ell+1)}, s^{(\ell+1)})$ *while keeping* $\overline{x}^{(\ell)}$ *unchanged.*

- UPDATE($\delta_{\overline{x}} \in \mathbb{R}^{n_{\mathrm{tot}}}, \delta_h \in \mathbb{R}^{n_{\mathrm{tot}}}, \delta_{\widehat{h}} \in \mathbb{R}^{n_{\mathrm{tot}} \times k}, \delta_{\widetilde{h}} \in \mathbb{R}^{n_{\mathrm{tot}} \times m}, \delta_{H_{w,\overline{x}}^{1/2} \widehat{x}}, \delta_{H_{w,\overline{x}}^{-1/2} \widehat{s}} \in \mathbb{R}^{n_{\mathrm{tot}}}$):

  *Update sketches of* $H_{w,\overline{x}^{(\ell)}}^{1/2} x^{(\ell+1)}$ *and* $H_{w,\overline{x}^{(\ell)}}^{-1/2} s^{(\ell+1)}$. *This effectively moves* $\overline{x}^{(\ell)}$ *to* $\overline{x}^{(\ell+1)}$ *while keeping* $(x^{(\ell+1)}, s^{(\ell+1)})$ *unchanged. Then advance timestamp* $\ell$.

  *Each update costs*
  $$\widetilde{O}(\|\delta_h\|_0 + \mathrm{nnz}(\delta_{\widehat{h}}) + \mathrm{nnz}(\delta_{\widetilde{h}}) + \|H_{w,\overline{x}}^{1/2} \widehat{x}\|_0 + \|H_{w,\overline{x}}^{-1/2} \widehat{s}\|_0).$$

---

**Algorithm 17** This is used by Algorithm 16.

1: **data structure** BATCHSKETCH ▷ Theorem E.7
2: **members**
3:     $\Phi \in \mathbb{R}^{r \times n_{\text{tot}}}$ ▷ All sketches need to share the same sketching matrix
4:     $\mathcal{S}, \chi$ partition tree
5:     $\ell \in \mathbb{N}$ ▷ Current timestamp
6:     VECTORSKETCH sketch$H_{w,\overline{x}}^{1/2}\widehat{x}$, sketch$H_{w,\overline{x}}^{-1/2}\widehat{s}$, sketch$h$, sketch$\widehat{h}$, sketch$\widetilde{h}$ ▷ Algorithm 9
7:     $\beta_x, \beta_s \in \mathbb{R}, \widehat{\beta}_x, \widehat{\beta}_s \in \mathbb{R}^d, \widetilde{\beta}_x, \widetilde{\beta}_s \in \mathbb{R}^m$
8:     $(\text{history}[t])_{t \geq 0}$ ▷ Snapshot of data at timestamp $t$. See Remark D.9.
9: **end members**
10: **procedure** INITIALIZE($\overline{x} \in \mathbb{R}^{n_{\text{tot}}}, h \in \mathbb{R}^{n_{\text{tot}}}, \widehat{h} \in \mathbb{R}^{n_{\text{tot}} \times k}, \widetilde{h} \in \mathbb{R}^{n_{\text{tot}} \times m}, H_{w,\overline{x}}^{1/2}\widehat{x}, H_{w,\overline{x}}^{-1/2}\widehat{s} \in$
    $\mathbb{R}^{n_{\text{tot}}}, \beta_x, \beta_s \in \mathbb{R}, \widehat{\beta}_x, \widehat{\beta}_s \in \mathbb{R}^d, \widetilde{\beta}_x, \widetilde{\beta}_s \in \mathbb{R}^m, \delta_{\text{apx}} \in \mathbb{R}$)
11:     Construct any valid partition tree $(\mathcal{S}, \chi)$
12:     $r \leftarrow \Theta(\log^3(n_{\text{tot}}) \log(1/\delta_{\text{apx}}))$
13:     Initialize $\Phi \in \mathbb{R}^{r \times n_{\text{tot}}}$ with iid $\mathcal{N}(0, \frac{1}{r})$
14:     $\beta_x \leftarrow \beta_x, \beta_s \leftarrow \beta_s, \widehat{\beta}_x \leftarrow \widehat{\beta}_x, \widehat{\beta}_s \leftarrow \widehat{\beta}_s, \widetilde{\beta}_x \leftarrow \widetilde{\beta}_x, \widetilde{\beta}_s \leftarrow \widetilde{\beta}_s$
15:     sketch$H_{w,\overline{x}}^{1/2}\widehat{x}$.INITIALIZE$(\mathcal{S}, \chi, \Phi, H_{w,\overline{x}}^{1/2}\widehat{x})$ ▷ Algorithm 9
16:     sketch$H_{w,\overline{x}}^{-1/2}\widehat{s}$.INITIALIZE$(\mathcal{S}, \chi, \Phi, H_{w,\overline{x}}^{-1/2}\widehat{s})$ ▷ Algorithm 9
17:     sketch$h$.INITIALIZE$(\mathcal{S}, \chi, \Phi, h)$ ▷ Algorithm 9
18:     sketch$\widehat{h}$.INITIALIZE$(\mathcal{S}, \chi, \Phi, \widehat{h})$ ▷ Algorithm 9. Here we construct one sketch for $\widehat{h}_{*,i}$ for
    every $i \in [k]$.
19:     sketch$\widetilde{h}$.INITIALIZE$(\mathcal{S}, \chi, \Phi, \widetilde{h})$ ▷ Algorithm 9. Here we construct one sketch for $\widetilde{h}_{*,i}$ for
    every $i \in [m]$.
20:     $\ell \leftarrow 0$
21:     Make snapshot history$[\ell]$ ▷ Remark D.9
22: **end procedure**
23: **procedure** MOVE($\beta_x, \beta_s \in \mathbb{R}, \widehat{\beta}_x, \widehat{\beta}_s \in \mathbb{R}^k, \widetilde{\beta}_x, \widetilde{\beta}_s \in \mathbb{R}^m$)
24:     $\beta_x \leftarrow \beta_x, \beta_s \leftarrow \beta_s, \widehat{\beta}_x \leftarrow \widehat{\beta}_x, \widehat{\beta}_s \leftarrow \widehat{\beta}_s, \widetilde{\beta}_x \leftarrow \widetilde{\beta}_x, \widetilde{\beta}_s \leftarrow \widetilde{\beta}_s$ ▷ Do not update $\ell$ yet
25: **end procedure**
26: **procedure** UPDATE($\delta_{\overline{x}} \in \mathbb{R}^{n_{\text{tot}}}, \delta_h \in \mathbb{R}^{n_{\text{tot}}}, \delta_{\widehat{h}} \in \mathbb{R}^{n_{\text{tot}} \times k}, \delta_{\widetilde{h}} \in \mathbb{R}^{n_{\text{tot}} \times m}, \delta_{H_{w,\overline{x}}^{1/2}\widehat{x}}, \delta_{H_{w,\overline{x}}^{-1/2}\widehat{s}} \in$
    $\mathbb{R}^{n_{\text{tot}}}$)
27:     sketch$H_{w,\overline{x}}^{1/2}\widehat{x}$.UPDATE$(\delta_{H_{w,\overline{x}}^{1/2}\widehat{x}})$ ▷ Algorithm 9
28:     sketch$H_{w,\overline{x}}^{-1/2}\widehat{s}$.UPDATE$(\delta_{H_{w,\overline{x}}^{-1/2}\widehat{s}})$ ▷ Algorithm 9
29:     sketch$h$.UPDATE$(\delta_h)$ ▷ Algorithm 9
30:     sketch$\widehat{h}$.UPDATE$(\delta_{\widehat{h}})$ ▷ Algorithm 9
31:     sketch$\widetilde{h}$.UPDATE$(\delta_{\widetilde{h}})$ ▷ Algorithm 9
32:     $\ell \leftarrow \ell + 1$
33:     Make snapshot history$[\ell]$ ▷ Remark D.9
34: **end procedure**
35: **end data structure**

---

- QUERY$x(\ell' \in \mathbb{N}, \epsilon \in \mathbb{R})$: *Given timestamp $\ell'$, return a set $S \subseteq [n]$ where*

$$S \supseteq \{i \in [n] : \|H_{w,\overline{x}^{(\ell')}}^{1/2} x_i^{(\ell')} - H_{w,\overline{x}^{(\ell)}}^{1/2} x_i^{(\ell+1)}\|_2 \geq \epsilon\},$$

*and*

$$|S| = O(\epsilon^{-2}(\ell - \ell' + 1) \sum_{\ell' \leq t \leq \ell} \|H_{w,\overline{x}^{(t)}}^{1/2} x^{(t)} - H_{w,\overline{x}^{(t)}}^{1/2} x^{(t+1)}\|_2^2 + \sum_{\ell' \leq t \leq \ell - 1} \|\overline{x}^{(t)} - \overline{x}^{(t+1)}\|_{2,0})$$

*where $\ell$ is the current timestamp.*

---

**Algorithm 18** BATCHSKETCH Algorithm 17 continued. This is used by Algorithm 16.

1: **data structure** BATCHSKETCH                                              ▷ Theorem E.7
2: **private:**
3: **procedure** QUERY$x$SKETCH$(v \in \mathcal{S})$        ▷ Return the value of $\Phi_{\chi(v)}(H_{w,\overline{x}}^{1/2}x)_{\chi(v)}$
4:     **return** sketch$H_{w,\overline{x}}^{1/2}\widehat{x}$.QUERY$(v)$ + sketch$h$.QUERY$(v) \cdot \beta_x$ + sketch$\widehat{h}$.QUERY$(v) \cdot \widehat{\beta}_x$ +
   sketch$\widetilde{h}$.QUERY$(v) \cdot \widetilde{\beta}_x$                                ▷ Algorithm 9
5: **end procedure**
6: **procedure** QUERY$s$SKETCH$(v \in \mathcal{S})$        ▷ Return the value of $\Phi_{\chi(v)}(H_{w,\overline{x}}^{-1/2}s)_{\chi(v)}$
7:     **return** sketch$H_{w,\overline{x}}^{-1/2}\widehat{s}$.QUERY$(v)$ + sketch$h$.QUERY$(v) \cdot \beta_s$ + sketch$\widehat{h}$.QUERY$(v) \cdot \widehat{\beta}_s$ +
   sketch$\widetilde{h}$.QUERY$(v) \cdot \widetilde{\beta}_s$                                ▷ Algorithm 9
8: **end procedure**
9: **public:**
10: **procedure** QUERY$x(\ell' \in \mathbb{N}, \epsilon \in \mathbb{R})$
11:     Same as Algorithm 7, QUERY$x$, using QUERY$x$SKETCH defined here instead of the one in
    Algorithm 7.
12: **end procedure**
13: **procedure** QUERY$s(\ell' \in \mathbb{N}, \epsilon \in \mathbb{R})$
14:     Same as Algorithm 7, QUERY$s$, using QUERY$s$SKETCH defined here instead of the one in
    Algorithm 7.
15: **end procedure**
16: **end structure**

---

*For every query, with probability at least $1 - \delta$, the return values are correct, and costs at most*

$$\widetilde{O}((k + m) \cdot (\epsilon^{-2}(\ell - \ell' + 1) \sum_{\ell' \leq t \leq \ell} \|H_{\overline{x}^{(t)}}^{1/2}x^{(t)} - H_{\overline{x}^{(t)}}^{1/2}x^{(t+1)}\|_2^2 + \sum_{\ell' \leq t \leq \ell-1} \|\overline{x}^{(t)} - \overline{x}^{(t+1)}\|_{2,0}))$$

*running time.*

- QUERY$s(\ell' \in \mathbb{N}, \epsilon \in \mathbb{R})$: *Given timestamp $\ell'$, return a set $S \subseteq [n]$ where*

$$S \supseteq \{i \in [n] : \|H_{w,\overline{x}^{(\ell')}}^{-1/2}s_i^{(\ell')} - H_{w,\overline{x}^{(\ell)}}^{-1/2}s_i^{(\ell+1)}\|_2 \geq \epsilon\}$$

*and*

$$|S| = O(\epsilon^{-2}(\ell - \ell' + 1) \sum_{\ell' \leq t \leq \ell} \|H_{w,\overline{x}^{(t)}}^{-1/2}s^{(t)} - H_{w,\overline{x}^{(t)}}^{-1/2}s^{(t+1)}\|_2^2 + \sum_{\ell' \leq t \leq \ell-1} \|\overline{x}^{(t)} - \overline{x}^{(t+1)}\|_{2,0})$$

*where $\ell$ is the current timestamp.*

*For every query, with probability at least $1 - \delta$, the return values are correct, and costs at most*

$$\widetilde{O}((k + m) \cdot (\epsilon^{-2}(\ell - \ell' + 1) \sum_{\ell' \leq t \leq \ell} \|H_{\overline{x}^{(t)}}^{1/2}s^{(t)} - H_{\overline{x}^{(t)}}^{1/2}x^{(t+1)}\|_2^2 + \sum_{\ell' \leq t \leq \ell-1} \|\overline{x}^{(t)} - \overline{x}^{(t+1)}\|_{2,0}))$$

*running time.*

*Proof.* The proof is essentially the same as proof of (Gu & Song, 2022, Theorem 4.21).  □

### E.4 ANALYSIS OF CENTRALPATHMAINTENANCE

**Lemma E.8** (Correctness of CENTRALPATHMAINTENANCE). *Algorithm 13 implicitly maintains the primal-dual solution pair $(x, s)$ via representation Eq. (12)(13). It also explicitly maintains $(\overline{x}, \overline{s}) \in \mathbb{R}^{n_{\text{tot}}} \times \mathbb{R}^{n_{\text{tot}}}$ such that $\|\overline{x}_i - x_i\|_{\overline{x}_i} \leq \overline{\epsilon}$ and $\|\overline{s}_i - s_i\|_{\overline{x}_i}^* \leq t\overline{\epsilon}w_i$ for all $i \in [n]$ with probability at least $0.9$.*

*Proof.* Same as proof of Lemma D.13.  □

**Lemma E.9.** *We bound the running time of* CENTRALPATHMAINTENANCE *as following.*

- CENTRALPATHMAINTENANCE.INITIALIZE *takes* $\widetilde{O}(n(k^{\omega-1} + m^{\omega-1}))$ *time.*

- *If* CENTRALPATHMAINTENANCE.MULTIPLYANDMOVE *is called $N$ times, then it has total running time*

$$\widetilde{O}((Nn^{-1/2} + \log(t_{\max}/t_{\min})) \cdot n(k+m)^{(\omega+1)/2}).$$

- CENTRALPATHMAINTENANCE.OUTPUT *takes* $\widetilde{O}(n(k+m))$ *time.*

*Proof.* INITIALIZE part: By Theorem E.3 and E.6.

OUTPUT part: By Theorem E.3.

MULTIPLYANDMOVE part: Between two restarts, the total size of $|L_x|$ returned by approx.QUERY is bounded by $\widetilde{O}(q^2 \zeta_x^2 / \epsilon_{\mathrm{apx},x}^2)$ by Theorem E.6. By plugging in $\zeta_x = 2\alpha$, $\epsilon_{\mathrm{apx},x} = \bar{\epsilon}$, we have $\sum_{\ell \in [q]} |L_x^{(\ell)}| = \widetilde{O}(q^2)$. Similarly, for $s$ we have $\sum_{\ell \in [q]} |L_s^{(\ell)}| = \widetilde{O}(q^2)$.

**Update time:** By Theorem E.3 and E.6, in a sequence of $q$ updates, total cost for update is $\widetilde{O}(q^2(k^2 + m^2))$. So the amortized update cost per iteration is $\widetilde{O}(q(k^2 + m^2))$. The total update cost is

$$\text{number of iterations} \cdot \text{time per iteration} = \widetilde{O}(Nq(k^2 + m^2)).$$

**Init/restart time:** We restart the data structure whenever $K > q$ or $|\bar{t} - t| > \bar{t}\epsilon_t$, so there are $O(N/q + \log(t_{\max}/t_{\min})\epsilon_t^{-1})$ restarts in total. By Theorem E.3 and E.6, time cost per restart is $\widetilde{O}(n(k^{\omega-1} + m^{\omega-1}))$. So the total initialization time is

$$\text{number of restarts} \cdot \text{time per restart} = \widetilde{O}((N/q + \log(t_{\max}/t_{\min})\epsilon_t^{-1}) \cdot n(k^{\omega-1} + m^{\omega-1})).$$

**Combine everything:** Overall running time is

$$\widetilde{O}(Nq(k^2 + m^2) + (N/q + \log(t_{\max}/t_{\min})\epsilon_t^{-1}) \cdot n(k^{\omega-1} + m^{\omega-1})).$$

Taking $\epsilon_t = \frac{1}{2}\bar{\epsilon}$, the optimal choice for $q$ is

$$q = n^{1/2}(k^2 + m^2)^{-1/2}(k^{\omega-1} + m^{\omega-1})^{1/2},$$

achieving overall running time

$$\widetilde{O}((Nn^{-1/2} + \log(t_{\max}/t_{\min})) \cdot n(k^2 + m^2)^{1/2}(k^{\omega-1} + m^{\omega-1})^{1/2})$$
$$= \widetilde{O}((Nn^{-1/2} + \log(t_{\max}/t_{\min})) \cdot n(k+m)^{(\omega+1)/2}). \qquad \square$$

*Proof of Theorem E.2.* Combining Lemma E.8 and E.9. $\qquad \square$

### E.5 PROOF OF MAIN STATEMENT

*Proof of Theorem E.1.* Use CENTRALPATHMAINTENANCE (Algorithm 13) as the maintenance data structure in Algorithm 20. Combining Theorem E.2 and Theorem F.1 finishes the proof. $\qquad \square$

## F ROBUST IPM ANALYSIS

In this section we present a robust IPM algorithm for quadratic programming. The algorithm is a modification of previous robust IPM algorithms for linear programming Lee et al. (2019); Lee & Vempala (2021).

Convention: Variables are in $n$ blocks of dimension $n_i$ ($i \in [n]$). Total dimension is $n_{\text{tot}} = \sum_{i \in [n]} n_i$. We write $x = (x_1, \ldots, x_n) \in \mathbb{R}^{n_{\text{tot}}}$ where $x_i \in \mathbb{R}^{n_i}$. We consider programs of the following form:

$$\min_{x \in \mathbb{R}^n} \quad \frac{1}{2} x^\top Q x + c^\top x \tag{16}$$
$$\text{s.t.} \quad Ax = b$$
$$x_i \in \mathcal{K}_i \qquad \forall i \in [n]$$

where $Q \in \mathcal{S}^{n_{\text{tot}}}$, $c \in \mathbb{R}^{n_{\text{tot}}}$, $A \in \mathbb{R}^{m \times n_{\text{tot}}}$, $b \in \mathbb{R}^m$, $\mathcal{K}_i \subset \mathbb{R}^{n_i}$ is a convex set. Let $\mathcal{K} = \prod_{i \in [n]} \mathcal{K}_i$.

**Theorem F.1.** *Consider the convex program* (16). *Let $\phi_i : \mathcal{K}_i \to \mathbb{R}$ be a $\nu_i$-self-concordant barrier for all $i \in [n]$. Suppose the program satisfies the following properties:*

- *Inner radius $r$: There exists $z \in \mathbb{R}^{n_{\text{tot}}}$ such that $Az = b$ and $B(z, r) \in \mathcal{K}$.*

- *Outer radius $R$: $\mathcal{K} \subseteq B(0, R)$ where $0 \in \mathbb{R}^{n_{\text{tot}}}$.*

- *Lipschitz constant $L$: $\|Q\|_{2 \to 2} \leq L$, $\|c\|_2 \leq L$.*

*Let $(w_i)_{i \in [n]} \in \mathbb{R}^n_{\geq 1}$ and $\kappa = \sum_{i \in [n]} w_i \nu_i$. For any $0 < \epsilon \leq \frac{1}{2}$, Algorithm 19 outputs an approximate solution $x$ in $O(\sqrt{\kappa} \log n \log \frac{n \kappa R}{\epsilon r})$ steps, satisfying*

$$\frac{1}{2} x^\top Q x + c^\top x \leq \min_{Ax = b, x \in \mathcal{K}} \left( \frac{1}{2} x^\top Q x + c^\top x \right) + \epsilon L R (R + 1),$$
$$\|Ax - b\|_1 \leq 3\epsilon(R \|A\|_1 + \|b\|_1),$$
$$x \in \mathcal{K}.$$

---

**Algorithm 19** Our main algorithm
---

1: **procedure** ROBUSTQPIPM($Q \in \mathcal{S}^{n_{\text{tot}}}, c \in \mathbb{R}^{n_{\text{tot}}}, A \in \mathbb{R}^{m \times n_{\text{tot}}}, b \in \mathbb{R}^m, (\phi_i : \mathcal{K}_i \to \mathbb{R})_{i \in [n]}, w \in \mathbb{R}^n$)
2:     /* Initial point reduction */
3:     $\rho \leftarrow LR(R+1)$, $x^{(0)} \leftarrow \arg\min_x \sum_{i \in [n]} w_i \phi_i(x_i)$, $s^{(0)} \leftarrow \epsilon \rho(c + Qx^{(0)})$
4:     $\overline{x} \leftarrow \begin{bmatrix} x^{(0)} \\ 1 \end{bmatrix}$, $\overline{s} \leftarrow \begin{bmatrix} s^{(0)} \\ 1 \end{bmatrix}$, $\overline{Q} \leftarrow \begin{bmatrix} \epsilon \rho Q & 0 \\ 0 & 0 \end{bmatrix}$, $\overline{A} \leftarrow \begin{bmatrix} A \mid b - Ax^{(0)} \end{bmatrix}$
5:     $\overline{w} \leftarrow \begin{bmatrix} w \\ 1 \end{bmatrix}$, $\overline{\phi}_i = \phi_i \forall i \in [n]$, $\overline{\phi}_{n+1}(x) := -\log x - \log(2 - x)$
6:     $(x, s) \leftarrow$ CENTERING($\overline{Q}, \overline{A}, (\overline{\phi}_i)_{i \in [n+1]}, \overline{w}, \overline{x}, \overline{s}, t_{\text{start}} = 1, t_{\text{end}} = \frac{\epsilon^2}{4\kappa}$)
7:     **return** $(x_{1:n}, s_{1:n})$
8: **end procedure**

---

### F.1 PRELIMINARIES

Previous works on linear programming (e.g. Lee et al. (2019), Lee & Vempala (2021)) use the following path:

$$s/t + \nabla \phi_w(x) = \mu,$$
$$Ax = b,$$
$$A^\top y + s = c$$

where $\phi_w(x) := \sum_{i=1}^n w_i \phi_i(x_i)$.

For quadratic programming, we modify the above central path as following:

$$s/t + \nabla \phi_w(x) = \mu,$$
$$Ax = b,$$
$$-Qx + A^\top y + s = c.$$

We make the following definitions.

---

**Algorithm 20** Subroutine used by Algorithm 19

---

1: **procedure** CENTERING($Q \in \mathcal{S}^{n_{\text{tot}}}, A \in \mathbb{R}^{m \times n_{\text{tot}}}, (\phi_i : \mathcal{K}_i \to \mathbb{R})_{i \in [n]}, w \in \mathbb{R}^n, x \in \mathbb{R}^{n_{\text{tot}}}, s \in \mathbb{R}^{n_{\text{tot}}}, t_{\text{start}} \in \mathbb{R}_{>0}, t_{\text{end}} \in \mathbb{R}_{>0}$)
2:     /* Parameters */
3:     $\lambda = 64 \log(256n \sum_{i \in [n]} w_i), \overline{\epsilon} = \frac{1}{1440}\lambda, \alpha = \frac{\overline{\epsilon}}{2}$
4:     $\epsilon_t = \frac{\overline{\epsilon}}{4}(\min_{i \in [n]} \frac{w_i}{w_i + \nu_i}), h = \frac{\alpha}{64\sqrt{\kappa}}$
5:     /* Definitions */
6:     $\phi_w(x) := \sum_{i \in [n]} w_i \phi_i(x_i)$
7:     $\mu_i(x, s, t) := s/t + w_i \nabla \phi_i(x_i), \forall i \in [n]$       ▷ Eq. (17)
8:     $\gamma_i(x, s, t) \leftarrow \|\mu_i^t(x, s)\|_{x_i}^*, \forall i \in [n]$       ▷ Eq. (18),
9:     $c_i(x, s, t) := \frac{\sinh(\frac{\lambda}{w_i}\gamma_i(x,s,t))}{\gamma_i(x,s,t)\sqrt{\sum_{j \in [n]} w_j^{-1} \cosh^2(\frac{\lambda}{w_j}\gamma_j(x,s,t))}}, \forall i \in [n]$       ▷ Eq. (22)
10:     $H_{w,x} := \nabla^2 \phi_w(x)$       ▷ Eq. (24)
11:     $B_{w,x,t} := Q + tH_{w,x}$       ▷ Eq. (25)
12:     $P_{w,x,t} := B_{w,x,t}^{-1/2} A^\top (AB_{w,x,t}^{-1} A^\top)^{-1} AB_{w,x,t}^{-1/2}$       ▷ Eq. (26)
13:     /* Main loop */
14:     $\overline{t} \leftarrow t \leftarrow t_{\text{start}}, \overline{x} \leftarrow x, \overline{s} \leftarrow s$
15:     **while** $t > t_{\text{end}}$ **do**
16:         Maintain $\overline{x}, \overline{s}, \overline{t}$ such that $\|\overline{x}_i - x_i\|_{\overline{x}_i} \leq \overline{\epsilon}, \|\overline{s}_i - s_i\|_{\overline{x}_i}^* \leq t\overline{\epsilon}w_i$ and $|\overline{t} - t| \leq \epsilon_t \overline{t}$
17:         $\delta_{\mu,i} \leftarrow -\alpha \cdot c_i(\overline{x}, \overline{s}, \overline{t}) \cdot \mu_i(\overline{x}, \overline{s}, \overline{t}), \forall i \in [n]$       ▷ Eq. (21)
18:         Pick $\delta_x$ and $\delta_s$ such that $A\delta_x = 0, \delta_s - Q\delta_x \in \text{Range}(A^\top)$ and

$$\|\delta_x - \overline{t}B_{w,\overline{x},\overline{t}}^{-1/2}(I - P_{w,\overline{x},\overline{t}})B_{w,\overline{x},\overline{t}}^{-1/2}\delta_\mu\|_{w,\overline{x}} \leq \overline{\epsilon}\alpha,$$

$$\|\overline{t}^{-1}\delta_s - (\delta_\mu - \overline{t}H_{w,\overline{x}}B_{w,\overline{x},\overline{t}}^{-1/2}(I - P_{w,\overline{x},\overline{t}})B_{w,\overline{x},\overline{t}}^{-1/2}\delta_\mu)\|_{w,\overline{x}}^* \leq \overline{\epsilon}\alpha.$$

19:         $t \leftarrow \max\{(1 - h)t, t_{\text{end}}\}, x \leftarrow x + \delta_x, s \leftarrow s + \delta_s$
20:     **end while**
21:     **return** $(x, s)$
22: **end procedure**

---

**Definition F.2.** For each $i \in [n]$, we define the $i$-th coordinate error

$$\mu_i(x, s, t) := \frac{s_i}{t} + w_i \nabla \phi_i(x_i) \tag{17}$$

We define $\mu_i$'s norm as

$$\gamma_i(x, s, t) := \|\mu_i(x, s, t)\|_{x_i}^*. \tag{18}$$

We define the soft-max function by

$$\Psi_\lambda(r) := \sum_{i=1}^m \cosh(\lambda \frac{r_i}{w_i}) \tag{19}$$

for some $\lambda > 0$ and the potential function is the soft-max of the norm of the error of each coordinate

$$\Phi(x, s, t) = \Psi_\lambda(\gamma(x, s, t)) \tag{20}$$

We choose the step direction $\delta_\mu$ as

$$\delta_{\mu,i} := -\alpha \cdot c_i(x, s, t) \cdot \mu_i(x, s, t) \tag{21}$$

where

$$c_i(x, s, t) := \frac{\sinh(\frac{\lambda}{w_i}\gamma_i(x, s, t))}{\gamma_i(x, s, t)\sqrt{\sum_{j \in [n]} w_j^{-1} \cosh^2(\frac{\lambda}{w_j}\gamma_j(x, s, t))}} \tag{22}$$

We define induced norms as following. Note that we include the weight vector $w$ in the subscript to avoid confusion.

**Definition F.3.** For each block $\mathcal{K}_i$, we define

$$\|v\|_{x_i} := \|v\|_{\nabla^2 \phi_i(x_i)},$$
$$\|v\|_{x_i}^* := \|v\|_{(\nabla^2 \phi_i(x_i))^{-1}}$$

for $v \in \mathbb{R}^{n_i}$.

For the whole domain $\mathcal{K} = \prod_{i=1}^n \mathcal{K}_i$, we define

$$\|v\|_{w,x} := \|v\|_{\nabla^2 \phi_w(x)} = \left(\sum_{i=1}^n w_i \|v_i\|_{x_i}^2\right)^{1/2},$$

$$\|v\|_{w,x}^* := \|v\|_{(\nabla^2 \phi_w(x))^{-1}} = \left(\sum_{i=1}^n w_i^{-1} (\|v_i\|_{x_i}^*)^2\right)^{1/2}$$

for $v \in \mathbb{R}^{n_{\text{tot}}}$.

The Hessian matrices of the barrier functions appear a lot in the computation.

**Definition F.4.** We define matrices $H_{x,i} \in \mathbb{R}^{n_i \times n_i}$ and $H_{w,x} \in \mathbb{R}^{n_{\text{tot}} \times n_{\text{tot}}}$ as

$$H_{x,i} := \nabla^2 \phi_i(x_i), \tag{23}$$
$$H_{w,x} := \nabla^2 \phi_w(x). \tag{24}$$

From the definition, we see that

$$H_{w,x,(i,i)} = w_i H_{x,i}.$$

The following equations are immediate from definition.

**Claim F.5.** *Let $H_{w,x} \in \mathbb{R}^{n_{\text{tot}} \times n_{\text{tot}}}$ be defined as Definition F.4. For $v \in \mathbb{R}^{n_{\text{tot}}}$, we have*

$$\|v\|_{w,x} = \|H_{w,x}^{1/2} v\|_2,$$
$$\|v\|_{w,x}^* = \|H_{w,x}^{-1/2} v\|_2.$$

**Claim F.6.** *For each $i \in [n]$, let $H_{x,i}$ be defined as Definition F.4. For $v \in \mathbb{R}^{n_i}$, $i \in [n]$, we have*

$$\|v\|_{x_i} = \|H_{x,i}^{1/2} v\|_2,$$
$$\|v\|_{x_i}^* = \|H_{x,i}^{-1/2} v\|_2.$$

We define matrices $B$ and $P$ used in the algorithm.

**Definition F.7.** Let $A, Q$ denote two fixed matrices. Let $H_{w,x} \in \mathbb{R}^{n_{\text{tot}} \times n_{\text{tot}}}$ be defined as Definition F.4. We define matrix $B_{w,x,t} \in \mathbb{R}^{n_{\text{tot}} \times n_{\text{tot}}}$ as

$$B_{w,x,t} := Q + t \cdot H_{w,x} \tag{25}$$

We define projection matrix $P_{w,x,t} \in \mathbb{R}^{n_{\text{tot}} \times n_{\text{tot}}}$ as

$$P_{w,x,t} \leftarrow B_{w,x,t}^{-1/2} A^\top (A B_{w,x,t}^{-1} A^\top)^{-1} A B_{w,x,t}^{-1/2}. \tag{26}$$

## F.2 DERIVING THE CENTRAL PATH STEP

In this section we explain how to derive the central path step.

We follow the central path

$$s/t + \nabla \phi_w(x) = \mu$$

$$Ax = b$$
$$-Qx + A^\top y + s = c$$

We perform gradient descent on $\mu$ with step $\delta_\mu$. Then Newton step gives

$$\frac{1}{t}\delta_s + \nabla^2 \phi_w(x)\delta_x = \delta_\mu \tag{27}$$

$$A\delta_x = 0 \tag{28}$$

$$-Q\delta_x + A^\top \delta_y + \delta_s = 0 \tag{29}$$

where $\delta_x$ (resp. $\delta_y$, $\delta_s$) is the step taken by $x$ (resp. $y$, $s$).

For simplicity, we define $H \in \mathbb{R}^{n_{\text{tot}} \times n_{\text{tot}}}$ to represent $\nabla^2 \phi_w(x)$.[8]

From Eq. (27) we get

$$\delta_s = t\delta_\mu - tH\delta_x. \tag{30}$$

Plug the above equation into Eq. (29) we get

$$-Q\delta_x + A^\top \delta_y + t\delta_\mu - tH\delta_x = 0. \tag{31}$$

Let $B = Q + tH$, multiply by $AB^{-1}$ we get

$$-A\delta_x + AB^{-1}A^\top \delta_y + tAB^{-1}\delta_\mu = 0.$$

Using Eq. (28) we get

$$AB^{-1}A^\top \delta_y + tAB^{-1}\delta_\mu = 0.$$

Solve for $\delta_y$ (assuming that $AB^{-1}A$ is invertible), we get

$$\delta_y = -t(AB^{-1}A^\top)^{-1}AB^{-1}\delta_\mu.$$

Plug into Eq. (31) we get

$$-B\delta_x - tA^\top(AB^{-1}A^\top)^{-1}AB^{-1}\delta_\mu + t\delta_\mu = 0.$$

Solve for $\delta_x$ we get

$$\delta_x = tB^{-1}\delta_\mu - tB^{-1}A^\top(AB^{-1}A^\top)^{-1}AB^{-1}\delta_\mu$$
$$= tB^{-1/2}(I - P)B^{-1/2}\delta_\mu$$

where $P = B^{-1/2}A^\top(AB^{-1}A^\top)^{-1}AB^{-1/2}$ is the projection matrix. Solve for $\delta_s$ in Eq. (30) we get

$$\delta_s = t\delta_\mu - t^2 HB^{-1/2}(I - P)B^{-1/2}\delta_\mu.$$

In summary, we have

$$\delta_x = tB^{-1/2}(I - P)B^{-1/2}\delta_\mu,$$
$$\delta_y = -t(AB^{-1}A^\top)^{-1}AB^{-1}\delta_\mu,$$
$$\delta_s = t\delta_\mu - t^2 HB^{-1/2}(I - P)B^{-1/2}\delta_\mu,$$
$$P = B^{-1/2}A^\top(AB^{-1}A^\top)^{-1}AB^{-1/2}.$$

These equations will guide the design of our actual algorithm.

---

[8]In this section, and in this section only, we omit the subscript in $H$, $B$, $P$ for simplicity.

### F.3 BOUNDING MOVEMENT OF POTENTIAL FUNCTION

The goal of this section is to bound the movement of potential function during the robust IPM algorithm.

In robust IPM, we do not need to follow the ideal central path exactly over the entire algorithm. Instead, we only use an approximate version. For convenience of analysis we state two assumptions (see Algorithm 20, Line 18).

**Assumption F.8.** We make the following assumptions on $\delta_x \in \mathbb{R}^{n_{\text{tot}}}$ and $\delta_s \in \mathbb{R}^{n_{\text{tot}}}$.

$$\|\delta_x - \bar{t}B_{w,\overline{x},\bar{t}}^{-1/2}(I - P_{w,\overline{x},\bar{t}})B_{w,\overline{x},\bar{t}}^{-1/2}\delta_\mu\|_{w,\overline{x}} \leq \bar{\epsilon}\alpha,$$

$$\|\bar{t}^{-1}\delta_s - (\delta_\mu - \bar{t}H_{w,\overline{x}}B_{w,\overline{x},\bar{t}}^{-1/2}(I - P_{w,\overline{x},\bar{t}})B_{w,\overline{x},\bar{t}}^{-1/2}\delta_\mu)\|_{w,\overline{x}}^* \leq \bar{\epsilon}\alpha.$$

The following lemma bounds the movement of potential function $\Psi$ assuming bound on $\delta_\gamma$.

**Lemma F.9** ((Ye, 2020, Lemma A.5)). *For any $r \in \mathbb{R}^{n_{\text{tot}}}$, and $w \in \mathbb{R}_{\geq 1}^{n_{\text{tot}}}$. Let $\alpha$ and $\lambda$ denote the parameters that are satisfying $0 \leq \alpha \leq \frac{1}{8\lambda}$.*

*Let $\epsilon_r \in \mathbb{R}^{n_{\text{tot}}}$ denote a vector satisfying*

$$(\sum_{i=1}^n w_i^{-1}\epsilon_{r,i}^2)^{1/2} \leq \alpha/8.$$

*Suppose that vector $\overline{r} \in \mathbb{R}^{n_{\text{tot}}}$ is satisfying the following property*

$$|r_i - \overline{r}_i| \leq \frac{w_i}{8\lambda}, \quad \forall i \in [n]$$

*We define vector $\delta_r \in \mathbb{R}^{n_{\text{tot}}}$ as follows:*

$$\delta_{r,i} := \frac{-\alpha \cdot \sinh(\frac{\lambda}{w_i}\overline{r}_i)}{\sqrt{\sum_{j=1}^n w_j^{-1}\cosh^2(\frac{\lambda}{w_j}\overline{r}_j)}} + \epsilon_{r,i}.$$

*Then, we have that*

$$\Psi_\lambda(r + \delta_r) \leq \Psi_\lambda(r) - \frac{\alpha\lambda}{2}(\sum_{i=1}^n w_i^{-1}\cosh^2(\lambda\frac{r_i}{w_i}))^{1/2} + \alpha\lambda(\sum_{i=1}^n w_i^{-1})^{1/2}$$

The following lemma bounds the norm of $\delta_\mu$.

**Lemma F.10** (Bounding norm of $\delta_\mu$).

$$\|\delta_\mu(\overline{x}, \overline{s}, \overline{t})\|_{w,\overline{x}}^* \leq \alpha.$$

*Proof.*

$$(\|\delta_\mu(\overline{x}, \overline{s}, \overline{t})\|_{w,\overline{x}}^*)^2 = \sum_{i=1}^n w_i^{-1}(\|\delta_{\mu,i}(\overline{x}, \overline{s}, \overline{t})\|_{\overline{x}_i}^*)^2$$

$$= \alpha^2 \sum_{i\in[n]} w_i^{-1}c_i^2(\overline{x}, \overline{s}, \overline{t}) \cdot \|\mu_i(\overline{x}, \overline{s}, \overline{t})\|_{\overline{x}_i}^2$$

$$= \alpha^2 \sum_{i\in[n]} w_i^{-1}c_i^2(\overline{x}, \overline{s}, \overline{t}) \cdot \|H_{\overline{x},i}^{-1/2}\mu_i(\overline{x}, \overline{s}, \overline{t})\|_2^2$$

$$= \alpha^2 \sum_{i\in[n]} w_i^{-1}c_i^2(\overline{x}, \overline{s}, \overline{t}) \cdot \gamma_i^2(\overline{x}, \overline{s}, \overline{t})$$

$$= \alpha^2 \sum_{i\in[n]} \frac{w_i^{-1}\sinh^2(\frac{\lambda}{w_i}\gamma_i(\overline{x}, \overline{s}, \overline{t}))}{\gamma_i^2(\overline{x}, \overline{s}, \overline{t}) \cdot \sum_{j\in[n]} w_j^{-1}\cosh^2(\frac{\lambda}{w_j}\gamma_j(\overline{x}, \overline{s}, \overline{t}))} \cdot \gamma_i^2(\overline{x}, \overline{s}, \overline{t})$$

$$= \alpha^2 \frac{\sum_{j \in [n]} w_j^{-1} \sinh^2(\frac{\lambda}{w_j} \gamma_j(\overline{x}, \overline{s}, \overline{t}))}{\sum_{j \in [n]} w_j^{-1} \cosh^2(\frac{\lambda}{w_j} \gamma_j(\overline{x}, \overline{s}, \overline{t}))}$$

$$\leq \alpha^2.$$

where the first step follows from Definition F.3, the second step follows from $\delta_{\mu,i}(\overline{x}, \overline{s}, \overline{t}) = -\alpha \cdot c_i(\overline{x}, \overline{s}, \overline{t}) \cdot \mu_i(\overline{x}, \overline{s}, \overline{t})$, the third step follows from norm of $\overline{x}_i$ (see Definition F.3), the forth step follows from $\gamma_i(\overline{x}, \overline{s}, \overline{t}) = \|H_{\overline{x},i}^{-1/2} \mu_i(\overline{x}, \overline{s}, \overline{t})\|_2$ (see Eq. (18)), the fifth step follows from $c_i(\overline{x}, \overline{s}, \overline{t})^2 = \frac{\sinh^2(\frac{\lambda}{w_i} \gamma_i(\overline{x}, \overline{s}, \overline{t}))}{\gamma_i^2(\overline{x}, \overline{s}, \overline{t}) \sum_{j \in [n]} w_j^{-1} \cosh^2(\frac{\lambda}{w_j} \gamma_j(\overline{x}, \overline{s}, \overline{t}))}$ (see Eq. (22)), the sixth step follows from canceling the term $\gamma_i^2(\overline{x}, \overline{s}, \overline{t})$, and the last step follows from $\cosh^2(x) \geq \sinh^2(x)$ for all $x$. $\square$

The following lemma bounds the norm of $\delta_x$ and $\delta_s$.

**Lemma F.11.** *For each $i \in [n]$, we define $\alpha_i := \|\delta_{x,i}\|_{\overline{x}_i}$. Then, we have*

$$\|\delta_x\|_{w,\overline{x}} = (\sum_{i \in [n]} w_i \alpha_i^2)^{1/2} \leq \frac{9}{8}\alpha. \tag{32}$$

*In particular, we have $\alpha_i \leq \frac{9}{8}\alpha$. Similarly, for $\delta_s$, we have*

$$\|\delta_s\|_{w,\overline{x}}^* = \sqrt{\sum_{i \in [n]} w_i^{-1}(\|\delta_{s,i}\|_{\overline{x}_i}^*)^2} \leq \frac{17}{8}\alpha \cdot t. \tag{33}$$

*Proof.* For $\delta_x$, we have

$$\|\delta_x\|_{w,\overline{x}} \leq \|\overline{t} H_{w,\overline{x}}^{1/2} B_{w,\overline{x},\overline{t}}^{-1/2}(I - P_{w,\overline{x},\overline{t}}) B_{w,\overline{x},\overline{t}}^{-1/2} \delta_\mu\|_2 + \overline{\epsilon}\alpha$$

$$\leq \|\overline{t}^{1/2}(I - P_{w,\overline{x},\overline{t}}) B_{w,\overline{x},\overline{t}}^{-1/2} \delta_\mu\|_2 + \overline{\epsilon}\alpha$$

$$\leq \|\overline{t}^{1/2} B_{w,\overline{x},\overline{t}}^{-1/2} \delta_\mu\|_2 + \overline{\epsilon}\alpha$$

$$\leq \|H_{w,\overline{x}}^{-1/2} \delta_\mu\|_2 + \overline{\epsilon}\alpha$$

$$\leq \alpha + \overline{\epsilon}\alpha$$

$$\leq \frac{9}{8}\alpha.$$

First step follows from Assumption F.8. Second step is because $\overline{t} H_{w,\overline{x}} \preceq B_{w,\overline{x},\overline{t}}$. Third step is because $P_{w,\overline{x},\overline{t}}$ is a projection matrix. Fourth step is because $\overline{t} H_{w,\overline{x}} \preceq B_{w,\overline{x},\overline{t}}$. Fifth step is by Lemma F.10. Sixth step is because $\overline{\epsilon} \leq \frac{1}{8}$.

For $\delta_s$, we have

$$\|\delta_s\|_{w,\overline{x}}^* \leq \|\overline{t}\delta_\mu\|_{w,\overline{x}}^* + \|\overline{t}^2 H_{w,\overline{x}} B_{w,\overline{x},\overline{t}}^{-1/2}(I - P_{w,\overline{x},\overline{t}}) B_{w,\overline{x},\overline{t}}^{-1/2} \delta_\mu\|_{w,\overline{x}}^* + \overline{\epsilon}\alpha\overline{t}$$

$$\leq \alpha\overline{t} + \alpha\overline{t} + \overline{\epsilon}\alpha\overline{t}$$

$$\leq \frac{17}{8}\alpha \cdot t.$$

First step is by triangle inequality and the assumption that

$$\delta_s \approx \overline{t}\delta_\mu - \overline{t}^2 H_{w,\overline{x}} B_{w,\overline{x},\overline{t}}^{-1/2}(I - P_{w,\overline{x},\overline{t}}) B_{w,\overline{x},\overline{t}}^{-1/2} \delta_\mu.$$

Second step is by same analysis as the analysis for $\delta_x$. Third step is by $\overline{t} \leq \frac{33}{32}t$ and $\overline{\epsilon} \leq \frac{1}{32}$. $\square$

The following lemma shows that $\mu^{\text{new}}$ is close to $\mu + \delta_\mu$ under an approximate step.

**Lemma F.12** (Variation of (Ye, 2020, Lemma A.9)). *For each $i \in [n]$, we define*

$$\beta_i := \|\epsilon_{\mu,i}\|_{x_i}^*$$

*For each $i \in [n]$, let*

$$\mu_i(x^{\text{new}}, s^{\text{new}}, t) = \mu_i(x, s, t) + \delta_{\mu,i} + \epsilon_{\mu,i}.$$

*Then, we have*

$$(\sum_{i=1}^{n} w_i^{-1} \beta_i^2)^{1/2} \le 15\overline{\epsilon}\alpha.$$

*Proof.* The proof is similar as (Ye, 2020, Lemma A.9), except for changing the definitions of $\epsilon_1$ and $\epsilon_2$:

$$\epsilon_1 := H_{w,\overline{x}}^{1/2}\delta_x - \overline{t} \cdot H_{w,\overline{x}}^{1/2} B_{w,\overline{x},\overline{t}}^{-1/2}(I - P_{w,\overline{x},\overline{t}}) B_{w,\overline{x},\overline{t}}^{-1/2}\delta_\mu,$$

$$\epsilon_2 := \overline{t}^{-1} H_{w,\overline{x}}^{-1/2}\delta_s - H_{w,\overline{x}}^{-1/2}(\delta_\mu - \overline{t} H_{w,\overline{x}} B_{w,\overline{x},\overline{t}}^{-1/2}(I - P_{w,\overline{x},\overline{t}}) B_{w,\overline{x},\overline{t}}^{-1/2}\delta_\mu).$$

One key step in the proof of Ye (2020) is the following property:

$$\delta_{\mu,i} = \overline{t}^{-1} \cdot \delta_{s,i} + H_{w,\overline{x}}\delta_{x,i} - H_{w,\overline{x}}^{1/2}(\epsilon_1 + \epsilon_2).$$

Under our new definition of $\epsilon_1$ and $\epsilon_2$, the above property still holds. Remaining parts of the proof are similar and we omit the details here. $\square$

The following lemma shows that error $\mu(\overline{x}, \overline{s}, \overline{t})$ on the robust central path is close to error $\mu(x, s, t)$ on the ideal central path. Furthermore, norms of errors $\gamma_i(x, s, t)$ and $\gamma_i(\overline{x}, \overline{s}, \overline{t})$ are also close to each other.

**Lemma F.13** ((Ye, 2020, Lemma A.10)). *Assume that $\gamma_i(x, s, t) \le w_i$ for all $i$. For all $i \in [n]$, we have*

$$\|\mu_i(x, s, t) - \mu_i(\overline{x}, \overline{s}, \overline{t})\|_{x_i}^* \le 3\overline{\epsilon}w_i.$$

*Furthermore, we have that*

$$|\gamma_i(x, s, t) - \gamma_i(\overline{x}, \overline{s}, \overline{t})| \le 5\overline{\epsilon}w_i.$$

*Proof.* Same as proof of (Ye, 2020, Lemma A.10). $\square$

The following lemma bounds the change of $\gamma$ under one robust IPM step.

**Lemma F.14** ((Ye, 2020, Lemma A.12)). *Assume $\Phi(x, s, t) \le \cosh(\lambda)$. For all $i \in [n]$, we define*

$$\epsilon_{r,i} := \gamma_i(x^{\text{new}}, s^{\text{new}}) - \gamma_i(x, s, t) + \alpha \cdot c_i(\overline{x}, \overline{s}, \overline{t}) \cdot \gamma_i(\overline{x}, \overline{s}, \overline{t}).$$

*Then, we have*

$$(\sum_{i=1}^{n} w_i^{-1} \epsilon_{r,i}^2)^{1/2} \le 90 \cdot \overline{\epsilon} \cdot \lambda\alpha + 4 \cdot \max_{i \in [n]}(w_i^{-1}\gamma_i(x, s, t)) \cdot \alpha.$$

*Proof.* The proof is similar to the proof of (Ye, 2020, Lemma A.12). By replacing corresponding references in Ye (2020) by our versions (Lemma F.11, F.12, F.13) we get proof of this lemma. $\square$

Finally, the following theorem bounds the movement of potential function $\Phi$ under one robust IPM step.

**Theorem F.15** (Variation of (Ye, 2020, Theorem A.15)). *Assume $\Phi(x, s, t) \le \cosh(\lambda/64)$. Then for any $0 \le h \le \frac{\alpha}{64\sqrt{\sum_{i \in [n]} w_i \nu_i}}$, we have*

$$\Phi(x^{\text{new}}, s^{\text{new}}, t^{\text{new}}) \le (1 - \frac{\alpha\lambda}{\sqrt{\sum_{i \in [n]} w_i}}) \cdot \Phi(x, s, t) + \alpha\lambda\sqrt{\sum_{i \in [n]} w_i^{-1}}.$$

*In particular, for any $\cosh(\lambda/128) \le \Phi(x, s, t) \le \cosh(\lambda/64)$, we have that*

$$\Phi(x^{\text{new}}, s^{\text{new}}, t^{\text{new}}) \le \Phi(x, s, t).$$

*Proof.* Similar to the proof of (Ye, 2020, Theorem A.15), but replacing lemmas with the corresponding QP versions. $\square$

### F.4 INITIAL POINT REDUCTION

In this section, we propose an initial point reduction scheme for quadratic programming. Our scheme is closer to Lee et al. (2019) rather than Ye (2020); Lee & Vempala (2021). The reason is that Lee & Vempala (2021)'s initial point reduction requires an efficient algorithm for finding the optimal solution to an unconstrained program, which may be difficult in quadratic programming.

**Lemma F.16** ((Nesterov, 1998, Theorem 4.1.7 and Lemma 4.2.4)). *Let $\phi$ be a $\nu$-self-concordant barrier. Then for any $x, y \in \mathrm{dom}(\phi)$, we have*

$$\langle \nabla\phi(x), y - x \rangle \leq \nu,$$

$$\langle \nabla\phi(y) - \nabla\phi(x), y - x \rangle \geq \frac{\|y - x\|_x^2}{1 + \|y - x\|_x}.$$

*Let $x^* = \arg\min_x \phi(x)$. For any $x \in \mathbb{R}^n$ such that $\|x - x^*\|_{x^*} \leq 1$, we have that $x \in \mathrm{dom}(\phi)$.*

**Lemma F.17** (QP version of (Lee et al., 2019, Lemma D.2)). *Work under the setting of Theorem F.1. Let $x^{(0)} = \arg\min_x \sum_{i\in[n]} w_i \phi_i(x_i)$. Let $\rho = \frac{1}{LR(R+1)}$. For any $0 < \epsilon \leq \frac{1}{2}$, the modified program*

$$\min_{\overline{A}\overline{x}=\overline{b}, \overline{x}\in\mathcal{K}\times\mathbb{R}_{\geq 0}} \left( \frac{1}{2}\overline{x}^\top \overline{Q}\overline{x} + \overline{c}^\top \overline{x} \right)$$

*with*

$$\overline{Q} = \begin{bmatrix} \epsilon\rho Q & 0 \\ 0 & 0 \end{bmatrix}, \qquad \overline{A} = [A \mid b - Ax^{(0)}], \qquad \overline{b} = b, \qquad \overline{c} = \begin{bmatrix} \epsilon\rho c \\ 1 \end{bmatrix}$$

*satisfies the following:*

- *$\overline{x} = \begin{bmatrix} x^{(0)} \\ 1 \end{bmatrix}$, $\overline{y} = 0 \in \mathbb{R}^m$ and $\overline{s} = \begin{bmatrix} \epsilon\rho(c + Qx^{(0)}) \\ 1 \end{bmatrix}$ are feasible primal dual vectors with $\|\overline{s} + \nabla\overline{\phi}_w(\overline{x})\|_{\overline{x}}^* \leq \epsilon$ where $\overline{\phi}_w(\overline{x}) = \sum_{i=1}^n w_i\phi_i(\overline{x}_i) - \log(\overline{x}_{n+1})$.*

- *For any $\overline{x} \in \mathcal{K} \times \mathbb{R}_{\geq 0}$ satisfying $\overline{A}\overline{x} = \overline{b}$ and*

$$\frac{1}{2}\overline{x}^\top \overline{Q}\overline{x} + \overline{c}^\top \overline{x} \leq \min_{\overline{A}\overline{x}=\overline{b}, \overline{x}\in\mathcal{K}\times\mathbb{R}_{\geq 0}} \left( \frac{1}{2}\overline{x}^\top \overline{Q}\overline{x} + \overline{c}^\top \overline{x} \right) + \epsilon^2, \tag{34}$$

*the vector $\overline{x}_{1:n}$ ( $\overline{x}_{1:n}$ is the first $n$ coordinates of $\overline{x}$ ) is an approximate solution to the original convex program in the following sense:*

$$\frac{1}{2}\overline{x}_{1:n}^\top Q\overline{x}_{1:n} + c^\top \overline{x}_{1:n} \leq \min_{Ax=b, x\in\mathcal{K}} \left( \frac{1}{2}x^\top Qx + c^\top x \right) + \epsilon\rho^{-1},$$

$$\|A\overline{x}_{1:n} - b\|_1 \leq 3\epsilon \cdot (R\|A\|_1 + \|b\|_1),$$

$$\overline{x}_{1:n} \in \mathcal{K}.$$

*Proof.* **First bullet point:** Direct computation shows that $(\overline{x}, \overline{y}, \overline{s})$ is feasible.

Let us compute $\|\overline{s} + \nabla\overline{\phi}_w(\overline{x})\|_{\overline{x}}^*$. We have

$$\|\overline{s} + \nabla\overline{\phi}_w(\overline{x})\|_{\overline{x}}^* = \|\epsilon\rho(c + Qx^{(0)})\|_{\nabla^2\phi_w(x^{(0)})^{-1}}$$

Lemma F.16 says that for all $x \in \mathbb{R}^n$ with $\|x - x^{(0)}\|_{w,x^{(0)}} \leq 1$, we have $x \in \mathcal{K}$, because $x^{(0)} = \arg\min_x \phi_w(x)$. Therefore for any $v$ such that $v^\top \nabla^2\phi_w(x^{(0)})v \leq 1$, we have $x^{(0)} \pm v \in \mathcal{K}$ and hence $\|x^{(0)} \pm v\|_2 \leq R$. This implies $\|v\|_2 \leq R$ for any $v^\top\nabla^2\phi_w(x^{(0)})v \leq 1$. Hence $(\nabla^2\phi_w(x^{(0)}))^{-1} \preceq R^2 \cdot I$. So we have

$$\|\overline{s} + \nabla\overline{\phi}_w(\overline{x})\|_{\overline{x}}^* = \|\epsilon\rho(c + Qx^{(0)})\|_{\nabla^2\phi_w(x^{(0)})^{-1}}$$
$$\leq \epsilon\rho R\|c + Qx^{(0)}\|_2$$
$$\leq \epsilon\rho R(\|c\|_2 + \|Q\|_{2\to2}\|x^{(0)}\|_2)$$

$$\leq \epsilon \rho R(L + LR)$$
$$\leq \epsilon.$$

**Second bullet point:** We define

$$\text{OPT} := \min_{Ax=b, x \in \mathcal{K}} \left( \frac{1}{2} x^\top Q x + c^\top x \right), \tag{35}$$

$$\overline{\text{OPT}} := \min_{\overline{A}\overline{x}=\overline{b}, \overline{x} \in \mathcal{K} \times \mathbb{R}_{\geq 0}} \left( \frac{1}{2} \overline{x}^\top \overline{Q} \overline{x} + \overline{c}^\top \overline{x} \right). \tag{36}$$

For any feasible $x$ in the original problem (35), $\overline{x} = \begin{bmatrix} x \\ 0 \end{bmatrix}$ is feasible in the modified problem (36).
Therefore we have

$$\overline{\text{OPT}} \leq \epsilon \rho (\frac{1}{2} x^\top Q x + c^\top x) = \epsilon \rho \cdot \text{OPT}.$$

Given a feasible $\overline{x}$ satisfying (34), we write $\overline{x} = \begin{bmatrix} \overline{x}_{1:n} \\ \tau \end{bmatrix}$ for some $\tau \geq 0$. Then we have

$$\epsilon \rho (\frac{1}{2} \overline{x}_{1:n}^\top Q \overline{x}_{1:n} + c^\top \overline{x}_{1:n}) + \tau \leq \overline{\text{OPT}} + \epsilon^2 \leq \epsilon \rho \cdot \text{OPT} + \epsilon^2.$$

Therefore

$$\frac{1}{2} \overline{x}_{1:n}^\top Q \overline{x}_{1:n} + c^\top \overline{x}_{1:n} \leq \text{OPT} + \epsilon \rho^{-1}.$$

We have

$$\tau \leq -\epsilon \rho (\frac{1}{2} \overline{x}_{1:n}^\top Q \overline{x}_{1:n} + c^\top \overline{x}_{1:n}) + \epsilon \rho \cdot \text{OPT} + \epsilon^2 \leq 3\epsilon$$

because $\left| \frac{1}{2} x^\top Q x + c^\top x \right| \leq LR(R+1)$ for all $x \in \mathcal{K}$.

Note that $\overline{x}$ satisfies $A\overline{x}_{1:n} + (b - Ax^{(0)})\tau = b$. So

$$\|A\overline{x}_{1:n} - b\|_1 \leq \|b - Ax^{(0)}\|_1 \cdot \tau.$$

This finishes the proof. $\qquad \square$

The following lemma is a generalization of (Lee et al., 2019, Lemma D.3) to quadratic program, and
with weight vector $w$.

**Lemma F.18** (QP version of (Lee et al., 2019, Lemma D.3)). *Work under the setting of Theorem F.1.
Suppose we have $\frac{s_i}{t} + w_i \nabla \phi_i(x_i) = \mu_i$ for all $i \in [n]$, $-Qx + A^\top y + s = c$ and $Ax = b$. If
$\|\mu_i\|_{x_i}^* \leq w_i$ for all $i \in [n]$, then we have*

$$\frac{1}{2} x^\top Q x + c^\top x \leq \frac{1}{2} x^{*\top} Q x^* + c^\top x^* + 4t\kappa$$

*where $x^* = \arg\min_{Ax=b, x \in \mathcal{K}} \left( \frac{1}{2} x^\top Q x + c^\top x \right)$.*

*Proof.* Let $x_\alpha = (1-\alpha)x + \alpha x^*$ for some $\alpha$ to be chosen. By Lemma F.16, we have $\langle \nabla \phi_w(x_\alpha), x^* - x_\alpha \rangle \leq \kappa$. (Note that $\phi_w$ is a $\kappa$-self-concordant barrier for $\mathcal{K}$.) Therefore we have

$$\frac{\kappa \alpha}{1 - \alpha} \geq \langle \nabla \phi_w(x_\alpha), x_\alpha - x \rangle$$

$$= \langle \nabla \phi_w(x_\alpha) - \nabla \phi_w(x), x_\alpha - x \rangle + \langle \mu - \frac{s}{t}, x_\alpha - x \rangle$$

$$\geq \sum_{i \in [n]} w_i \frac{\|x_{\alpha,i} - x_i\|_{x_i}^2}{1 + \|x_{\alpha,i} - x_i\|_{x_i}} + \langle \mu, x_\alpha - x \rangle - \frac{1}{t} \langle c - A^\top y + Qx, x_\alpha - x \rangle$$

$$\geq \sum_{i\in[n]} w_i \frac{\alpha^2 \|x_i^* - x_i\|_{x_i}^2}{1 + \alpha \|x_i^* - x_i\|_{x_i}} - \alpha \sum_{i\in[n]} \|\mu_i\|_{x_i}^* \|x_i^* - x_i\|_{x_i} - \frac{\alpha}{t} \langle c + Qx, x^* - x \rangle.$$

First step is because $\langle \nabla \phi_w(x_\alpha), x^* - x_\alpha \rangle \leq \nu$. Second step is because $\mu = \frac{s}{t} + \nabla \phi_w(x)$. Third step is by Lemma F.16 and $c = -Qx + A^\top y + s$. Fourth step is by Cauchy-Schwarz and $Ax_\alpha = Ax$.

So we get

$$\frac{1}{t}(x^\top Q x + c^\top x)$$

$$\leq \frac{1}{t}(x^\top Q x^* + c^\top x^*) + \frac{\kappa}{1 - \alpha} + \sum_{i\in[n]} \|\mu_i\|_{x_i}^* \|x_i^* - x_i\|_{x_i} - \sum_{i\in[n]} w_i \frac{\alpha \|x_i^* - x_i\|_{x_i}^2}{1 + \alpha \|x_i^* - x_i\|_{x_i}}$$

$$\leq \frac{1}{t}(\frac{1}{2}x^\top Q x + \frac{1}{2}x^{*\top} Q x^* + c^\top x^*) + \frac{\kappa}{1 - \alpha} + \sum_{i\in[n]} w_i \|x_i^* - x_i\|_{x_i} - \sum_{i\in[n]} w_i \frac{\alpha \|x_i^* - x_i\|_{x_i}^2}{1 + \alpha \|x_i^* - x_i\|_{x_i}}$$

$$= \frac{1}{t}(\frac{1}{2}x^\top Q x + \frac{1}{2}x^{*\top} Q x^* + c^\top x^*) + \frac{\kappa}{1 - \alpha} + \sum_{i\in[n]} w_i \frac{\|x_i^* - x_i\|_{x_i}}{1 + \alpha \|x_i^* - x_i\|_{x_i}}$$

$$\leq \frac{1}{t}(\frac{1}{2}x^\top Q x + \frac{1}{2}x^{*\top} Q x^* + c^\top x^*) + \frac{\kappa}{1 - \alpha} + \sum_{i\in[n]} \frac{w_i}{\alpha}$$

$$\leq \frac{1}{t}(\frac{1}{2}x^\top Q x + \frac{1}{2}x^{*\top} Q x^* + c^\top x^*) + \frac{\kappa}{\alpha(1 - \alpha)}.$$

First step is by rearranging terms in the previous inequality. Second step is by AM-GM inequality and $\|\mu_i\|_{x_i}^* \leq w_i$. Third step is by merging the last two terms. Fourth step is by bounding the last term. Fifth step is by $\sum_{i\in[n]} w_i \leq \sum_{i\in[n]} w_i \nu_i = \kappa$.

Finally,

$$\frac{1}{2}x^\top Q x + c^\top x \leq \frac{1}{2}x^{*\top} Q x^* + c^\top x^* + \frac{\kappa t}{\alpha(1 - \alpha)}$$

$$\leq \frac{1}{2}x^{*\top} Q x^* + c^\top x^* + 4\kappa t.$$

First step is by rearranging terms in the previous inequality. Second step is by taking $\alpha = \frac{1}{2}$. This finishes the proof. □

### F.5 Proof of Theorem F.1

In this section we combine everything and prove Theorem F.1.

*Proof of Theorem F.1.* Lemma F.17 shows that the initial $x$ and $s$ satisfies

$$\|\mu\|_{w,x}^* \leq \epsilon.$$

This implies $w_i^{-1} \|\mu_i\|_{x_i}^* \leq \epsilon$ because $w_i \geq 1$.

Because $\epsilon \leq \frac{1}{\lambda}$, we have

$$\Phi(x, s, t) = \sum_{i\in[n]} \cosh(\lambda w_i^{-1} \|\mu_i\|_{x_i}^*) \leq n \cosh(1) \leq \cosh(\lambda/64)$$

for the initial $x$ and $s$, by the choice of $\lambda$.

Using Theorem F.15, we see that

$$\Phi(x, s, t) \leq \cosh(\lambda/64)$$

during the entire algorithm.

So at the end of the algorithm, we have $w_i^{-1} \|\mu_i\|_{x_i}^* \leq \frac{1}{64}$ for all $i \in [n]$. In particular, $\|\mu_i\|_{x_i}^* \leq w_i$ for all $i \in [n]$.

Therefore, applying Lemma F.18 we get

$$\frac{1}{2}x^\top Q x + c^\top x \leq \frac{1}{2}x^{*\top}Q x^* + c^\top x^* + 4t\kappa$$
$$\leq \frac{1}{2}x^{*\top}Q x^* + c^\top x^* + \epsilon^2$$

where we used the stop condition for $t$ at the end.

So Lemma F.17 shows how to get an approximate solution for the original quadratic program with error $\epsilon L R(R+1)$.

The number of iterations is because we decrease $t$ by a factor of $1-h$ every iteration, and the choice $h = \frac{\alpha}{64\sqrt{\kappa}}$. □

# G  GAUSSIAN KERNEL SVM: ALMOST-LINEAR TIME ALGORITHM AND HARDNESS

In this section, we provide both algorithm and hardness for Gaussian kernel SVM problem. For the algorithm, we utilize a result due to Aggarwal & Alman (2022) in conjunction with our low-rank QP solver to obtain an $O(n^{1+o(1)}\log(1/\epsilon))$ time algorithm. For the hardness, we build upon the framework outlined in Backurs et al. (2017) and improve their results in terms of dependence on dimension $d$.

We start by proving a simple lemma that shows that if $K = UV^\top$ for low-rank $U, V$, then the quadratic objective $K \circ (yy^\top)$ also admits such a factorization via a simple scaling.

**Lemma G.1.** *Let $U, V \in \mathbb{R}^{n\times k}$ and $y \in \mathbb{R}^n$. Then, there exists a pair of matrices $\widetilde{U}, \widetilde{V} \in \mathbb{R}^{n\times k}$ such that*

$$\widetilde{U}\widetilde{V}^\top = (UV^\top) \circ (yy^\top)$$

*moreover, $\widetilde{U}, \widetilde{V}$ can be computed in time $O(nk)$.*

*Proof.* The proof relies on the following identity for Hadamard product: for any matrix $A$ and conforming vectors $x, y$ (all real), one has

$$A \circ (yx^\top) = D_y A D_x$$

where $D_y, D_x \in \mathbb{R}^{n\times n}$ are diagonal matrices that put $y, x$ on their diagonals. Thus, we can simply compute $\widetilde{U}, \widetilde{V}$ as follows:

$$\widetilde{U} = D_y U,$$
$$\widetilde{V} = D_y V,$$

consequently,

$$\widetilde{U}\widetilde{V}^\top = D_y UV^\top D_y$$
$$= (yy^\top) \circ (UV^\top)$$
$$= (UV^\top) \circ (yy^\top),$$

as desired. Moreover, the diagonal scaling of $U, V$ can be indeed performed in $O(nk)$ time, as advertised. □

Throughout this section, we will let $B$ denote the squared radius of the dataset.

## G.1  ALMOST-LINEAR TIME ALGORITHM FOR GAUSSIAN KERNEL SVM

We state a result due to Aggarwal & Alman (2022), in which they present an optimal-degree polynomial approximation to the function $e^{-x}$ and consequentially, this produces an efficient approximate scheme to the Batch Gaussian Kernel Density Estimation problem.

We start by introducing a notion that captures the minimum degree polynomial that well-approximates $e^{-x}$:

**Definition G.2.** Let $f : [0, B] \to \mathbb{R}$, we let $q_{B;\epsilon}(f) \in \mathbb{N}$ denote the minimum degree of a non-constant polynomial $p(x)$ such that

$$\sup_{x \in [0,B]} |p(x) - f(x)| \leq \epsilon$$

Utilizing the Chebyshev polynomial machinery together with the orthgonal polynomial families, Aggarwal & Alman (2022) provides the following characterization on $q_{B;\epsilon}(f)$:

**Theorem G.3** (Theorem 1.2 of Aggarwal & Alman (2022))**.** *Let $B \geq 1$ and $\epsilon \in (0, 1)$. Then*

$$q_{B;\epsilon}(e^{-x}) = \Theta(\max\{\sqrt{B \log(1/\epsilon)}, \frac{\log(1/\epsilon)}{\log(B^{-1} \log(1/\epsilon))}\})$$

**Theorem G.4** (Corollary 1.7 of Aggarwal & Alman (2022))**.** *Let $x_1, \ldots, x_n \in \mathbb{R}^d$ be a dataset with squared radius $B$ and $\epsilon \in (0, 1)$. Let $q = q_{B;\epsilon}(e^{-x})$. Let $K \in \mathbb{R}^{n \times n}$ be the Gaussian kernel matrix formed by $x_1, \ldots, x_n$. Finally, let $k = \binom{2d+2q}{2q}$. Then, there exists a deterministic algorithm that computes a pair of matrices $U, V \in \mathbb{R}^{n \times k}$ such that for any vector $v \in \mathbb{R}^n$,*

$$\|Kv - UV^\top v\|_\infty \leq \epsilon \|v\|_1.$$

*Moreover, matrices $U, V$ can be computed in time $O(nkd)$.*

Even though $\ell_\infty$ error in terms of $\ell_1$ norm of vector $v$ seems quite weak, it can be conveniently translated into more standard guarantees, e.g., spectral norm error. The following lemma provides a conversion of errors that come in handy later when integrating the kernel approximation to our low-rank QP solver.

**Lemma G.5.** *Let $K \in \mathbb{R}^{n \times n}$ be a PSD kernel matrix and $\epsilon \in (0, 1)$ be a parameter. Let $\widetilde{K} \in \mathbb{R}^{n \times n}$ be an approximation to $K$ with the guarantee that for any $v \in \mathbb{R}^n$,*

$$\|Kv - \widetilde{K}v\|_\infty \leq \epsilon \|v\|_1,$$

*then*

$$|v^\top K v - v^\top \widetilde{K} v| \leq \epsilon \|v\|_1^2 \leq \epsilon n \|v\|_2^2.$$

*Proof.* The proof is a simple application of Hölder's inequality:

$$\begin{aligned}
|v^\top (Kv - \widetilde{K}v)| &= |\langle v, Kv - \widetilde{K}v \rangle| \\
&\leq \|v\|_1 \|Kv - \widetilde{K}v\|_\infty \\
&\leq \epsilon \|v\|_1^2 \\
&\leq \epsilon n \|v\|_2^2,
\end{aligned}$$

where the second step is by Hölder's inequality, and the last step is by Cauchy-Schwarz. This completes the proof. $\square$

We can now combine the Gaussian kernel low-rank decomposition with our low-rank QP solver to provide an almost-linear time algorithm for Gaussian kernel SVM. We restate the kernel SVM formulation here.

**Definition G.6** (Restatement of Definition 1.3)**.** Given a data matrix $X \in \mathbb{R}^{n \times d}$ and labels $y \in \mathbb{R}^n$. Let $Q \in \mathbb{R}^{n \times n}$ denote a matrix where $Q_{i,j} = \mathsf{K}(x_i, x_j) \cdot y_i y_j$ for $i, j \in [n]$. The hard-mragin kernel SVM problem with bias asks to solve the following program.

$$\begin{aligned}
\max_{\alpha \in \mathbb{R}^n} \quad & \mathbf{1}_n^\top \alpha - \frac{1}{2} \alpha^\top Q \alpha \\
\text{s.t.} \quad & \alpha^\top y = 0 \\
& \alpha \geq 0.
\end{aligned}$$

**Theorem G.7.** *Let Gaussian kernel SVM training problem be defined as above with kernel function $\mathsf{K}(x_i, x_j) = \exp(-\|x_i - x_j\|_2^2)$. Suppose the dataset has squared radius $B \geq 1$, and let $\epsilon \in (0, 1)$ be the precision parameter. Suppose the program satisfies the following:*

- *There exists a point $z \in \mathbb{R}^n$ such that there is an Euclidean ball with radius $r$ centered at $z$ that is contained in the constraint set.*

- *The constraint set is enclosed by an Euclidean ball of radius $R$, centered at the origin.*

*Then, there exists a randomized algorithm that outputs an approximate solution $\widehat{\alpha} \in \mathbb{R}^n$ such that $\widehat{\alpha} \geq 0$, moreover,*

$$\mathbf{1}_n^\top \widehat{\alpha} - \frac{1}{2}\widehat{\alpha}^\top Q \widehat{\alpha} \geq \mathrm{OPT} - \epsilon,$$

$$\|\widehat{\alpha}^\top y\|_1 \leq 3\epsilon,$$

*where $\mathrm{OPT}$ denote the optimal cost of the objective function. Let $q = q_{B;\Theta(\epsilon/nR^2)}(e^{-x})$ and $k = \binom{2d+2q}{2q}$. Then, the vector $\widehat{\alpha}$ can be computed in expected time*

$$\widetilde{O}(nk^{(\omega+1)/2}\log(nR/(\epsilon r))).$$

*Proof.* Throughout the proof, we set $\epsilon_1 = O(\epsilon/(nR^2))$. We will craft an algorithm that first computes an approximate Gaussian kernel together with a proper low-rank factorization, then use this proxy kernel matrix to solve the quadratic program. We will use $K$ to denote the exact Gaussian kernel matrix, $Q$ to denote the exact quadratic matrix.

**Approximate the Gaussian kernel matrix with finer granularity.** We start by invoking Theorem G.4 using data matrix $X$ with accuracy parameter $\epsilon_1$. We let $\widetilde{K} = UV^\top$ to denote this approximate kernel matrix, and we let $\widetilde{Q} = D_y UV^\top D_y$ to denote the approximate quadratic matrix. Owing to Lemma G.5, we know that for any vector $x \in \mathbb{R}^n$,

$$|x^\top(Q - \widetilde{Q})x| = |(D_y x)^\top (K - \widetilde{K})(D_y x)|$$
$$\leq \epsilon_1 n \|D_y x\|_2^2$$
$$= \epsilon_1 n \|x\|_2^2,$$

where we use the fact that $y \in \{\pm 1\}^n$. This also implies that

$$\|Q - \widetilde{Q}\| \leq \epsilon_1 n \tag{37}$$

this simple bound will come in handy later.

**Solving the approximate program to high precision.** Given $\widetilde{Q}$, we solve the following program:

$$\max_{\alpha \in \mathbb{R}^n} \mathbf{1}_n^\top \alpha - \frac{1}{2}\alpha^\top \widetilde{Q}\alpha$$
$$\text{s.t. } \alpha^\top y = 0$$
$$\alpha \geq 0$$

by invoking Theorem E.1. To do so, we need a bound on the Lipschitz constant of the program, i.e., the spectral norm of $\widetilde{Q}$ and $\ell_2$ norm of $\mathbf{1}$. The latter is clearly $\sqrt{n}$, we shall show the first term is at most $(1 + \epsilon_1) \cdot n$.

Note that

$$\|Q\| = \|D_y K D_y\|$$
$$\leq \mathrm{tr}[D_y K D_y]$$
$$= \mathrm{tr}[K]$$
$$\leq n,$$

where we use $K$ is PSD. Combining with Eq. (37) and triangle inequality, we have

$$\|\widetilde{Q}\| \leq \|Q\| + \|Q - \widetilde{Q}\|$$
$$\leq (1 + \epsilon_1) \cdot n.$$

With these Lipschitz constants, we examine the error guarantee provided by Theorem E.1: it produces a vector $\widehat{\alpha} \in \mathbb{R}^n$ such that

$$\mathbf{1}_n^\top \widehat{\alpha} - \frac{1}{2}\widehat{\alpha}^\top \widetilde{Q}\widehat{\alpha} \geq \max_{\alpha^\top y = 0, x \geq 0}(\mathbf{1}_n^\top \alpha - \frac{1}{2}\alpha^\top \widetilde{Q}\alpha) - O(\epsilon_1 n R^2),$$

$$\|\widehat{\alpha}^\top y\|_1 \leq O(\epsilon_1 n R),$$

we mainly focus on the first error bound, as we need to understand the quality of $\widehat{x}$ when plug into the program with $Q$.

We will follow a chain of triangle inequalities, so we first bound

$$|\widehat{\alpha}^\top (\widetilde{Q} - Q)\widehat{\alpha}| \leq \epsilon n \|\widehat{\alpha}\|_2^2$$
$$\leq \epsilon n R^2.$$

Next, let

$$\alpha' := \arg \max_{\alpha^\top y = 0, \alpha \geq 0} \mathbf{1}_n^\top \alpha - \frac{1}{2}\alpha^\top \widetilde{Q}\alpha,$$

$$\alpha^* := \arg \max_{\alpha^\top y = 0, \alpha \geq 0} \mathbf{1}_n^\top \alpha - \frac{1}{2}\alpha^\top Q\alpha,$$

then we have the following

$$\mathbf{1}_n^\top \alpha' - \frac{1}{2}\alpha'^\top \widetilde{Q}\alpha' \geq \mathbf{1}_n^\top \alpha^* - \frac{1}{2}(\alpha^*)^\top \widetilde{Q}\alpha^*$$

$$\geq \mathbf{1}_n^\top \alpha^* - \frac{1}{2}(\alpha^*)^\top Q\alpha^* - O(\epsilon_1 n R^2)$$

$$= \mathrm{OPT} - O(\epsilon_1 n R^2),$$

where the second step is by applying Lemma G.5 to $\alpha^*$. Now we are ready to bound the final error:

$$\mathbf{1}_n^\top \widehat{\alpha} - \frac{1}{2}\widehat{\alpha}^\top Q\widehat{\alpha} \geq \mathbf{1}_n^\top \widehat{\alpha} - \frac{1}{2}\widehat{\alpha}^\top \widetilde{Q}\widehat{\alpha} - O(\epsilon_1 n R^2)$$

$$\geq \mathbf{1}_n^\top \alpha' - \frac{1}{2}\alpha'^\top \widetilde{Q}\alpha' - O(\epsilon_1 n R^2)$$

$$\geq \mathrm{OPT} - O(\epsilon_1 n R^2).$$

The final error guarantee follows by the choice of $\epsilon_1$, and we indeed design an algorithm that outputs a vector $\widehat{\alpha}$ with

$$\mathbf{1}^\top \widehat{\alpha} - \frac{1}{2}\widehat{\alpha}^\top Q\widehat{\alpha} \geq \mathrm{OPT} - \epsilon,$$

$$\|\widehat{\alpha}^\top y\|_1 \leq \epsilon.$$

**Runtime analysis.** It remains to analyze the runtime of our proposed algorithm. We first compute an approximate kernel $\widetilde{K}$ with parameter $\epsilon_1$, owing to Theorem G.4, we have

$$q_{B;\epsilon_1}(e^{-x}) = \Theta(\max\{\sqrt{B \log(nR/\epsilon)}, \frac{\log(nR/\epsilon)}{\log(B^{-1}\log(nR/\epsilon))}\})$$

then by setting $k = \binom{2d+2q}{2q}$, the matrix $\widetilde{K}$ can be computed in time $O(nkd)$. Given this rank-$k$ factorization, the program can then be solved with precision $\epsilon_1$ in time

$$\widetilde{O}(nk^{(\omega+1)/2}\log(nR/(\epsilon r))),$$

as desired. □

**Remark G.8.** To understand the value range of $k$, let us consider the set of parameters:

$$d = O(\log n), \epsilon = 1/\mathrm{poly}\, n, R = \mathrm{poly}\, n, B = \Theta(1),$$

under this setting, $O(\log(nR/\epsilon)) = O(\log n)$ and the degree $q$ is

$$q = \Theta(\sqrt{\log n})$$

the rank $k$ is then

$$
\begin{aligned}
k &= \binom{2d + 2q}{2q} \\
&\leq \Theta((\log n)^{\frac{1}{2}\sqrt{\log n}}) \\
&= \Theta(2^{\Theta(\log \log n \sqrt{\log n})}) \\
&= n^{o(1)},
\end{aligned}
$$

consequentially, our algorithm runs in almost-linear time in $n$:

$$\widetilde{O}(n^{1+o(1)} \log n).$$

It is worth noting to achieve the almost-linear runtime, the data radius $B$ can be further relaxed. In fact, as long as

$$B = o\left(\frac{\log n}{\log \log n}\right),$$

we can ensure that $k = n^{o(1)}$ and subsequently the almost-linear runtime.

The runtime we obtain can be viewed as a consequence of the "phase transition" phenomenon observed in Aggarwal & Alman (2022), in which they prove that if $B = \omega(\log n)$, then quadratic time in $n$ is essentially needed to approximate the Gaussian kernel assuming SETH.

### G.2 HARDNESS OF GAUSSIAN KERNEL SVM WITH LARGE RADIUS

In this section, we show that for $d = O(\log n)$, any algorithm that solves the program associated to hard-margin Gaussian kernel SVM would require $\Omega(n^{2-o(1)})$ time for $B = \omega(\log n)$. This justifies the choice of $B$ in Remark G.8. To prove the hardness result, we need to introduce the approximate Hamming nearest neighbor problem.

**Definition G.9.** For $\delta > 0$ and $n, d \in \mathbb{N}$, let $\{a_1, \ldots, a_n\}, \{b_1, \ldots, b_n\} \subseteq \{0,1\}^d$ be two sets of vectors, and let $t \in \{0, 1, \ldots, d\}$ be a threshold. The $(1 + \delta)$-*Approximate Hamming Nearest Neighbor* problem asks to distinguish the following two cases:

- If there exists some $a_i$ and $b_j$ such that $\|a_i - b_j\|_1 \leq t$, output "Yes";

- If for any $i, j \in [n]$, we have $\|a_i - b_j\|_1 > (1 + \delta) \cdot t$, output "No".

Note that the algorithm can output any value if the datasets fall in neither of these two cases. We will utilize the following hardness result due to Rubinstein.

**Theorem G.10** (Rubinstein (2018))**. *Assuming SETH, for every $q > 0$, there exists $\delta > 0$ and $C > 0$ such that $(1 + \delta)$-Approximate Hamming Nearest Neighbor in dimension $d = C \log n$ requires time $\Omega(n^{2-q})$.*

A final ingredient is a rewriting of the dual SVM into its primal form, without resorting to optimize over an infinite-dimensional hyperplane.

**Lemma G.11.** *Consider the dual hard-margin kernel SVM defined as*

$$
\max_{\alpha \in \mathbb{R}^n} \mathbf{1}^\top \alpha - \frac{1}{2} \sum_{i,j \in [n] \times [n]} \alpha_i \alpha_j y_i y_j \mathsf{K}(w_i, w_j),
$$

$$
\text{s.t. } \alpha^\top y = 0,
$$

$$
\alpha \geq 0.
$$

*The primal program can be written as*

$$
\min_{\alpha \in \mathbb{R}^n} \frac{1}{2} \sum_{i,j \in [n] \times [n]} \alpha_i \alpha_j y_i y_j \mathsf{K}(w_i, w_j),
$$

$$\text{s.t. } y_i f(w_i) \geq 1,$$
$$\alpha \geq 0,$$

where $f : \mathbb{R}^d \to \mathbb{R}$ is defined as

$$f(w) = \sum_{j=1}^{n} \alpha_j y_j \mathsf{K}(w_j, w) - b.$$

*Moreover, the primal and dual program has no duality gap and the optimal solution $\alpha$ to both programs are the same.*

*Proof.* Recall that the primal hard-margin SVM is the following program:

$$\min_{v} \frac{1}{2} \|v\|_2^2,$$
$$\text{s.t. } y_i(v^\top \phi(w_i) - b) \geq 1,$$

where $b \in \mathbb{R}$ is the bias term and $\phi : \mathbb{R}^d \to \mathbb{R}^K$ is the feature mapping corresponding to the kernel in the sense that $\mathsf{K}(w_i, w_j) = \phi(w_i)^\top \phi(w_j)$. The optimal weight $v = \sum_{i=1}^{n} \alpha_i y_i \phi(w_i)$ where $\alpha \in \mathbb{R}^n$ is the optimal solution to the dual program. Consequently,

$$\|v\|_2^2 = (\sum_{i=1}^{n} \alpha_i y_i \phi(w_i))^2$$
$$= \sum_{i,j \in [n] \times [n]} \alpha_i \alpha_j y_i y_j \phi(w_i)^\top \phi(w_j)$$
$$= \sum_{i,j \in [n] \times [n]} \alpha_i \alpha_j y_i y_j \mathsf{K}(w_i, w_j)$$
$$= \alpha^\top Q \alpha,$$

where the matrix $Q$ is the usual

$$Q = (yy^\top) \circ K,$$

the constraint can be rewritten as

$$y_i(v^\top \phi(w_i) - b) = y_i((\sum_{i=1}^{n} \alpha_i y_i \phi(w_i))^\top \phi(w_i) - b)$$
$$= y_i(\sum_{j=1}^{n} \alpha_j y_j \phi(w_j)^\top \phi(w_i)) - y_i b$$
$$= y_i(\sum_{j=1}^{n} \alpha_j y_j \mathsf{K}(w_i, w_j)) - y_i b$$
$$= y_i f(w_i),$$

where $f : \mathbb{R}^d \to \mathbb{R}$ is defined as

$$f(w) = \sum_{j=1}^{n} \alpha_j y_j \mathsf{K}(w_j, w) - b.$$

Thus, we can alternatively write the primal as

$$\min_{\alpha \in \mathbb{R}^n} \frac{1}{2} \alpha^\top Q \alpha,$$
$$\text{s.t. } y_i f(w_i) \geq 1.$$

For the strong duality and optimal solution, see, e.g., Muller et al. (2001). $\square$

We will now prove the almost-quadratic lower bound for Guassian kernel SVM. Our proof strategy is similar to that of Backurs et al. (2017) with different set of parameters. It is also worth noting that the Backurs et al. (2017) construction

- Requires the dimension $d = \Theta(\log^3 n)$;
- Requires the squared dataset radius $B = \Theta(\log^4 n)$.

We will improve both of these results.

**Theorem G.12.** *Assuming* SETH, *for every $q > 0$, there exists a hard-margin Gaussian kernel SVM without the bias term as defined in Definition 1.3 with $d = \Theta(\log n)$ and error $\epsilon = \exp(-\Theta(\log^2 n))$ for inputs whose squared radius is at most $B = \Theta(\log^2 n)$ requiring time $\Omega(n^{2-q})$ to solve.*

*Proof.* Let $l = \sqrt{2(c'\delta)^{-1}\log n}$. We will provide a reduction from $(1 + \delta)$-Approximate Hamming Nearest Neighbor to Gaussian kernel SVM. Let $A := \{a_1, \ldots, a_n\}, B := \{b_1, \ldots, b_n\} \subseteq \{0, 1\}^d$ be the datasets, we assign label 1 to all vectors $a_i$ and label $-1$ to all vectors $b_j$, moreover, we scale both $A$ and $B$ by $l$, this results in two datasets with points in $\{0, l\}^d$. The squared radius of this dataset is then

$$B = \max\{\max_{i,j} \|la_i - la_j\|_2^2, \max_{i,j} \|lb_i - lb_j\|_2^2, \max_{i,j} \|la_i - lb_j\|_2^2\}$$
$$\leq l^2 d$$
$$= \Theta(\delta^{-1} \log^2 n).$$

To simplify the notation, we will implictly assume $A$ and $B$ are scaled by $l$ without explicitly writing out $la_i$, $lb_j$. Now consider the following three programs:

- Classifying $A$:

$$\min_{\alpha \in \mathbb{R}_{\geq 0}^n} \frac{1}{2} \sum_{i,j \in [n] \times [n]} \alpha_i \alpha_j \mathsf{K}(a_i, a_j),$$

$$\text{s.t.} \sum_{j=1}^n \alpha_j \mathsf{K}(a_i, a_j) \geq 1, \qquad \forall i \in [n] \tag{38}$$

- Classifying $B$:

$$\min_{\beta \in \mathbb{R}_{\geq 0}^n} \frac{1}{2} \sum_{i,j \in [n] \times [n]} \beta_i \beta_j \mathsf{K}(b_i, b_j),$$

$$\text{s.t.} -\sum_{j=1}^n \beta_j \mathsf{K}(b_i, b_j) \leq -1, \qquad \forall i \in [n] \tag{39}$$

- Classifying both $A$ and $B$:

$$\min_{\alpha,\beta \in \mathbb{R}_{\geq 0}^n} \frac{1}{2} \sum_{i,j \in [n] \times [n]} \alpha_i \alpha_j \mathsf{K}(a_i, a_j) + \frac{1}{2} \sum_{i,j \in [n] \times [n]} \beta_i \beta_j \mathsf{K}(b_i, b_j) - \sum_{i,j \in [n] \times [n]} \alpha_i \beta_j \mathsf{K}(a_i, b_j),$$

$$\text{s.t.} \sum_{j=1}^n \alpha_j \mathsf{K}(a_i, a_j) - \sum_{j=1}^n \beta_j \mathsf{K}(a_i, b_j) \geq 1, \qquad \forall i \in [n],$$

$$\sum_{j=1}^n \alpha_j \mathsf{K}(b_i, a_j) - \sum_{j=1}^n \beta_j \mathsf{K}(b_i, b_j) \leq -1, \qquad \forall i \in [n] \tag{40}$$

We will prove that the optimal solution $\alpha_i^*$'s and $\beta_i^*$'s are both lower and upper bounded. Use $\mathrm{Val}(A), \mathrm{Val}(B)$ and $\mathrm{Val}(A, B)$ to denote the value of program (38), (39) and (40) respectively, then note that

$$\mathrm{Val}(A) \leq \frac{n^2}{2}$$

by plugging in $\alpha = \mathbf{1}$ and setting all vectors to be the same. On the other hand,

$$\mathrm{Val}(A) \geq \frac{1}{2} \sum_{i=1}^{n} (\alpha_i^*)^2 \mathsf{K}(a_i, a_i)$$

$$= \frac{1}{2} \sum_{i=1}^{n} (\alpha_i^*)^2.$$

Combining these two, we can conclude that for any $\alpha_i^*$, it must be $\alpha_i^* \leq n$. For the lower bound, consider the inequality constraint for the $i$-th point:

$$\alpha_i^* + \sum_{j \neq i} \alpha_j^* \mathsf{K}(a_i, a_j) \geq 1,$$

to estimate $\mathsf{K}(a_i, a_j)$, note that $\|a_i - a_j\|_2^2 = \|a_i - a_j\|_1 \geq 1$ for $j \neq i$,[9] and

$$\begin{aligned}
\mathsf{K}(a_i, a_j) &= \exp(-l^2 \|a_i - a_j\|_2^2) \\
&= \exp(-2(c'\delta)^{-1} \log n \|a_i - a_j\|_1) \\
&\leq \exp(-2(c'\delta)^{-1} \log n) \\
&\leq n^{-10}/100,
\end{aligned}$$

combining with $\alpha_j^* \leq n$, we have

$$\begin{aligned}
\alpha_i^* &\geq 1 - \sum_{j \neq i} \alpha_j^* \mathsf{K}(a_i, a_j) \\
&\geq 1 - n \cdot n \cdot n^{-10}/100 \\
&\geq 1/2.
\end{aligned}$$

This lower bound on $\alpha_i^*$ is helpful when we attempt to lower bound $\mathrm{Val}(A, B)$ with $\mathrm{Val}(A) + \mathrm{Val}(B)$. Following the outline of Backurs et al. (2017), we consider the three dual programs:

- Dual of classifying $A$:

$$\max_{\alpha \in \mathbb{R}_{\geq 0}^n} \sum_{i=1}^{n} \alpha_i - \frac{1}{2} \sum_{i,j} \alpha_i \alpha_j \mathsf{K}(a_i, a_j) \tag{41}$$

- Dual of classifying $B$:

$$\max_{\beta \in \mathbb{R}_{\geq 0}^n} \sum_{i=1}^{n} \beta_i - \frac{1}{2} \sum_{i,j} \beta_i \beta_j \mathsf{K}(b_i, b_j) \tag{42}$$

- Dual of classifying $A$ and $B$:

$$\max_{\alpha, \beta \in \mathbb{R}_{\geq 0}^n} \sum_{i=1}^{n} \alpha_i + \sum_{i=1}^{n} \beta_i - \frac{1}{2} \sum_{i,j} \alpha_i \alpha_j \mathsf{K}(a_i, a_j) - \frac{1}{2} \sum_{i,j} \beta_i \beta_j \mathsf{K}(b_i, b_j) + \sum_{i,j} \alpha_i \beta_j \mathsf{K}(a_i, b_j) \tag{43}$$

as the SVM program exhibits strong duality, we know that the optimal value of the primal equals to the dual, so we can alternatively bound $\mathrm{Val}(A, B)$ using the dual program. Plug in $\alpha^*, \beta^*$ to program (43), we have

$$\begin{aligned}
\mathrm{Val}(A, B) &\geq \sum_{i=1}^{n} \alpha_i^* + \sum_{i=1}^{n} \beta_i^* - \frac{1}{2} \sum_{i,j} \alpha_i^* \alpha_j^* \mathsf{K}(a_i, a_j) - \frac{1}{2} \sum_{i,j} \beta_i^* \beta_j^* \mathsf{K}(b_i, b_j) + \sum_{i,j} \alpha_i^* \beta_j^* \mathsf{K}(a_i, b_j) \\
&= \mathrm{Val}(A) + \mathrm{Val}(B) + \sum_{i,j} \alpha_i^* \beta_j^* \mathsf{K}(a_i, b_j),
\end{aligned}$$

---

[9]We without loss of generality that during preprocess, we have remove duplicates in $A$ and $B$.

to bound the third term, we consider the pair $(a_{i_0}, b_{j_0})$ such that $\|a_{i_0} - b_{j_0}\|_1 \leq t - 1$, and note that

$$\sum_{i,j} \alpha_i^* \beta_j^* \mathsf{K}(a_i, b_j) \geq \alpha_{i_0}^* \beta_{j_0}^* \mathsf{K}(a_{i_0}, b_{j_0})$$

$$\geq \frac{1}{4} \exp(-2(c'\delta)^{-1} \log n \cdot (t-1)).$$

To wrap up, we have

$$\mathrm{Val}(A, B) \geq \mathrm{Val}(A) + \mathrm{Val}(B) + \frac{1}{4} \exp(-2(c'\delta)^{-1} \log n \cdot (t-1))$$

We now prove the "No" case, where for any $a_i, b_j$, $\|a_i - b_j\|_1 \geq t$. We have

$$\mathsf{K}(a_i, b_j) = \exp(-l^2 \|a_i - b_j\|_2^2)$$

$$\leq \exp(-2(c'\delta)^{-1} \log n \cdot t),$$

we let $m := \exp(-2(c'\delta)^{-1} \log n \cdot t)$, set $\alpha' = \alpha^* + 10n^2 m$ and $\beta' = \beta^* + 10n^2 m$, we let $V$ to denote the value when evaluating program (40) with $\alpha', \beta'$. We will essentially show that

$$\mathrm{Val}(A, B) \leq V$$

and

$$V \leq \mathrm{Val}(A) + \mathrm{Val}(B) + 400n^6 m$$

chaining these two gives us a certificate for the "No" case. To prove the first assertion, we show that $\alpha', \beta'$ are feasible solution to program (40) since

$$\sum_{j=1}^n \alpha_j' \mathsf{K}(a_i, a_j) = \sum_{j=1}^n (\alpha_j^* \mathsf{K}(a_i, a_j) + 10n^2 m \mathsf{K}(a_i, a_j))$$

$$= \alpha_i^* + 10n^2 m + \sum_{j \neq i} (\alpha_j^* + 10n^2 m) \mathsf{K}(a_i, a_j)$$

$$\geq \alpha_i^* + \sum_{j \neq i} \alpha_j^* \mathsf{K}(a_i, a_j) + 10n^2 m$$

$$= 10n^2 m + \sum_{j=1}^n \alpha_j^* \mathsf{K}(a_i, a_j)$$

$$\geq 10n^2 m + 1$$

where we use $\alpha_i^*$ satisfy the inequality constraint of program (38). We compute an upper bound on $\sum_{j=1}^n \beta_j' \mathsf{K}(a_i, b_j)$:

$$\sum_{j=1}^n \beta_j' \mathsf{K}(a_i, b_j) \leq \sum_{j=1}^n 2nm$$

$$= 2n^2 m,$$

where we use the fact that $m = \exp(-2(c'\delta)^{-1} \log n \cdot t) \leq n^{-10}/10$ therefore $\beta^* + 10n^2 m \leq 2n$. Thus, it must be the case that

$$\sum_{j=1}^n \alpha_j' \mathsf{K}(a_i, a_j) - \sum_{j=1}^n \beta_j' \mathsf{K}(a_i, b_j) \geq 8n^2 m + 1$$

$$\geq 1,$$

as desired. The other linear constraint follows by a symmetric argument. This indeed shows that $\alpha', \beta'$ are feasible solutions to program (40) and $\mathrm{Val}(A, B) \leq V$.

To prove an upper bound on $V$, we note that

$$V = \frac{1}{2} \sum_{i,j} \alpha_i' \alpha_j' \mathsf{K}(a_i, a_j) + \frac{1}{2} \sum_{i,j} \beta_i' \beta_j' \mathsf{K}(b_i, b_j) - \sum_{i,j} \alpha' \beta_j' \mathsf{K}(a_i, b_j)$$

$$\leq \frac{1}{2}\sum_{i,j}\alpha_i'\alpha_j'\mathsf{K}(a_i,a_j) + \frac{1}{2}\sum_{i,j}\beta_i'\beta_j'\mathsf{K}(b_i,b_j),$$

we bound the first quantity, as the second follows similarly:

$$\frac{1}{2}\sum_{i,j}\alpha_i'\alpha_j'\mathsf{K}(a_i,a_j) = \frac{1}{2}\sum_{i,j}(\alpha_i^*\alpha_j^* + 10n^2m(\alpha_i^* + \alpha_j^*) + 100n^4m^2)\mathsf{K}(a_i,a_j)$$

$$\leq \mathrm{Val}(A) + \sum_{i,j}10n^3m\mathsf{K}(a_i,a_j) + \sum_{i,j}100n^4m^2\mathsf{K}(a_i,a_j)$$

$$\leq \mathrm{Val}(A) + 10n^5m + 100n^6m^2$$

$$\leq \mathrm{Val}(A) + 200n^6m,$$

we can thus conclude

$$V \leq \mathrm{Val}(A) + \mathrm{Val}(B) + 400n^6m.$$

Chaining these two, we obtain the following threshold for the "No" case:

$$\mathrm{Val}(A,B) \leq \mathrm{Val}(A) + \mathrm{Val}(B) + 400n^6m.$$

Finally, we observe that

$$400n^6\exp(-2(c'\delta)^{-1}\log n \cdot t) \ll \frac{1}{4}\exp(-2(c'\delta)^{-1}\log n \cdot (t-1)),$$

we can therefore distinguish these two cases.

Note that when one considers solving the program with additive error, we need to make sure that the error is smaller than the distinguishing threshold, i.e.,

$$\epsilon \leq \frac{1}{4}\exp(-2(c'\delta)^{-1}\log n \cdot (t-1))$$

$$\leq \frac{1}{4}\exp(-2(c'\delta)^{-1}d\log n)$$

$$= \exp(-\Theta(\log^2 n)),$$

where we use $t \leq d$ and $d = \Theta(\log n)$. This concludes the proof. $\qquad\square$

**Remark G.13.** Our proof can be interpreted as using a stronger complexity theoretical tool in place of the one used by Backurs et al. (2017), to obtain a better dependence on dimension $d$ and $B$. We also note that the construction due to Backurs et al. (2017) has the relation that $B = \Theta(d\log n)$, this is because in order to lower bound $\mathrm{Val}(A,B)$, one has to lower bound the optimal values of $\alpha_i^*$'s and $\beta_j^*$'s. To do so, one needs to further scale up $a_i$'s and $b_j$'s so that within datasets $A$ and $B$, the radius is at least $\Theta(\log n)$. This is in contrast to the Batch Gaussian KDE studied in Aggarwal & Alman (2022), where they show the almost-quadratic lower bound can be achieved for both $d, B = \Theta(\log n)$.

Similar to Backurs et al. (2017), we obtain hardness results for hard-margin kernel SVM with bias, and soft-margin kernel SVM with bias.

**Corollary G.14.** *Assuming* SETH*, for every $q > 0$, there exists a hard-margin Gaussian kernel SVM with the bias term with $d = \Theta(\log n)$ and error $\epsilon = \exp(-\Theta(\log^2 n))$ for inputs whose squared radius is at most $B = \Theta(\log^6 n)$ requiring time $\Omega(n^{2-q})$ to solve.*

*Proof.* The proof is similar to Backurs et al. (2017). Given a hard instance of Theorem G.12, except we append $\Theta(\log n)$ entries with magnitude $\Theta(\log^2 n)$ instead of distributing the values across $\Theta(\log^3 n)$ entries. Rest of the proof follows exactly the same as Backurs et al. (2017). $\qquad\square$

**Corollary G.15.** *Assuming* SETH*, for every $q > 0$, there exists a soft-margin Gaussian kernel SVM with the bias term with $d = \Theta(\log n)$ and error $\epsilon = \exp(-\Theta(\log^2 n))$ for inputs whose squared radius is at most $B = \Theta(\log^6 n)$ requiring time $\Omega(n^{2-q})$ to solve.*

**Remark G.16.** Compared to the construction of Backurs et al. (2017) in which they distribute a total mass of $\Theta(\log^3 n)$ across $\Theta(\log^3 n)$ entries so that they ensure after the reduction, the vectors take values in $\{-1, 0, 1\}$, we instead distribute the mass across $\Theta(\log n)$ entries so that each entry has magnitude $\Theta(\log^2 n)$. To make the reduction work, the total mass of $\Theta(\log^3 n)$ is needed, and for Backurs et al. (2017), it is fine to append an extra $\Theta(\log^3 n)$ entries as their hardness result for hard-margin SVM without bias does require $d = \Theta(\log^3 n)$. For us, we need to restrict $d = \Theta(\log n)$ at the price of each entry has a larger magnitude of $\Theta(\log^2 n)$. This blows up the squared radius from $\log^2 n$ to $\log^6 n$. We note that the Backurs et al. (2017) construction has squared radius $\log^4 n$.

