# OpenReview forum: "Faster Algorithms for Structured Linear and Kernel Support Vector Machines"
_ICLR.cc/2025/Conference — ICLR 2025 Poster_

### Official Review · Reviewer_CgdP · 2024-10-20

**Soundness:** 3
**Presentation:** 2
**Contribution:** 2
**Rating:** 5
**Confidence:** 3

**Summary:**

This paper studies fast algorithms for structured linear and kernel support vector machines (SVMs). SVMs are a type of machine learning model used for tasks such as classification and regression. The main contribution of the paper is to provide an algorithm with almost linear time for solving SVM problems with low tree width or low-rank structure. This is related to sparse patterns in the Cholesky decomposition of a matrix, etc. The paper also considers how to apply the proposed method to SVMs using Gaussian kernels, and provides an estimate of the computational complexity. No numerical investigation is given into the effectiveness of the method for real-world problems, etc.

**Strengths:**

- The authors provide intensive mathematical analysis of robust interior point methods for quadratic optimization problems.

**Weaknesses:**

- The paper states that the dimensionality constraint is d=O(log(n)), but this seems like a very strict assumption for real-world data analysis. It would be good to consider how the computational complexity changes when this constraint is relaxed.
- The effectiveness of the proposed method is unclear because there are no numerical experiments. It would be useful for practitioners to see things like implementation ideas and estimates of the data size that can actually be handled.
- As only the Gaussian kernel is used as the kernel, the range of applicability is narrow. Is it possible to apply the algorithm to other kernels, while maintaining the computational efficiency? How does the computation time of the algorithm relate to the kernel width of the Gaussian kernel?
- For kernel methods, low-rank approximation methods such as the Nystrom approximation have been developed. In addition, it is known that when there are a large number of data points, it is more computationally efficient to use deep neural networks, which have developed significantly in recent years. Given this situation, it is necessary to explain the situations in which the proposed method in this paper is effective based on real-world problems.

**Questions:**

see strengths and weaknesses.

**Comment after the rebuttal period**: the authors' sincere responses resolved some of my concerns. I would like to express my gratitude for their efforts. Therefore, I decided to raise my score. However, I still believe that adding numerical experiments is necessary to ensure acceptance.

---

> ### Author Response · Authors · 2024-11-20
>
> We appreciate the reviewer for their insightful comments, and we would like to address your concerns. Before doing so, we would like to highlight one important contribution that may have been overlooked.
>
> We develop a generic algorithm for convex quadratic programming, with a runtime of $O(n(d+m)^{(\omega+1)/2}\log(1/\epsilon))$, assuming the quadratic objective matrix $Q$ has a rank-$d$ factorization and $m$ is the number of linear constraints. To the best of our knowledge, this is the first algorithm for convex quadratic programming that has linear dependence on $n$ and solves the program to high precision.
>
> As a direct application, we adapt this algorithm to linear SVM: given a size-$n$ dataset in $\mathbb{R}^d$, we can solve the linear SVM in $O(nd^{(\omega+1)/2}\log(1/\epsilon))$ time. Prior fastest algorithms with $\log(1/\epsilon)$ dependence all have super-linear dependence on $n$, making them inefficient for large datasets. Moreover, our approach can be generalized to various kernel SVM schemes where the kernel matrix can be low-rank factored in subquadratic time. While we agree it is important to evaluate the practical performance of the algorithm, we also wish to highlight its theoretical contribution.
>
> ---
>
> ### Q: The paper states that the dimension $d=O(\log n)$ seems to be very restrictive for real-world data analysis.
>
> **A:** As proven in our hardness result (Theorem 1.5), there exists some $d=\Omega(\log n)$ such that the corresponding kernel SVM cannot be solved in $O(n^{2-o(1)})$ time unless the Strong Exponential Time Hypothesis is false. In fact, our proof of Theorem 1.5 explicitly constructs an instance of Gaussian kernel SVM with $d = C\log n$ for some large constant $C$, such that solving it efficiently is hard. Essentially, this implies that the best one can hope for in the $d = \Omega(\log n)$ regime is to explicitly form the kernel matrix by filling all $n^2$ entries. Therefore, we respectfully disagree that our assumption is “restrictive,” as it represents the theoretical best-case scenario.
>
> ---
>
> ### Q: The application to the Gaussian kernel seems narrow. Can it be extended to other kernels? How does the width of the Gaussian kernel affect the runtime?
>
> **A:** Yes, our technique can indeed be extended to a large family of kernels, as parametrized in Alman, Chu, Schild, and Song (2020). Specifically, their work shows that as long as the kernel function takes the form ${\sf K}(u, v) = f(\|u-v\|_2^2)$ and $f$ can be approximated by a low-degree polynomial of degree $o(\log n)$, one can compute a rank-$n^{o(1)}$ factorization of the kernel matrix in $n^{1+o(1)}$ time. This family includes polynomial kernels and $p$-convergent kernels (Song, Woodruff, Yu, and Zhang, 2021).
>
> Regarding the width parameter of the Gaussian kernel, it directly impacts the squared radius of the dataset. Let $R$ denote the original squared radius, and $\sigma$ the width. Then the new squared radius is $\frac{R^2}{2\sigma^2}$. As long as $\frac{R^2}{2\sigma^2} \leq O(\frac{\log n}{\log\log n})$, our algorithm can be applied.
>
> ---
>
> ### Q: How do your methods compare to the Nyström approximation, which also provides a low-rank approximation to the kernel matrix?
>
> **A:** The Nyström method does indeed produce good low-rank approximations to kernel matrices. For instance, Musco and Musco (2017) show that one can construct a rank-$s$ factorization of the kernel matrix using $O(ns)$ kernel function evaluations and an additional $O(ns^{\omega-1})$ time. However, there are two significant challenges in applying the Nyström approximation and then solving the SVM:
>
> 1. **Rank-Approximation Tradeoff**: If one desires a small rank (e.g., $s=n^{o(1)}$), a large regularization parameter $\lambda$ is required, leading to poor approximation.
> 2. **Impact of Regularization**: The additional $\lambda I$ term complicates the downstream SVM algorithm, introducing an error term dependent on the $\ell_2$ norm squared of the optimal solution and $\lambda$. This increases runtime by a factor of $\log(\lambda)$, and there is no clear theoretical guideline for selecting $\lambda$.
>
> ---
>
> ### Q: For large-scale datasets, deep neural networks practically offer better computational complexity. How does your method compare with deep neural networks?
>
> **A:** Theoretically, deep neural networks are non-convex models, making their convergence analysis challenging. The fastest known algorithms for training over-parameterized networks with $m$ neurons per layer require $m \sim n^4$, and the runtime is $O(m^{1.93})$ (Song, Zhang, and Zhang, 2024). Since $m = n^4$, this leads to $O(n^{7.72})$ time—significantly more expensive than our approach, which runs in $O(n^{1+o(1)})$ time. Additionally, kernel SVMs are much simpler and easier to analyze compared to deep neural networks. We believe simpler, convex methods like kernel SVMs are preferable in many settings and designing theoretically sound and efficient algorithms for them is valuable.

---

> > ### Author Response · Authors · 2024-11-20
> >
> > ---
> >
> > ### References
> >
> > - Alman, Chu, Schild, and Song (2020): Algorithms and Hardness for Linear Algebra on Geometric Graphs. *FOCS 2020*.
> > - Song, Woodruff, Yu, and Zhang (2021): Fast Sketching of Polynomial Kernels of Polynomial Degree. *ICML 2021*.
> > - Musco and Musco (2017): Recursive Sampling for the Nyström Method. *NeurIPS 2017*.
> > - Song, Zhang, and Zhang (2024): Training Multi-Layer Over-Parameterized Neural Networks in Subquadratic Time. *ITCS 2024*.

---

> > > ### Comment · Reviewer_CgdP · 2024-11-23
> > >
> > > We would like to sincerely thank the authors for their detailed and insightful responses. The authors' provision of relevant references and clear explanations on each point has addressed several concerns in my review.
> > >
> > > > As for $d=O(\log(n))$ condition:
> > >
> > > I understand that this represents the best theoretical evaluation under the Strong Exponential Time Hypothesis (SETH), and I consider it an outstanding theoretical achievement. However, from the perspective of practical application to real-world problems, it seems reasonable to assume that the condition $d = O(\log(n))$ may not always be satisfied. To address this gap, possible directions could include conducting theoretical evaluations of average-case complexity (using smoothed analysis, etc.) or demonstrating, from a practical perspective, that the algorithm performs effectively even when this condition is violated in real-world data. I look forward to further advancements in this research.
> > >
> > > > Extension of Gaussian kernel:
> > >
> > > It is wonderful to hear that a broader class of kernel functions also works within the theoretical framework presented in this paper. The condition regarding the dependency on the kernel width also appears reasonable, as the commonly used median heuristic approximately results in $\sigma = O(\sqrt{\text{data-dimension}})$ [Ramdas+('15)]. My concern on this point has been fully addressed thanks to the authors' insightful comments. Therefore, I am now ready to raise my score.
> > >
> > > Ramdas+, On the Decreasing Power of Kernel and Distance Based Nonparametric Hypothesis Tests in High Dimensions, AAAI 2015.
> > >
> > > > Practical Performance:
> > >
> > > It is well known that the theoretical analysis of deep neural networks (DNNs) is challenging and is currently an area of intensive research. At the same time, it is also widely recognized that DNNs demonstrate excellent performance from a practical perspective. Personally, I would like to evaluate the usefulness of theoretical methods from a practical standpoint. Even if a method has strong theoretical properties, it does not necessarily guarantee high performance when applied to real-world problems.  I strongly encourage efficient implementations of algorithms as well as their evaluation on real-world data.

---

> > > > ### Author Response · Authors · 2024-11-24
> > > >
> > > > Thank you for your followup. We have updated our manuscript with a discussion on kernel width and the commonly used choice suggested by [Ramdas et al., 2015]. Regarding experiments, we agree with you that it's important to evaluate the performance of a theoretical algorithm on real-world datasets. Due to limited time and computational resources, we are not able to perform experiments for our algorithm. However, we note that this line of works [Cohen et al., 2019; Lee et al., 2019; Jiang et al., 2021; Dong et al., 2021; Gu and Song, 2022] are all theoretical works without experimental evaluations. In fact, a common belief in this area is that the robust interior point method studied in this line of works and our work is efficient in practice, yet it remains a major open question due to the nontrivial interplay of linear algebraic data structures. We leave the practical implementation of the algorithm as an open question as well.
> > > >
> > > > References
> > > >
> > > > Ramdas et al., 2015: On the Decreasing Power of Kernel and Distance Based Nonparametric Hypothesis Tests in High Dimensions, AAAI 2015.
> > > >
> > > > Cohen et al., 2019: Solving Linear Programs in the Current Matrix Multiplication Time. STOC 2019.
> > > >
> > > > Lee et al., 2019: Solving Empirical Risk Minimization in the Current Matrix Multiplication Time. COLT 2019.
> > > >
> > > > Jiang et al., 2021: Faster Dynamic Matrix Inverse for Faster LPs. STOC 2021.
> > > >
> > > > Dong et al., 2021:  A Nearly-Linear Time Algorithm for Linear Programs with Small Treewidth: A Multiscale Representation of Robust Central Path. STOC 2021.
> > > >
> > > > Gu and Song, 2022: A Faster Small Treewidth SDP Solver. 2022.

---

### Official Review · Reviewer_6s2J · 2024-10-30

**Soundness:** 4
**Presentation:** 3
**Contribution:** 3
**Rating:** 8
**Confidence:** 3

**Summary:**

The paper proposed a nearly-linear time algorithm for solving quadratic programming problems, especially designed for kernel SVMs. It combines the robust interior point method, low-treewidth setting, low-rank factorization and low-rank kernel matrix approximation techniques to enable the almost-linear time convergence. Though the assumptions to achieve this convergence rate is relatively strong, it is the first almost-linear convergence algorithm for quadratic programming problems or kernel SVMs.

**Strengths:**

Novel Approach: The paper proposes an innovative nearly-linear time algorithm for solving quadratic programming problems with applications to kernel SVMs. The use of robust interior point method, low-rank factorization and low-rank matrix approximation make the approach computationally appealing.

Relevance: Given the broad usage of SVMs in machine learning, the proposed methodology could provide substantial computational improvements if the assumptions of the method are satisfied.

Complexity Analysis: The paper includes a rigorous complexity analysis and explores different scenarios for kernel SVMs under various dataset radius conditions. These insights provide a clear understanding of the expected performance under different cases. It also adheres to previous literature.

**Weaknesses:**

Empirical Validation: While the theoretical contribution is significant, empirical results demonstrating the practical runtime improvements and accuracy on real datasets would be beneficial. This would help validate the theoretical findings and highlight the algorithm’s scalability and precision.

Assumptions and Constraints: The assumptions, especially regarding the squared radius, may limit applicability in some scenarios. The paper could benefit from discussing the practical implications of these assumptions on dataset characteristics.

Comparison with Existing Methods: Though the theoretical basis is sound, a more detailed comparison with existing SVM solvers (e.g., LIBSVM, SVM-Light) in terms of computational complexity and memory usage could strengthen the contribution by illustrating the specific benefits and trade-offs of the proposed algorithm.

**Questions:**

1.	In section 1, the authors denote $B$ as the squared radius, then in section 2, the authors redefine $B = Q + t H$. Please make sure the notation is constant.

2.	Though the theoretical results of this paper are very strong, it is highly recommended that the authors could do some experiments to verify the theoretical results.

3.	It seems the assumption that square radius $B = O(log(n)/loglog(n))$ is very strong. It would be great if the authors could share some kernel SVM problems which satisfy this assumption.

4.	Overall, in section 2, The technical details are very heavy but not clear. It would great if the authors could give more intuitions about these approaches. For example, consider a more detailed discussion on how the low-rank factorization impacts specific types of SVM problems or datasets.

5.	Small typos: for example, in line 1540, is it “memebers” or “members”? Please double check.

---

> ### Author Response · Authors · 2024-11-20
>
> We thank the reviewer for their thoughtful comments and insights. Regarding your suggestion to compare with algorithms such as LIBSVM or SVM-Light, it is worth noting that these algorithms are all first-order methods with runtime dependence polynomially on $\epsilon^{-1}$. As we illustrated in the manuscript, this leads to a significant runtime discrepancy even when $\epsilon = 10^{-4}$. Below, we address your questions in detail:
>
> ---
>
> ### Q: In Section 1, the authors used $B$ as the squared radius, while in Section 2, it was used as $B = Q + tH$. It would be good to make them consistent.
>
> **A:** Thank you for pointing this out. We have updated the notation for $Q + tH$ to $M$ to avoid confusion and ensure consistency.
>
> ---
>
> ### Q: It seems the assumption that the squared radius $B = O(\log n / \log\log n)$ is very strong.
>
> **A:** Indeed, this assumption might appear restrictive at first glance. However, as our lower bound result shows, assuming the Strong Exponential Time Hypothesis, even for $B = \Omega(\log^2 n)$, there is no algorithm that solves the Gaussian kernel SVM in $O(n^{2-o(1)})$ time. This implies that one would need to explicitly form the kernel matrix before running a downstream algorithm to solve the SVM. We conjecture that the lower bound can be strengthened to $B = \Omega(\log n)$, which would reflect a phase transition for fast algorithms in kernel linear algebra, as indicated by the works of Alman, Chu, Schild, and Song (2020) and Aggarwal and Alman (2022).
>
> ---
>
> ### Q: Section 2 is very heavy in technical details. Could the authors provide some intuitions behind these techniques?
>
> **A:** Thank you for this suggestion. The current structure of Section 2 highlights the challenges in solving the primal-dual central path equations of QPs efficiently. We further demonstrate that, if the quadratic objective matrix has certain structures, such as low-rank or low-treewidth, it is possible to overcome these obstacles and design efficient algorithms. These techniques rely on crafting nontrivial maintenance data structures and performing robustness analyses to ensure that the approaches converge to a solution.
>
> We believe that Section 1 provides a comprehensive summary of our results, including comparisons with other works and potential applications. Additionally, Section 2.1 offers a high-level overview of the generic framework for solving the program, while Section 2.4 discusses how these techniques extend to the kernel SVM setting. To address your feedback, we have added a paragraph recommending readers to skip Sections 2.2 and 2.3 on their first read, as they are highly technical. We hope this adjustment will make the manuscript more accessible without sacrificing the necessary details.
>
> ---
>
> ### Q: There are typos such as on line 1540, where “memebers” should be “members.”
>
> **A:** We appreciate you pointing this out. We have corrected the typos and conducted a thorough proofread of the manuscript to ensure it is free of errors.
>
> ---
>
> ### References
>
> - Alman, Chu, Schild, and Song (2020): Algorithms and Hardness for Linear Algebra on Geometric Graphs. *FOCS 2020*.
> - Aggarwal and Alman (2022): Optimal-degree polynomial approximations for exponentials and Gaussian kernel density estimation. *CCC 2022*.

---

> > ### Comment · Reviewer_6s2J · 2024-11-27
> > **Reply to the Authors**
> >
> > 1. Thanks for the update.
> >
> > 2. The authors can later try to see if the lower bound can be strengthened to $B = \Omega(\log n)$. At this point, no action is needed.
> >
> > 3. Thanks for adding a paragraph to give the readers a better intuition.
> >
> > 4. Thanks for cleaning the typos.
> >
> > Thanks for your hard work, I've raised my score.

---

### Official Review · Reviewer_WSha · 2024-11-03

**Soundness:** 4
**Presentation:** 4
**Contribution:** 4
**Rating:** 8
**Confidence:** 4

**Summary:**

The authors develop faster quadratic program solvers using interior point methods for problems where the quadratic form matrix has low rank. The technique is then applied to yield SVM solvers with complexity linearly/nearly-linearly in the number of samples for linear/kernel SVMs.

**Strengths:**

SVMs are classical subjects in machine learning, and thoroughly understanding the time complexity of optimizing SVMs is important. This paper takes an important step toward this. The technique for solving QP faster using IPM with the efficient Hessian updates could be beneficial for other problems in machine learning.

**Weaknesses:**

While this paper addresses a theoretical problem in complexity of SVMs, it'd still be interesting to see some actual experimental results.

**Questions:**

The $O(\epsilon^{-2})$ time copmlexity mentioned on Line 121 was sharpened to $O(\epsilon^{-1})$ in [1]. Of course the point that the author makes about the logarithmic dependency still stands, but I think the improvement in [1] should be mentioned since the issue raised on line 124 is alleviated to some degree.

The complexity lower bound on the SVMs considered also implies complexity lower bound on QPs, correct? How does this compare to existing complexity lower bounds for QPs, if any?

Is there a reference for the result on $T$ on line 291?

What is the graph structure on the bags on line 310?

Can the authors give references for the sparse Cholesky factory result on Line 318?

What happens in the kernel SVM case when $d$ is constant (both for the upper bound and lower bound)? Why must $d$ grow with $n$ in order for the result to hold? For the lower bound, I presume it has to do with leveraging hardness results in [2]. Can the authors explain more about the $d = \Theta (\log n)$ assumption?

---

[1] Shalev-Shwartz, Shai, Yoram Singer, and Nathan Srebro. "Pegasos: Primal estimated sub-gradient solver for svm." Proceedings of the 24th international conference on Machine learning. 2007.

[2] Alman, Josh, and Ryan Williams. "Probabilistic polynomials and hamming nearest neighbors." 2015 IEEE 56th Annual Symposium on Foundations of Computer Science. IEEE, 2015.

---

> ### Author Response · Authors · 2024-11-20
>
> We thank the reviewer for their insightful comments and for pointing out the reference that establishes an $O(\epsilon^{-1})$ iteration complexity for first-order methods. We have incorporated a discussion of this work into our updated manuscript, with the changes highlighted in red. Regarding your questions:
>
> ---
>
> ### Q: The lower bound for SVM also gives a lower bound for QP. How does this compare to other existing lower bounds for QPs?
>
> **A:** We are not aware of any direct lower bounds specifically for QPs. However, in addition to the work of Backurs, Indyk, and Schmidt (2017) and our own, there are lower bounds established for linear programs, such as those in Kyng, Wang, and Zhang (2020) and Ding, Kyng, and Zhang (2022). These works show that any linear program with polynomially bounded integer entries can be converted into a 2-commodity flow instance with $O(\mathrm{nnz}(A))$ edges. Consequently, for any $a > 1$, if one can solve an integer-capacitated 2-commodity flow in $O(|E|^a \mathrm{poly} \log(\epsilon^{-1}))$ time, this would imply the existence of an LP solver with a runtime of $O(\mathrm{nnz}(A)^a \mathrm{poly} \log(\epsilon^{-1}))$. Since LPs are a subclass of QPs, this provides a type of lower bound for QPs, albeit without directly addressing the quadratic objective matrix $Q$.
>
> ---
>
> ### Q: Is there a reference for the result on $T$ in line 291?
>
> **A:** Yes, this result is standard in the primal-dual IPM literature. See, for instance, Ye (2020), Dong, Lee, and Song (2021), Gu and Song (2022), and it is also implicit in Cohen, Lee, and Song (2019).
>
> ---
>
> ### Q: What is the graph structure of the bags mentioned in line 310?
>
> **A:** A "bag" is simply a collection of vertices satisfying the following properties:
>
> 1. The union of all the bags equals $V$.
> 2. For any edge $(u, v) \in E$, there exists a bag that contains both $u$ and $v$.
> 3. If each bag is treated as a supervertex and edges are added accordingly (i.e., for two bags $B_1$ and $B_2$, if there exists $u \in B_1$, $v \in B_2$, and $(u, v)$ is an edge, then $B_1$ and $B_2$ are connected in the resulting tree), the induced graph is a tree.
>
> This structure explains the term "tree decomposition." Furthermore, for any vertex $u \in V$, all the bags containing $u$ must form a subtree. For more details, please refer to Definition A.1 in the appendix.
>
> ---
>
> ### Q: Can the authors provide references on the Cholesky factorization mentioned in line 318?
>
> **A:** Sparse Cholesky factorization for matrices with sparsity patterns corresponding to graphs of small treewidth was first proposed in Schreiber (1982) and later developed in Davis and Hager (1999). More recent applications of this procedure appear in Ye (2020) for linear programs, Dong, Lee, and Ye (2021), and Gu and Song (2022) for semidefinite programs. For further exposition, see Section A.3.
>
> ---
>
> ### Q: Regarding $d$ and $\log n$: Must $d$ grow with $n$? Can the authors explain the assumption $d = \Theta(\log n)$?
>
> **A:** Thank you for highlighting this. Upon review, we recognize this as an oversight. The correct assumptions are as follows:
>
> - For our almost-linear time algorithm, we require $d = O(\log n)$.
> - For our hardness result, we require $d = \Omega(\log n)$.
>
> Our lower-bound instance has a dimension $\Omega(\log n)$. The origin of this lower bound traces back to Alman and Williams (2015), which establishes hardness results for Bichromatic Hamming Closest-Pair using Orthogonal Vectors. The precise hardness result we used is from Rubinstein (2018). We have revised all occurrences of $d = \Theta(\log n)$ to $d = O(\log n)$ or $d = \Omega(\log n)$, and these changes have been marked in red.
>
> ---
>
> ### References
>
> - Backurs, Indyk, and Schmidt (2017): On the Fine-Grained Complexity of Empirical Risk Minimization: Kernel Methods and Neural Networks. *NeurIPS 2017*.
> - Kyng, Wang, and Zhang (2020): Packing LPs are Hard to Solve Accurately, Assuming Linear Equations are Hard. *SODA 2020*.
> - Ding, Kyng, and Zhang (2022): Two-Commodity Flow Is Equivalent to Linear Programming Under Nearly-Linear Time Reductions. *ICALP 2022*.
> - Ye (2020): Fast Algorithm for Solving Structured Convex Programs. 2020.
> - Dong, Lee, and Ye (2021): A Nearly-Linear Time Algorithm for Linear Programs with Small Treewidth: A Multiscale Representation of Robust Central Path. *STOC 2021*.
> - Gu and Song (2022): A Faster Small Treewidth SDP Solver. 2022.
> - Cohen, Lee, and Song (2019): Solving Linear Programs in the Current Matrix Multiplication Time. *STOC 2019*.
> - Schreiber (1982): A New Implementation of Sparse Gaussian Elimination. *TOMS 1982*.
> - Davis and Hager (1999): Modifying a Sparse Cholesky Factorization. *SIAM Journal on Matrix Analysis and Applications 1999*.
> - Alman and Williams (2015): Probabilistic Polynomials and Hamming Nearest Neighbors. *FOCS 2015*.
> - Rubinstein (2018): Hardness of Approximate Nearest Neighbor Search. *STOC 2018*.

---

> > ### Comment · Reviewer_WSha · 2024-11-25
> >
> > Thanks, I've raised my score. Sounds like hardness results for $d = O ( \log n )$, and in particular, constant $d$, is still open (please correct me if I'm wrong). I think that is also an interesting case, and I'd highly encourage the authors to briefly touch upon this (one sentence) if possible. Also, it sounds to me like this open problem is common to other complexity theoretic results.

---

### Official Review · Reviewer_Vy1p · 2024-11-25

**Soundness:** 3
**Presentation:** 3
**Contribution:** 3
**Rating:** 6
**Confidence:** 4

**Summary:**

The authors design the first nearly-linear time algorithm for solving SVM problems via quadratic programs whenever the quadratic objective admits a low-rank factorization, and the number of linear constraints is small. They have derived the algorithm complexity analysis with different cases, which improves the existing result.

**Strengths:**

The paper clearly states the contribution, the story is easy to follow. This is a solid work on solving SVM dual problem based on quadratic programming.

**Weaknesses:**

1. No experiment to demonstrate the superiority to validate the new claimed contributions.
2. The result works for vanilla SVM and Gaussian Kernel, however, the analysis on other kernel options such as polynomial is blank, thus, whether it can be generalized into broader applications remains unknown.
3. What if the kernel is a combination of kernels, say K_new=K1+K2 where K1 and K2 are Guassian kernel?

**Questions:**

1. I know Eq. (1) is quadratic programming, but I believe the inequality constraint should be generalized into $Ax \le 0$, instead of simple $x\le 0$, though it is a special case.
2. For example, when ε is set to be 10^{−3} to account for the usual machine precision errors, these algorithms would require at least 10^6 iterations to converge. I don't understand this, as the complexity is O(d/ε), I thought it should be 1000*d.
3. I know this is a theoretic paper, but I still hope the authors can compare the convergence with others which will be more convincing to reviewers and audiences. My experience told me that though libsvm is old, the speed is amazingly fast. I definitely hope the proposed algorithm can outperform libsvm.

---

> ### Author Response · Authors · 2024-11-25
>
> We appreciate the reviewer for their thoughtful comments. We address your concerns below.
>
> ---
>
> ### Concern: The result works for linear SVM and Gaussian kernels; however, the analysis on other kernel options, such as polynomial kernels, is missing.
>
> **Answer:** Our technique can indeed be extended to other kernels. As indicated in Alman, Chu, Schild, and Song (2020), as long as the kernel function takes the form ${\sf K}(u, v) = f(\\|u - v\\|_2^2)$ and $f$ can be approximated by a low-degree polynomial of degree $o(\log n)$, there exists a rank-$n^{o(1)}$ factorization of the kernel matrix that can be computed in $O(n^{1+o(1)})$ time. This family includes a large subclass of polynomial kernels called $p$-convergent kernels (Song, Woodruff, Yu, and Zhang, 2021).
>
> ---
>
> ### Concern: What if the kernel is a combination of kernels, such as $K_{\rm new} = K_1 + K_2$, where both $K_1$ and $K_2$ are Gaussian kernels?
>
> **Answer:** In this case, we could run our algorithm separately on $K_1$ and $K_2$. Suppose $K_1$ has width $\sigma_1$, $K_2$ has width $\sigma_2$, and let $R$ denote the radius of the dataset. As long as $\frac{R^2}{2\sigma_1^2}, \frac{R^2}{2\sigma_2^2} \leq o(\frac{\log n}{\log\log n})$, our algorithm can solve each individual kernel in $O(n^{1+o(1)})$ time. The results for $K_1$ and $K_2$ can then be combined.
>
> ---
>
> ### Question: The inequality constraint should be generalized into $Ax \leq b$ instead of $x \leq 0$.
>
> **Answer:** We are using the slack form of convex quadratic programming, where the constraint region is the intersection of a linear subspace ($Ax = b$) and the non-negative orthant ($x \geq 0$). The formulation with $Ax \leq b$ that you are referring to is called the standard form, where the constraint region is the intersection of the polytope $Ax \leq b$ and the non-negative orthant $x \geq 0$.
>
> One can reduce a standard form constraint to the slack form by increasing the dimension from $n$ to $2n$ and introducing slack variables: $Ax + s = b$, $x \geq 0$, $s \geq 0$. Slack form is commonly used for designing algorithms, such as the simplex method. Here, we also adopt the slack form and note that this formulation is without loss of generality.
>
> ---
>
> ### Question: When $\epsilon = 10^{-3}$, the best first-order algorithm takes time $\widetilde{O}(\epsilon^{-1} d)$, so it would take $10^3$ iterations rather than $10^6$.
>
> **Answer:** Thank you for pointing this out. We have corrected this in our updated manuscript.
>
> ---
>
> ### Question: It would be better if you could perform some experiments and compare against LIBSVM.
>
> **Answer:** We agree that it is important to evaluate the performance of a theoretical algorithm on real-world datasets. However, due to limited time and computational resources, we were unable to perform experiments for our algorithm.
>
> We note that this line of works (Cohen, Lee and Song, 2019; Lee, Song and Zhang, 2019; Jiang, Song, Weinstein and Zhang, 2021; Dong, Lee and Ye, 2021; Gu and Song, 2022) are all theoretical studies without experimental evaluations. In fact, a common belief in this area is that the robust interior point method studied in these works, as well as our work, is efficient in practice. However, this remains a significant open question due to the nontrivial interplay of linear algebraic data structures. We leave the practical implementation of the algorithm as an open problem.
>
> ---
>
> ### References
>
> - Alman, Chu, Schild, and Song (2020): *Algorithms and Hardness for Linear Algebra on Geometric Graphs*. FOCS 2020.
> - Song, Woodruff, Yu, and Zhang (2021): *Fast Sketching of Polynomial Kernels of Polynomial Degree*. ICML 2021.
> - Cohen, Lee and Song (2019): *Solving Linear Programs in the Current Matrix Multiplication Time*. STOC 2019.
> - Lee, Song and Zhang (2019): *Solving Empirical Risk Minimization in the Current Matrix Multiplication Time*. COLT 2019.
> - Jiang, Song, Weinstein and Zhang (2021): *Faster Dynamic Matrix Inverse for Faster LPs*. STOC 2021.
> - Dong, Lee and Ye (2021): *A Nearly-Linear Time Algorithm for Linear Programs with Small Treewidth*. STOC 2021.
> - Gu and Song (2022): *A Faster Small Treewidth SDP Solver*. 2022.

---

### Author Response · Authors · 2024-11-30
**Thanks reviewer for the comments, feedback and discussions**

We sincerely thank the reviewers for their time, efforts, and constructive feedback on our manuscript. The comments have been invaluable in enhancing the clarity and presentation of the paper. We have addressed each of the reviewer comments and reflected the changes in red in the manuscript. We hope these revisions address your concerns. We remain available to any further feedback or clarifications and look forward to more discussions until the end of the discussion period.

---

### Meta-Review · Area_Chair_am4n · 2024-12-19

**Metareview:**

This paper investigates solving SVMs via quadratic programming and proposes a novel, efficient, nearly-linear time algorithm for cases where the quadratic objective admits a low-rank factorization and the number of linear constraints is small—settings that are typical for SVMs. The results presented are of broad interest to the ML and optimization communities.

The reviewers reached a consensus that the contribution of this work is significant, and I therefore recommend accepting the paper. However, as suggested by **all** reviewers, it would be beneficial if the authors could provide numerical results, in some form, to demonstrate the practical numerical advantages of the proposed algorithm.
Including such results would likely enhance the paper’s impact within the community.

**Additional Comments On Reviewer Discussion:**

The reviewers raised the following points:

- Lack of empirical support (raised by **all** reviewers): While the authors have chosen **not** to include **any** experimental evaluations in this paper, the reviewers appear generally satisfied with the authors’ response on this matter.
- Clarifications on specific aspects: These include the application to general kernels (raised by Reviewers Vy1p and CgdP), the assumption of  $d = \Theta(\log n)$  (raised by Reviewers WSha and CgdP), and comparisons to existing results (raised by Reviewers 6s2J and CgdP). The authors successfully addressed these concerns during the rebuttal, leading to improved scores from the reviewers.

I have carefully considered all of the above points in making my final decision.

---

### Decision · Program_Chairs · 2025-01-22

Accept (Poster)